palaeontology/ecology/taxonomy and systematics

Australia, mammal, taxonomy, biogeography, Pliocene, Pleistocene, locomotion

**Author for correspondence:**
Natalie M. Warburton
e-mail: n.warburton@murdoch.edu.au

# The skeleton of *Congruus kitcheneri*, a semiarboreal kangaroo from the Pleistocene of southern Australia

## Natalie M. Warburton[1,2] and Gavin J. Prideaux[3]

[1]Centre for Climate-Impacted Terrestrial Ecosystems, Harry Butler Research Institute, Murdoch University, Australia
[2]Department of Earth and Planetary Sciences, Western Australian Museum, Kew Street, Welshpool, WA, Australia
[3]College of Science and Engineering, Flinders University, South Australia 5042, Australia

NMW, 0000-0002-8498-3053

The macropodine kangaroo, *Wallabia kitcheneri*, was first described in 1989 from a Pleistocene deposit within Mammoth Cave, southwestern Australia, on the basis of a few partial dentaries and maxilla fragments. Here, we recognize *W. kitcheneri* within the Pleistocene assemblages of the Thylacoleo Caves, south-central Australia, where it is represented by several cranial specimens and two near-complete skeletons, a probable male and female. We reallocate this species to the hitherto monotypic genus *Congruus*. *Congruus kitcheneri* differs from all other macropodid species by having a highly unusual pocket within the wall of the nasal cavity. It is distinguished from *C. congruus* by having a longer, narrower rostrum, a taller occiput and a deeper jugal. *Congruus* is closest to *Protemnodon* in overall cranial morphology but is smaller and less robust. In most postcranial attributes, *Congruus* also resembles *Protemnodon*, including general limb robustness and the atypical ratio of 14 thoracic to five lumbar vertebrae. It is distinguished by the high mobility of its glenohumeral joints, the development of muscle attachment sites for strong adduction and mobility of the forelimb, and large, robust manual and pedal digits with strongly recurved distal phalanges. These adaptations resemble those of tree-kangaroos more than ground-dwelling macropodines. We interpret this to imply that *C. kitcheneri* was semiarboreal, with a propensity to climb and move slowly through trees. This is the first evidence for the secondary adoption of a climbing habit within crown macropodines.

# 1. Introduction

Macropodids (kangaroos and relatives) descended from arboreal possum-like ancestors during the Palaeogene before becoming the main ground-dwelling mammalian herbivores of the Australian continent over the past 20 million years (Myr) [1]. There are more than 60 extant species, including bettongs, wallabies and kangaroos, and many more extinct taxa, including giant short-faced kangaroos. As in ungulates, smaller extant macropodids are mainly solitary fungivores or browsers (dicot consumers), while larger species tend to be gregarious grazers (grass consumers) [2]. The greatest diversity within the group is expressed by the subfamily Macropodinae, most genera of which probably originated during the late Miocene into very early Pliocene [3,4].

Eleven of the 12 extant macropodine genera are characterized by species that are principally ground dwelling. They employ a bipedal hopping mode of locomotion when moving at speed, and a 'pentapedal' mode, involving the use of the tail as a 'fifth limb' when moving slowly [5]. These locomotory styles are facilitated by striking adaptations in physiology and skeletal anatomy, including elongation of the hindlimbs and feet, and reconfiguration of the ankle joint to minimize lateral and rotational movement during hopping [1]. The one extant genus that deviates from this pattern is *Dendrolagus* Müller, 1840 [6], which contains the tree-kangaroos of New Guinea and extreme northeastern Australia. They descended from ground-dwelling ancestors by ascending into the trees. Tree-kangaroos can hop bipedally as well as move their hindlimbs alternately [7]. Their arboreal adeptness is reflected in a range of skeletal modifications manifested also within the larger, extinct species of the genus *Bohra* Flannery & Szalay, 1982 [8], which are known from the Pliocene and Pleistocene [1,9–12]. The occurrence of two species of *Bohra* within the Pleistocene Thylacoleo Caves deposits of south-central Australia, which have been inferred to have accumulated under a moderate to relatively low rainfall regime [13–15], reveals that the modern association between tree-kangaroos and wetter forest habitats is an artefact of differential extinction in the late Quaternary.

Insights generated by our studies of the *Bohra* skeleton have provided the impetus to take a closer look at the morphology of other Pleistocene kangaroos, first and foremost other macropodine taxa represented in the Thylacoleo Caves of the Nullarbor Plain, south-central Australia (figure 1). These caves preserve Pleistocene vertebrate assemblages noteworthy for their exceptional fossil preservation and taxonomic diversity [13,16]. One such macropodine is represented by two adult skeletons, herein argued to represent a male and a female. We recognize them as specimens of *Wallabia kitcheneri* Flannery, 1989 [17], first described on the basis of several partial dentaries from the Pleistocene of southwestern Australia. This species has been reallocated to the genus *Congruus* McNamara, 1994 [18], on various species lists (e.g. [13,16,19,20]), but never with an accompanying taxonomic justification. This paper provides that justification, describes the skeletal morphology of the species and reflects on inferred sexual differences and locomotory adaptations. A complete taxonomic revision of *Congruus*, including an assessment of material from eastern Australia, and a refinement of its phylogenetic position, will be presented in forthcoming papers.

# 2. Materials and methods

Description style, terminology and mensuration follow Murray [21], Wells & Tedford [22], Prideaux [23], Prideaux & Warburton [1,9,10] and Warburton & Prideaux [24]. Serial designation of the cheek dentition follows Flower [25] and Luckett [26]. Measurements were taken with Mitutoyo digital callipers (Kawasaki, Japan) and are presented in tables 1–9. Dentary depth and width are measured below the abutment of m2 and m3. Upper teeth are designated by upper case abbreviations (e.g. P3, M2); lower teeth by lower case abbreviations (e.g. i1, m3). Deciduous premolars are denoted by the prefix 'd' (e.g. dP2, dp3). Crown dimensions: L, length (along tooth midline); AW, anterior width (above anterior root); PW, posterior width (above posterior root); AH, anterior height (buccal side); PH, posterior height (buccal side). Molar crown height was measured only for minimally worn specimens (i.e. no exposed dentine).

Direct comparison of dental and bony morphology was made with extant and extinct macropodine taxa of medium to large body size, namely species of *Bohra*, *Dendrolagus*, *Lagorchestes*, *Macropus*, *Notamacropus*, *Onychogalea*, *Osphranter*, *Petrogale*, *Protemnodon*, *Setonix*, *Thylogale* and *Wallabia*. These taxa capture much of the locomotor diversity exhibited within the subfamily. Comparative specimens are housed in the collections of Flinders Palaeontology Adelaide, Murdoch University Veterinary

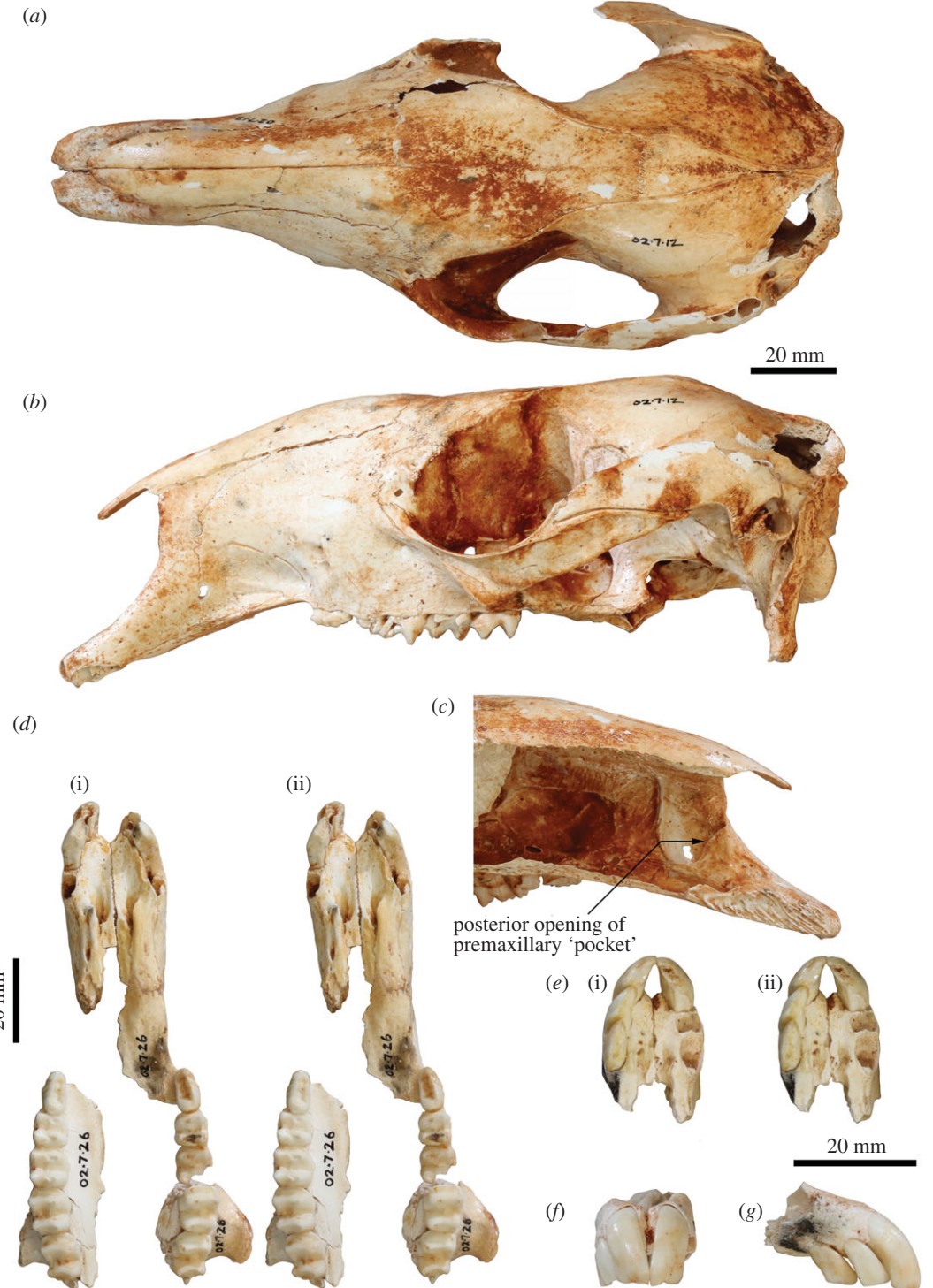

**Figure 1.** Adult cranium and upper dentition of *Congruus kitcheneri*. Adult cranium (WAM 02.7.12) in (*a*) dorsal view and (*b*) lateral view. (*c*) Mesial view of nasal cavity of WAM 02.7.12 premaxilla showing location of premaxillary 'pocket'. (*d*) WAM 02.7.26 in occlusal view (stereo). Juvenile premaxilla (WAM 02.7.21) in (*e*) occlusal view (stereo), (*f*) anterior view and (*g*) right lateral view.

Anatomy Department Perth, Western Australian Museum (WAM) Perth, Queensland Museum Brisbane and Museums Victoria, Melbourne.

   Although direct comparison between relative adaptations of marsupial and placental mammals can be problematic [27,28] due to their distinct evolutionary trajectories, form–function parallels nevertheless exist. Differences in skeletal anatomy often reflect different modes of habitat use, including arboreal versus ground-dwelling locomotor adaptations and feeding. As such, functional studies of musculoskeletal systems of placentals were helpful for informing our interpretations.

**Table 1.** Dimensions (in mm) of the adult cranium of *Congruus kitcheneri* and *C. congruus*.

| species<br>dimension | *Congruus kitcheneri* | | *Congruus congruus* |
|---|---|---|---|
| | WAM 02.7.12 | WAM 03.5.3 | SAM P33475 |
| cranial (condylobasal) length | 181.65 | 181.47 | 160.69 |
| cranial width (across zygomatic arches) | 80.47 | 85.05 | 77.94 |
| rostrum length (anterior edge of orbit to I1) | 88.11 | 84.89 | 76.61 |
| width across frontals (above centre of orbit) | 39.88 | 42.66 | 41.76 |
| distance between masseteric processes | 66.76 | 67.69 | 60.83 |
| distance between tips of paroccipital processes | 49.31 | 42.55 | 46.82 |
| occipital height | 56.62 | 55.91 | 48.22 |
| diastema length | 46.79 | 43.32 | 37.68 |
| palatal length | 112.36 | 109.93 | 96.16 |
| palatal width at M1 protoloph | 27.90 | 28.03 | 27.68 |
| palatal width at M4 protoloph | 31.94 | 29.60 | |

**Table 2.** Dimensions (in mm) of the adult dentary of *Congruus kitcheneri*.

| dimension | WAM 02.7.12 | WAM 02.7.15 | WAM 02.7.26 | WAM 03.5.3 | WAM 19.7.202 |
|---|---|---|---|---|---|
| dentary length | 160.21 | | | | |
| dentary depth | 18.14 | 17.40 | 23.2 | | 23.3 |
| dentary width | 10.71 | 8.84 | 12.4 | 10.74 | 12.4 |

# 3. Results

## 3.1. Systematic palaeontology

**Diprotodontia** Owen, 1866 [29]
**Phalangeriformes** Woodburne, 1984 [30]
**Macropodidae** Gray, 1821 [31]
**Macropodinae** Gray, 1821 [31]
**Macropodini** Gray, 1821 [31]

*Congruus* McNamara, 1994 [18]

**Type species:** *Congruus congruus* McNamara, 1994 [18], by monotypy.
**Included species:** *Congruus kitcheneri* (Flannery, 1989) [17]
**Revised diagnosis:** The two species of *Congruus* are united, to the exclusion of all other macropodine taxa, by a unique combination of cranial and upper dental features, rather than any single distinctive trait. Features of the dentary, lower dentition and postcranial skeleton of *C. kitcheneri* are treated as generic traits until they are described for another species of *Congruus*.

In overall cranial morphology, *Congruus* is most similar to *Dorcopsis, Wallabia* and *Protemnodon brehus*. It differs from *Dorcopsis* by having smaller orbits, a narrower diastema region, no C1, a deeper, more domed rostrum, a more anteriorly extended incisor-bearing portion of the premaxilla and a shorter P3 relative to the molars. It differs from *Pr. brehus* by having a shallower zygomatic arch, a smaller masseteric process, an anteriorly less robust premaxilla, a more domed nasal, a less concave frontal region and a longer P3 relative to the molars. The doming of the nasal region resembles that of *Ony. unguifera*. The sulcus on the mesial surface of the premaxilla of *C. congruus* is similar to that observed in some species of *Osphranter*, *Dendrolagus* and *Lagorchestes*, but the enlarged 'pocket' that characterizes *C. kitcheneri* is unique. The entirely bony (non-fenestrate) secondary palate distinguishes *Congruus* from all macropodine genera, except for *Dendrolagus, Bohra, Protemnodon, Osphranter* and *Macropus*.

**Table 3.** Upper premolar dimensions (mm) of *Congruus kitcheneri* and *C. congruus*.

| species/specimen | dP2 | | | | | dP3 | | | | | P3 | | | | |
|---|---|---|---|---|---|---|---|---|---|---|---|---|---|---|---|
| | L | AW | PW | AH | PH | L | AW | PW | AH | PH | L | AW | PW | AH | PH |
| *Congruus kitcheneri* | | | | | | | | | | | | | | | |
| WAM 66.8.18/66.9.29 | | | | | | | | | | | 10.94 | 4.20 | 5.30 | | |
| WAM 02.7.15 | | | | | | 7.22 | 5.29 | 5.00 | 3.84 | 4.59 | 11.49 | 4.91 | 5.71 | 6.06 | 6.76 |
| WAM 02.7.21 | | | | | | | | | | | 9.96 | 4.31 | 4.80 | 4.42 | 5.59 |
| WAM 02.7.26 | | | | | | 8.23 | 5.78 | 6.63 | 3.88 | 4.72 | 10.89 | 5.16 | 6.17 | 5.59 | 5.77 |
| WAM 19.7.201 | 6.75 | 4.36 | 4.62 | 4.72 | 4.84 | | | | | | | | | | |
| *Congruus congruus* | | | | | | | | | | | | | | | |
| SAM P33475 | | | | | | | | | | | 9.07 | 4.49 | 4.95 | 4.83 | 4.08 |

**Table 4.** Lower premolar dimensions (mm) of *Congruus kitcheneri*.

| specimen | dp2 | | | | | dp3 | | | | | p3 | | | | |
|---|---|---|---|---|---|---|---|---|---|---|---|---|---|---|---|
| | L | AW | PW | AH | PH | L | AW | PW | AH | PH | L | AW | PW | AH | PH |
| WAM 66.8.17/66.9.47 | | | 3.26 | | 3.67 | 6.50 | 4.11 | 4.48 | | | 8.60 | 3.10 | 3.70 | 4.98 | 5.43 |
| WAM 66.9.40 | 7.03 | 3.74 | 4.23 | 3.50 | 3.67 | 7.75 | 5.08 | 5.22 | 5.22 | 5.73 | | | | | |
| WAM 66.9.72 | 6.14 | 2.71 | 2.91 | 3.03 | 3.75 | 6.28 | 3.60 | 4.22 | | 4.25 | 8.08 | | | 4.37 | 4.48 |
| WAM 02.7.12 | | | | | | | | | | | 9.56 | 4.11 | 4.77 | 5.66 | 5.89 |
| WAM 02.7.15 | | | | | | | | | | | 10.70 | 4.01 | 4.99 | 7.09 | 6.95 |
| WAM 02.7.21 | 6.01 | 3.35 | 3.64 | 4.10 | 4.15 | 6.35 | 4.14 | 4.15 | 3.54 | 3.32 | 9.74 | 3.37 | 4.14 | 4.76 | 5.43 |
| WAM 02.7.22 | | | | | | 6.11 | 4.05 | 4.10 | 3.98 | 4.18 | | | | | |
| WAM 02.7.26 | | | | | | | | | | | 9.81 | 4.12 | 4.87 | 5.62 | 6.25 |
| WAM 19.7.202 | | | | | | | | | | | 10.90 | 3.90 | 4.43 | 6.22 | 6.16 |
| WAM 19.7.210 | 6.38 | 3.23 | 3.45 | 7.07 | 3.72 | 6.77 | 3.64 | 4.48 | 4.49 | 5.58 | | | | | |

**Table 5.** Upper molar dimensions (mm) of *Congruus kitcheneri* and *C. congruus*.

| species/specimen | M1 | | | | | M2 | | | | | M3 | | | | | M4 | | | | |
|---|---|---|---|---|---|---|---|---|---|---|---|---|---|---|---|---|---|---|---|---|
| | L | AW | PW | AH | PH | L | AW | PW | AH | PH | L | AW | PW | AH | PH | L | AW | PW | AH | PH |
| *Congruus kitcheneri* | | | | | | | | | | | | | | | | | | | | |
| WAM 66.8.18/66.9.29 | 7.23 | 7.04 | 7.60 | | | | | | | | | | | | | | | | | |
| WAM 02.7.12 | 7.40 | 7.47 | | | | 10.58 | | 8.36 | | | 10.69 | 9.12 | | 3.87 | | 11.94 | 8.97 | 7.66 | 4.32 | 5.42 |
| WAM 02.7.15 | | | | | | | | | | | 11.84 | 9.10 | 8.61 | 5.18 | 5.89 | 11.98 | 10.49 | 8.23 | 5.62 | 5.75 |
| WAM 02.7.21 | 7.85 | 6.59 | 6.7 | 3.46 | 4.53 | 9.87 | 7.42 | 7.36 | 4.88 | 5.32 | 10.57 | 8.31 | 7.59 | 5.11 | 5.39 | | | | | |
| WAM 02.7.26 | 8.00 | 8.06 | 8.17 | 6.17 | 5.55 | 9.89 | 9.00 | 9.08 | 6.52 | 5.25 | 11.75 | 9.45 | 9.54 | 6.41 | 5.66 | 11.89 | 8.66 | 8.54 | 6.63 | 5.24 |
| WAM 03.5.3 | 8.40 | | | | | 10.38 | 9.33 | 9.68 | | | 11.21 | 10.18 | 9.03 | 5.29 | 5.70 | 12.24 | 9.76 | 8.10 | 5.24 | 5.70 |
| WAM 03.5.6 | 8.96 | 7.76 | 7.68 | 3.88 | 4.27 | 10.51 | 8.92 | 8.55 | 4.64 | 5.17 | 11.47 | 8.54 | 7.76 | 5.07 | 5.01 | | | | | |
| WAM 19.7.211 | 8.04 | 7.17 | 8.04 | 4.83 | 5.38 | 9.99 | 8.40 | | 4.97 | | | | | | | | | | | |
| *Congruus congruus* | | | | | | | | | | | | | | | | | | | | |
| SAM P33475 | 8.42 | 6.86 | 7.05 | | | 10.30 | 7.89 | 7.50 | 4.48 | 5.05 | 11.13 | 8.33 | 7.92 | 5.25 | 5.82 | 10.75 | | | | |

**Table 6.** Lower molar dimensions (mm) of *Congruus kitcheneri*.

| specimen | m1 | | | | | m2 | | | | | m3 | | | | | m4 | | | | |
|---|---|---|---|---|---|---|---|---|---|---|---|---|---|---|---|---|---|---|---|---|
| | L | AW | PW | AH | PH | L | AW | PW | AH | PH | L | AW | PW | AH | PH | L | AW | PW | AH | PH |
| WAM 66.8.17/66.9.47 | 7.16 | 5.12 | 5.45 | | | 9.71 | 6.18 | 6.52 | 5.87 | 7.17 | 11.80 | 7.29 | 7.15 | 7.33 | 8.25 | | | | | |
| WAM 66.9.72 | 8.10 | 5.17 | 5.27 | 4.68 | 6.08 | 9.92 | 6.49 | 6.17 | 6.82 | | | | | | | | | | | |
| WAM 02.7.12 | 7.44 | 6.00 | 5.95 | | | 9.32 | 7.19 | 7.51 | | | 10.70 | 8.01 | 7.98 | 5.37 | 6.98 | 12.10 | 8.15 | 7.50 | 6.15 | 7.43 |
| WAM 02.7.15 | 8.06 | 5.80 | 6.13 | 3.95 | 5.07 | 9.72 | 6.76 | 7.13 | 6.30 | 7.94 | 12.20 | 7.91 | 8.03 | 7.71 | 8.69 | 12.60 | 8.45 | 8.31 | 6.30 | 8.17 |
| WAM 02.7.21 | | | | | | 9.43 | 6.47 | 6.62 | 6.77 | 8.06 | 11.00 | 7.56 | 6.93 | 7.59 | 8.62 | | | | | |
| WAM 02.7.22 | 7.40 | 5.05 | 5.43 | 5.83 | 7.04 | 8.93 | 6.41 | 6.68 | 7.63 | 7.96 | | | | | | | | | | |
| WAM 02.7.26 | 8.15 | 5.80 | 6.40 | 3.85 | 4.56 | 10.10 | 7.20 | 7.49 | 4.89 | 5.65 | 11.20 | 8.27 | 8.59 | 6.67 | 7.49 | 11.90 | 9.01 | 8.30 | 7.20 | 6.78 |
| WAM 03.5.3 | | | 6.79 | | | 9.75 | 7.35 | 7.95 | | | 11.40 | 8.56 | 8.76 | 5.29 | 6.50 | 12.00 | 8.87 | 8.34 | 7.54 | 8.10 |
| WAM 19.7.202 | 7.53 | 5.60 | 6.30 | | | 9.59 | 6.86 | 7.39 | | | 11.10 | 8.51 | 8.74 | 5.58 | 6.55 | 12.40 | 8.74 | 7.97 | 7.07 | 6.21 |
| WAM 19.7.210 | 8.45 | 5.42 | 6.06 | 6.14 | 7.62 | | | | | | | | | | | | | | | |

**Table 7.** Dimensions (in mm) of the axial skeleton of *Congruus kitcheneri* compared with *Osphranter rufus*. Abbreviations: C, cervical; Ca, caudal; H, height; L, length; Lu, lumbar; S, sacral; T, thoracic; W, width.

| specimen dimension | *Congruus kitcheneri* WAM 02.7.12 | | | *Osphranter rufus* FUR 1 | | |
|---|---|---|---|---|---|---|
| | L | H | W | L | H | W |
| C1 | | 26.5 | 61.8 | | 66.8 | 30.2 |
| C2 | 42.6 | | | 47.7 | | |
| C3 | 19.7 | | | 20.3 | | |
| C4 | 18.5 | | | 17.6 | | |
| C5 | 17.4 | | | 17.8 | | |
| C6 | 16.9 | | | 15.9 | | |
| C7 | 18.7 | | | 16.2 | | |
| T1 | 17.8 | 11.2 | 17.3 | 19.2 | 14.1 | 22.2 |
| T2 | 17.7 | 10.2 | 11.0 | 21.7 | 11.4 | 15.5 |
| T3 | 18.0 | 10.8 | 11.4 | 20.4 | 13.8 | 14.4 |
| T4 | 17.7 | 11.6 | 12.6 | 21.2 | 13.9 | 16.0 |
| T5 | 17.1 | 12.9 | 11.9 | 22.0 | 15.9 | 15.0 |
| T6 | 16.6 | 13.5 | 12.6 | 21.2 | 15.7 | 14.5 |
| T7 | 16.4 | 13.6 | 12.7 | 21.4 | 15.0 | 15.5 |
| T8 | 17.0 | 15.6 | 15.1 | 21.9 | 15.7 | 14.7 |
| T9 | 16.8 | 15.0 | 13.4 | 22.2 | 15.5 | 16.8 |
| T10 | 16.4 | 15.1 | 15.6 | 22.0 | 16.4 | 15.7 |
| T11 | 16.8 | 16.3 | 18.2 | 18.5 | 19.5 | 16.6 |
| T12 | 16.1 | 17.8 | 18.6 | 17.3 | 20.6 | 21.6 |
| T13 | 15.7 | 17.6 | 20.6 | 16.0 | 21.1 | 21.1 |
| T14 | 14.8 | 21.7 | 24.4 | | | |
| Lu1 | 23.3 | 24.7 | 28.7 | 24.8 | 22.5 | 27.0 |
| Lu2 | 25.2 | 25.5 | 32.1 | 32.9 | 24.9 | 29.7 |
| Lu3 | 31.7 | 27.1 | 25.5 | 35.8 | 25.3 | 31.7 |
| Lu4 | 31.4 | 26.9 | 37.2 | 39.5 | 26.7 | 33.0 |
| Lu5 | 29.7 | 26.2 | 39.6 | 35.9 | 26.1 | 36.7 |
| Lu6 | | | | 33.2 | 26.4 | 39.8 |
| S1 | 27.1 | 25.8 | 40.9 | 33.3 | 22.4 | 38.8 |
| S2 | 27.0 | | | 30.3 | | |
| Ca1 | 23.6 | 24.6 | 26.3 | 30.3 | 25.0 | 29.9 |
| Ca2 | 19.4 | 25.8 | 27.8 | 29.8 | 26.2 | 28.0 |
| Ca3 | 19.3 | 25.0 | 25.7 | 33.5 | 25.7 | 26.5 |
| Ca4 | 21.6 | 23.7 | 24.3 | 42.4 | 26.4 | 26.5 |
| Ca5 | 28.3 | 22.6 | 24.3 | 55.3 | 27.8 | 27.0 |
| Ca6 | 40.1 | 23.1 | 24.9 | 64.8 | 27.3 | 29.8 |
| Ca7 | 47.3 | 23.6 | 27.7 | 67.9 | 28.0 | 34.5 |
| Ca8 | 51.9 | 24.2 | 29.8 | 67.7 | 28.1 | 33.8 |
| Ca9 | 51.0 | 23.8 | 28.6 | 67.5 | 28.5 | 34.0 |
| Ca10 | 49.6 | 23.7 | 28.1 | 62.5 | 26.5 | 36.1 |
| Ca11 | 49.0 | 21.8 | 27.3 | 59.1 | 25.8 | 33.6 |

(*Continued.*)

| specimen | Congruus kitcheneri WAM 02.7.12 | | | Osphranter rufus FUR 1 | | |
|---|---|---|---|---|---|---|
| dimension | L | H | W | L | H | W |
| Ca12 | 49.7 | 20.0 | 23.5 | 55.8 | 26.2 | 30.5 |
| Ca13 | 44.7 | 18.3 | 20.8 | 50.6 | 23.6 | 27.0 |
| Ca14 | 40.1 | 16.9 | 19.3 | 45.6 | 24.0 | 25.6 |
| Ca15 | 38.0 | 15.4 | 17.1 | | | |
| Ca16 | 35.3 | | | | | |
| Ca17 | 29.6 | | | | | |
| Ca18 | 25.1 | | | | | |
| Ca19 | 21.6 | | | | | |
| Ca20 | 20.4 | | | | | |
| Ca21 | 14.4 | | | | | |

A buccal crest on I3 that is equal in length to or longer than the main (lingual) crest is also an attribute of *Dendrolagus* and *T. thetis*, but the I3 of *Congruus* lacks the lingual shelf that typifies *Dendrolagus* and is much shorter than the I3 of *T. thetis*. The upper molars of *Congruus* most closely resemble those of *Thylogale* and *Protemnodon* in morphology and are intermediate in size between them. *Congruus* shares with *Thylogale* the same general crown outline, moderate crown height, thin lophs, a similarly proportioned precingulum and fine, low postproto- and postmetaconule-cristae. However, *Thylogale* differs by having a more distinct eminence or crest in the region of stylar cusp C (buccal to the postparacrista), especially on M1–2. The lophs and enamel are distinctly thinner in *Congruus* than *Protemnodon*. By comparison, the upper molars of *Dendrolagus*, *Bohra*, *Petrogale*, *Dorcopsis*, *Dorcopsulus* and *Setonix* are lower crowned and longer relative to their width, have thicker lophs and bear a better-developed postparacrista. All other genera within the Macropodini are higher crowned, have thicker lophs and enamel, have a stronger postprotocrista and/or lack a postparacrista.

*Congruus* has a more elongate, gracile dentary than any other macropodine, although *Pr. snewini* does bear some resemblance to it in this attribute. The *Congruus* dentary is distinguished by having a more anteroventrally oriented diastema (and thus i1), by being shallower beneath the anterior end of the cheek-tooth row and by having a more posteriorly declined anterior edge of the ascending ramus.

The procumbent, lanceolate i1 of *Congruus* is 20–30% larger than but very similar in morphology to that of *Wallabia*, both in unworn and worn states. *Congruus* may be distinguished by its more posteriorly extended ventral lobe of enamel on the buccal surface. The lower molars of *Congruus* are very similar in morphology to those of *Thylogale*, but more than twice as large. The lophids and lower molar enamel are distinctly thinner in *Congruus* than in *Protemnodon*. The p3 is shorter and wider than in *Thylogale* and *Protemnodon*. Otherwise, the lower cheek dentition of *Congruus* bears some general resemblance to that of *Kurrabi*, but the p3 is shorter, cheek-tooth row flatter, and lower molars have more a weakly developed paracristid and cristid obliqua.

The axial skeleton is distinct from those of other macropodines in having cranial facets that project well forward of the dorsal arch of C1, C2–C7 with very narrow centra mediolaterally, and 14 thoracic vertebrae and five lumbar vertebrae. The thoracic and lumbar centra are relatively small, with relatively short spinous and transverse processes, with no lumbar anapophyses.

The humerus is relatively long and straight with low proximal tuberosities, very strongly developed deltoid and pectoral crests, a projecting medial epicondyle and lateral extensor crest, and a large rounded capitulum. These features resemble those seen in *Bohra*, but are not as well developed. By contrast, *Protemnodon* has more strongly projecting proximal tuberosities, a deep bicipital groove and a more robust proximal humeral shaft craniocaudally. The radius and ulna are relatively more robust and with stronger muscle attachment scars than in *Macropus* and *Osphranter*, but with a more sinuous, shorter olecranon than in *Protemnodon.* The radial facets of the scapholunatum and hamatum are large and smoothly convex. The triquetrum is short and broad. The capitatum forms a hinge joint with the scapholunatum, and the hamatum has an inflected distodorsal margin. The metacarpals are more

**Table 8.** Dimensions (in mm) of the forelimb skeleton of *Congruus kitcheneri* compared with *Osphranter rufus*. Abbreviations: dist., distal; H, height; L, length; min., minimum; prox., proximal; W, width; *incomplete.

| specimen | *Congruus kitcheneri* WAM 02.7.12 | | | *Congruus kitcheneri* WAM 02.7.26 | | | *Osphranter rufus* FUR 1 | | |
|---|---|---|---|---|---|---|---|---|---|
| dimension | L | W | H | L | W | H | L | W | H |
| SCAPULA | | | | | | | | | |
| glenoid fossa | | 28.0 | 18.0 | | 32.5 | 22.7 | | 27.8 | 34.4 |
| neck (minimum) | | 29.2 | | | 34.5 | | | 40.2 | |
| vertebral border | 79.5 | | | 93.9* | | | 109.4 | | |
| CLAVICLE | | | | 79.8 | | | 100.9 | | |
| HUMERUS | 181* | 56.8* | | 210.0 | 66.5* | | 238 | 69.9 | |
| RADIUS | 197.4 | | | 216* | | | 317 | | |
| proximal epiphysis | | 18.7 | 16.1 | | | | | 21.6 | 17.6 |
| distal epiphysis | | 20.6 | 16.9 | | 23.5 | 18.8 | | 26.7 | 21.3 |
| ULNA | 228* | | | 245* | | | 365 | | |
| proximal diaphysis | | 17.5 | 9.8 | | 21.5 | 11.9 | | 25.1 | 14.5 |
| distal diaphysis | | 14.4 | 10.5 | | 16.8 | 12.9 | | 16.6 | 13.7 |
| CARPALS | | | | | | | | | |
| triquetrum | 8.4 | 13.3 | 10.4 | 15.9 | | | 15.8 | 20.1 | 14.6 |
| scapholunatum | 14.8 | 24.6 | 8.7 | 15.3 | 27.9 | 10 | 15.3 | 30.5 | 13 |
| hamatum | 15.3 | 21.7 | 20.9 | 18.7 | 27.3 | 21.8 | 19.6 | 23.5 | 20.5 |
| capitatum | 11.1 | 13.7 | 11.8 | 14.6 | 16.2 | 15 | 14.5 | 14.6 | 17.3 |
| trapezium | 12.6 | 10.6 | 6.1 | | | | 13.0 | 10.0 | 6.8 |
| trapezoid | | | | 7.2 | 8.3 | 10.6 | | | |
| pisiform | 16.9 | 8.9 | | | | | 26.4 | 17.4 | |

(*Continued.*)

**Table 8.** (*Continued.*)

| specimen | *Congruus kitcheneri* WAM 02.7.12 | | | *Congruus kitcheneri* WAM 02.7.26 | | | *Osphranter rufus* FUR 1 | | |
|---|---|---|---|---|---|---|---|---|---|
| dimension | L | W | H | L | W | H | L | W | H |
| METACARPALS | | dist. W | dist. H | | dist. W | dist. H | | dist. W | dist. H |
| metacarpal I | | | | 24.2 | 12.2 | 11.6 | 21.4 | 9.7 | 9.3 |
| metacarpal II | 32.4 | 11.2 | 10.1 | 34.6 | 13.2 | 11.9 | 30.1 | 12.3 | 10.9 |
| metacarpal III | 39.5 | 16 | 11.5 | 42.1 | 17.7 | 13.6 | 34.0 | 14.9 | 11.9 |
| metacarpal IV | 35.7 | 15.5 | 11.6 | 37.9 | 17.2 | 13.9 | 28.3 | 13.4 | 10.9 |
| metacarpal V | 25.2 | 10.9 | 9.5 | 28.6 | 14.1 | 11.4 | 22.5 | 10.6 | 10.1 |
| PHALANGES | | min. W | prox. H | | min. W | prox. H | | min. W | prox. H |
| proximal phalanx I | | | | 24* | 7.5 | | | | |
| proximal phalanx II | | | | 31.1 | 9.1 | | 23.3 | 8.7 | |
| proximal phalanx III | 34.3 | 12.0 | | 36.2 | 14.2 | | 25.6 | 10.6 | |
| proximal phalanx IV | 32.2 | 11.6 | | 37.5 | 13.9 | | 22.6 | 10.0 | |
| proximal phalanx V | | | | 28.4 | 10.3 | | 17.6 | 8.9 | |
| middle phalanx II | 21.1 | 8.0 | | 22.7 | 11.4 | | 13.7 | 10.4 | |
| middle phalanx III | 29.4 | 11.2 | | 30.6 | 14.1 | | 16.2 | 11.7 | |
| middle phalanx IV | 28.8 | 11.9 | | 31.5 | 14.1 | | 13.3 | 11.4 | |
| middle phalanx V | | | | 23.8 | 10.0 | | 11.8 | 9.9 | |
| distal phalanx I | 24.9 | | 13.0 | 25.4 | | 14.0 | 16.4 | | 8.2 |
| distal phalanx II | | | | 31.5 | | 17.2 | 21.3 | | 10.2 |
| distal phalanx III | 37.8 | | 19.5 | 39* | | 22.0 | 24.7 | | 10.4 |
| distal phalanx IV | 35.6 | | 17.0 | 48.7 | | 22.7 | 22.6 | | 9.7 |
| distal phalanx V | 27.1 | | 14.7 | 31.5 | | 16.3 | 17.7 | | 9.2 |

**Table 9.** Dimensions (in mm) of the hindlimb skeleton of *Congruus kitcheneri* compared with *Osphranter rufus*. Abbreviations: dist., distal; H, height; L, length; min., minimum; prox., proximal; W, width; *incomplete.

| specimen | Congruus kitcheneri WAM 02.7.12 | | | Congruus kitcheneri WAM 02.7.26 | | | Osphranter rufus FUR 1 | | |
|---|---|---|---|---|---|---|---|---|---|
| measures | L | W | H | L | W | H | L | W | H |
| INNOMINATE | | | | | | | | | |
| ilium | 150 | 47.0 | | | | | 183.3 | 51.6 | |
| ischium | 115 | | | | | | 140 | | |
| acetabulum | | 33.6 | 35.5 | | | | | 38.7 | 40.7 |
| EPIPUBIC | | | | 76.9 | | | 90.0 | | |
| FEMUR | 250 | 72.7 | | 255 | 77.1* | | 314.7 | 75.2 | |
| mid-shaft circumference | 90.0 | | | 102 | | | 75.2 | | |
| distal epiphysis | | 63.0 | 53.1 | | 67.7 | 59.1 | | 64.9 | 59.6 |
| TIBIA | 395 | | | | | | 600 | | |
| proximal epiphysis | | 57.5 | 59.5 | | 59.8 | 66.3 | | 63.9 | 72.8 |
| mid-shaft | | 23.9 | 22.6 | | 22.8 | 22.8 | | 26.2 | 22.8 |
| length of fibular facet | 220 | | | | | | 355 | | |
| FIBULA | | | | | | | | | |
| proximal tibial facet | | 23.6 | 12.9 | | | | | 29.7 | 9.9 |
| TARSALS | | | | | | | | | |
| calcaneum | 65.3 | 34.2 | 20.8 | | | | 95.9 | 35.5 | 27.2 |
| talus | 29.3 | 33.4 | | | | | 39.1 | 39.9 | |
| cuboid | 18.6 | 24.5 | 21.7 | | | | 21.9 | 27.5 | 32.9 |
| METATARSALS | | dist. W | | | | | | | |
| metatarsal IV | 94.1 | 22.7 | | | | | 177 | 26.4 | |
| metatarsal V | 76.7 | 14.2 | | | | | 158 | 9.3 | |

(*Continued.*)

**Table 9.** (*Continued.*)

| specimen | *Congruus kitcheneri* WAM 02.7.12 | | | *Congruus kitcheneri* WAM 02.7.26 | | | *Osphranter rufus* FUR 1 | | |
|---|---|---|---|---|---|---|---|---|---|
| measures | L | W | H | L | W | H | L | W | H |
| | | min. W | prox. H | | min. W | prox. H | | min. W | prox. H |
| PHALANGES | | | | | | | | | |
| proximal phalanx IV | 39.5 | 13.1 | | | | | 48.2 | 15.2 | |
| proximal phalanx V | 23.9 | 9.4 | | | | | | | |
| middle phalanx IV | 31.8 | 11.7 | | | | | 29.0 | 14.6 | |
| middle phalanx V | 17.8 | 9.4 | | | | | | | |
| distal phalanx IV | 35.0 | | 17.1 | | | | 30.0 | | 15.4 |
| distal phalanx V | 24.7 | | 12.6 | | | | | | |

robust than in *Macropus* and *Osphranter*. Metacarpal II is narrower than metacarpals III and IV and has a unique distal abaxial deflection. The proximal articular surface of metacarpal III forms an acute triangle shape, and the proximal articulations of the metacarpals I and V are strongly abducted and mobile. The middle phalanges are long relative to the metacarpals, and the distal phalanges are massive, long, laterally compressed and strongly recurved, being most like those of *Dendrolagus* but more robust.

The ilium is proportionately shorter, the iliac fossa is more deeply concave, the descending body of the ischium is narrower and less concave mesially, and the acetabulum is slightly deeper in *Congruus* than in *Macropus* and *Osphranter*. The deep circular acetabulum is distinct from those of smaller ground-dwelling macropodines, while the deep longitudinal groove along the caudolateral border of the ilium is unique. The femur is very robust with a medially displaced quadratus tubercle. The tibial plateau is transversely wide and asymmetrical in outline, with a distinctly larger medial condyle compared with the reduced lateral condyle. The calcaneus is robust and broad posteriorly compared with extant macropodines of similar size and is most similar in shape to that of *Thylogale*. The plantar tuberosity of the cuboid is very long and wide. Metatarsals IV and V are relatively short, compared with those of the length of the calcaneus. The middle and distal phalanges are long, and the ungual process strongly recurved. These aspects of the digital rays are most similar to those of *Bohra*, but are not as accentuated.

**Etymology:** From the Latin, meaning agreeable or congruent, in reference to the cranial similarities shared between the nominotypic species and other members of the crown macropodine clade (Macropodini).

*Congruus kitcheneri* (Flannery, 1989) [17]

**Holotype:** **WAM 66.8.17/WAM 66.9.47**, partial left and right juvenile dentaries (each preserving base of i1, dp2–3, m1–3, m4 in crypt). The buccal surface of the right p3 has been exposed in the crypt by fenestration of the lateral surface of the ramus (WAM 66.9.47), as was the left p3, which was extracted ([17]; figure 1*b–d*), but its whereabouts are now unknown (H. Ryan, May 2018, personal communication). The holotype was originally based solely on the right dentary (WAM 66.9.47), while the left dentary (WAM 66.8.17) was listed as a paratype. However, we can be confident that the two dentaries belong to the same individual; they express near-identical craniodental dimensions, morphology, dental wear and preservation. This attribution is reinforced by the known associations between left and right elements among the other paratypes of *C. kitcheneri* from Mammoth Cave, and the fact that element associations for individuals of other species are a feature of this assemblage (e.g. [32,33])

**Type Locality:** *Mammoth Cave, near Witchcliffe, southwestern Western Australia.* The holotype was found in the Mammoth Cave collection in the Western Australian Museum, the majority of which was collected by Ludwig Glauert, from the 'Glauert deposit', in the very early twentieth century [17,32,34]. Specimens retrieved from different stratigraphic layers were combined into one bulk, time-averaged sample, so the exact stratigraphic provenance within the now mined-out deposit and the geological age of the holotype are unknown. However, four uranium–thorium ages on calcite flowstones that sandwiched the Glauert deposit (of which remnants remain in the cave today), and three optically stimulated luminescence ages on quartz grains, bracket the age of the holotype and associated specimens to 75–44 ka [35,36]. A late Pleistocene age places the Mammoth Cave deposit within the Naracoortean land mammal age [37].

**Paratypes:** **WAM 66.8.18/66.9.29** left and right adult maxillae, preserving P3, M1. **WAM 66.9.71/WAM 66.9.72**, left and right juvenile dentaries (each preserving incomplete i1, dp2–3, m1–2, m3 in crypt, right p3 exposed in crypt by fenestration). **WAM 66.9.40/WAM 66.9.41**, partial left and right juvenile dentaries (preserving dp2–3, m1). **WAM 66.9.39**, partial right juvenile dentary (preserving dp3, m1, m2–3 in crypt, p3 exposed in crypt). **WAM 66.9.42**, partial left juvenile dentary (preserving dp3, partial m2 in crypt). These specimens are part of the same early twentieth-century collection from Mammoth Cave as the holotype.

**Referred material from the Thylacoleo Caves, Nullarbor Plain, Western Australia:**

*Leaena's Breath Cave.* **WAM 02.7.12**, adult skeleton. Cranium and dentaries near complete, missing right jugal, right paroccipital process, left I1, I3, right I1–3, left and right P3 crowns, fragments of left and right M1–3, right p3. Postcranial elements: complete vertebral column, including seven cervical vertebrae, 14 thoracic vertebrae, five lumbar vertebrae, sacrum, 22 caudal vertebrae and several chevron fragments; ribs; partial left and right scapulae; partial left and right humeri; left and right radii; right ulna, left ulnar distal epiphysis; left and right scapholunatums; partial right triquetrum; right pisiform; right trapezium; right capitatum; right hamatum; left metacarpal IV; right metacarpals II–III, V; left proximal, medial and distal phalanges of digits II–IV, right distal phalanges of manual digits III, V; left ilium with acetabulum, base of pubic ramus and partial descending body of ischium, fragments of left ischiatic table and ischiatic ramus; right ilium with acetabulum, root of ischium, and disarticulated epiphysis of the right ischiatic

tuber; left and right femora (proximal articular epiphyses missing); partial left and complete right tibiae; left and right fibular fragments; right calcaneus; right talus; right cuboid; right navicular; right ectocuneiform; left and right metatarsals IV–V; right proximal, medial and distal phalanges of pedal digits IV–V. Kitch Corner Site. Ages of most fossils scattered within the Leaena's Breath Cave rockpile are unknown, but are probably middle Pleistocene [13]. Naracoortean land mammal age. Collected by Paul Devine, Eve Taylor and Gavin Prideaux, July 2002.

**WAM 03.5.3**, adult cranium (preserving M1–4), missing incisors, left nasal and anterior portion of left frontal; partial right dentary (preserving m1–4); partial atlas; thoracic vertebra (T4?); distal portion of right humerus; medial condyle of right femur. Paul's Pit Site. Age as for WAM 02.7.12. Collected by Paul Devine, 16 May 2003.

**WAM 03.5.6**, partial juvenile skeleton. Partial cranium, consisting of premaxillae (missing I1–3), partial right maxilla (preserving M1–2, M4 unerupted), neurocranium. Postcranial elements: three cervical vertebrae, three thoracic vertebrae, two lumbar vertebrae, four caudal vertebrae; three partial ribs; partial left and right scapulae; left humerus; partial left ulna; partial left tibia; partial left fibula; right talus; left metatarsals IV–V. Gav's Wallaby Site. Age as for WAM 02.7.12. Collected by Paul Devine, Eve Taylor and Gavin Prideaux, July 2002.

**WAM 05.4.73**, juvenile left squamosal, M1, right i1. Rockpile, near LBC Thylacoleo #1. Age as for WAM 02.7.12. Collected by Lindsay Hatcher, 30 April 2004.

**WAM 19.7.201**, juvenile cranium (preserving partially erupted right I3, left and right dP2–3, partially erupted M2), missing left premaxilla, nasals, occiput, right squamosal and basicranium. Unit 1, Pit A, quadrat A4, depth 25–30 cm. Magnetic polarity and optically stimulated luminescence dating point to a middle Pleistocene age for Unit 1 [13]. Naracoortean land mammal age. Collected by James Moore, 11 May 2013.

**WAM 19.7.202**, right adult dentary (preserving base of i1, p3, m1–4). Unit 1, Pit C, quadrat C1, depth 10–15 cm. Age as for WAM 19.7.201. Collected by Paul Devine and Carey Burke, August 2011.

**WAM 19.7.203**, left metacarpal II. Unit 1, Pit B, quadrat B1, depth 0–5 cm. Age as for WAM 19.7.201. Collected by field party led by Gavin Prideaux, April 2013.

**WAM 19.7.204**, juvenile manual distal phalanx. Unit 1, Pit B, quadrat B7, depth 10–20 cm. Age as for WAM 19.7.201. Collected by field party led by Gavin Prideaux, April 2013.

**WAM 19.7.205**, juvenile distal phalanx of pedal digit V. Unit 1, Pit B, quadrat B7, depth 10–20 cm. Age as for WAM 19.7.201. Collected by field party led by Gavin Prideaux, April 2013.

**WAM 19.7.206a**, right juvenile humerus. Unit 1, Pit B, quadrat B8, depth 20–25 cm. Age as for WAM 19.7.201. Collected by field party led by Gavin Prideaux, April 2013. The same individual as WAM 19.7.206b.

**WAM 19.7.206b**, left juvenile humerus. Unit 1, Pit B, quadrat B4, depth 25–30 cm. Age as for WAM 19.7.201. Collected by field party led by Gavin Prideaux, April 2013. Same individual as WAM 19.7.206a.

**WAM 19.7.207**, juvenile distal phalanx of pedal digit IV. Unit 1, Pit B, quadrat B5, depth 30–35 cm. Age as for WAM 19.7.201. Collected by field party led by Gavin Prideaux, May 2013.

**WAM 19.7.208**, left adult femur. Unit 1, Pit B, quadrat B1, depth 30–35 cm. Age as for WAM 19.7.201. Collected by field party led by Gavin Prideaux, April 2013.

**WAM 19.7.209**, juvenile distal manual phalanx. Unit 1, Pit B, quadrat B1, depth 0–5 cm. Age as for WAM 19.7.201. Collected by Gavin Prideaux, April 2009.

**WAM 19.7.210**, left juvenile dentary (preserving i1, dp2–3, m1, unerupted m2–3). Unit 3, Pit B, quadrat B3, depth, 80–85 cm. The reversed magnetic polarity of sediments has been used to suggest an early Pleistocene age for Unit 3 [13]. Naracoortean land mammal age. Collected by field party led by Gavin Prideaux, April 2013.

**WAM 19.7.211**, partial left juvenile maxilla (preserving M1–2). Unit 3, Pit A, quadrat A4, depth 120–125 cm. Age as for WAM 19.7.210. Collected by field party led by Gavin Prideaux, April 2013.

**WAM 19.7.212**, proximal portion of right fibula. Unit 3, Pit B, quadrat B3, depth, 115–120 cm. Age as for WAM 19.7.210. Collected by field party led by Gavin Prideaux, April 2013.

**WAM 19.7.213**, juvenile manual distal phalanx. Unit 3, Pit B, quadrat B1, depth 90–95 cm. Age as for WAM 19.7.210. Collected by field party led by Gavin Prideaux, April 2013.

*Flightstar Cave.* **WAM 02.7.15**, partial young adult skeleton (P3, M4 partially erupted). Fragmented cranium, consisting of the entire dorsal surface as well as premaxillae (preserving left and right I1), partial maxillae (preserving left P3, M3–4, right M4), occiput; partial left dentary and near-complete right dentary (preserving left i1, m3–4, right i1, p3, m1–4). Postcranial elements: seven cervical vertebrae, one thoracic vertebra, seven lumbar vertebrae, one sacral vertebra, six caudal vertebrae; left and right scapula fragments; partial left and right humeri; left and partial right ulnae, several carpals,

metacarpals and manual phalanges; partial left and right innominates; partial left femur; partial left tibia, right tibia distal epiphysis; partial fibula; left talus; right cuboid; left and right metatarsals IV–V; several phalanges. The precise location from which this specimen was collected is recorded with the Department of Earth and Planetary Sciences, Western Australian Museum. Ages of most fossils scattered within the Flightstar Cave rockpile are unknown, but are probably middle Pleistocene [13]. Collected by field party led by John Long and Gavin Prideaux in July 2002.

**WAM 02.7.21**, partial juvenile skeleton (P3, M3–4 unerupted). including partial premaxillae (preserving partial left I1, right I1–3), left and right maxilla fragments (preserving left dP3, M1–2, right dP3, M1–3, unerupted P3), several neurocranial fragments; partial left and right dentaries (preserving left i1, dp2–3, m2, right i1, m2–3, unerupted p3 exposed in crypt). Postcranial elements: 11 thoracic vertebrae, four lumbar vertebrae, six caudal vertebrae; ribs; scapula fragment; partial left and complete right humeri; partial left and complete right ulnae, left and right radii; several carpals, metacarpals and manual phalanges; partial left and right innominates; partial left and complete right femora; partial left and complete right tibiae; partial fibula; left and right calcanei; left talus; right cuboid; left and right metatarsals IV–V; several phalanges. Flightstar Cave. Site, age and collection details as for WAM 02.7.15.

**WAM 02.7.22**, right juvenile premaxilla (missing I1–3) and right juvenile dentary (preserving dp3, m1–2, m3 unerupted). Postcranial elements: five cervical vertebrae, six thoracic vertebrae, seven lumbar vertebrae, four caudal vertebrae; ribs; left and partial right humeri; partial left and complete right ulnae; partial left and right radii; right ilium; left and right femora; left and partial right tibiae; left and right calcanei and tali; left and right metatarsals IV–V. Flightstar Cave. Site, age and collection details as for WAM 02.7.15.

**WAM 02.7.26**, partial adult skeleton. Fragmented cranium, including premaxillae (preserving partial left and right I1, I2), partial maxillae (preserving left and right P3, M1–4), several neurocranial fragments; left and right dentaries (preserving incomplete left and right i1, p3, m1–4, but missing ascending rami). Postcranial elements: seven cervical vertebrae (including partial atlas), 13 thoracic vertebrae (T4 missing), five lumbar vertebrae, sacrum, three caudal vertebrae (Ca2–Ca4); ribs; four sternebrae; partial left and right scapulae; left clavicle; near-complete left humerus and proximal portion of right humerus; fragments of left and right radii and ulnae; near-complete left and right manus, consisting of both scapholunatums, hamatums and capitatums, right trapezoid; left and right metacarpals I–V; left and right proximal, medial and distal phalanges of manual digits I–V; fragments of left ilium with acetabulum and proximal ischium; distal portion of right ilium; right epipubic; left and right femora; fragments of left and right tibiae. Flightstar Cave. Site, age and collection details as for WAM 02.7.15.

**Revised diagnosis:** Distinguished from all other macropodid species by having a pocket on the mesial surface of the premaxilla, and many tiny foramina across the premaxillary surface, and a premaxilla–maxilla suture positioned halfway along the diastema. Distinguished from *C. congruus* by having the following features: proportionally narrower rostrum and frontals; marked extension of the incisor-bearing portion of the premaxilla and thus a more elongate rostrum; more posteriorly emphasized dorsal doming of the nasals; deeper jugal; proportionally taller occiput; I3 buccal crest that is slightly longer and less incurved posteriorly toward the main (lingual) crest; P3 with a more convex buccal surface, shorter lingual cingulum and more distinct main crest cuspules and ridgelets; upper molars with more lingually tapered lophs, a less distinct preparacrista, a less incurved postparacrista, a weaker postprotocrista and no preprotocrista or urocrista on any molars. The most distinctive attributes of the dentary, lower dentition and postcranial skeleton of *C. kitcheneri* are currently listed under the generic diagnosis, because these elements are unknown for *C. congruus*.

**Etymology:** Named in honour of Darrell Kitchener, former curator of Mammalogy at the Western Australian Museum.

**Description and comparisons:**

**Cranium**. This description is based on both WAM 02.7.12 (figure 1) and WAM 03.5.3 (figure 2), which postcranial differences suggests are a female and male, respectively. The specimens express very similar levels of molar wear, suggesting that they are of a similar ontogenetic age. Only a few very subtle morphological differences are evident between the two near-identically sized crania.

The incisor-bearing portion of the premaxilla is distinctly elongate and peppered with tiny foramina less than or equal to 1 mm in diameter on the dorsal and lateral surface (figure 2*a*). This portion of the premaxilla is slightly deeper and broader in WAM 03.5.3 (figure 2). Together, the alveoli for I1–3 compose *ca* 40% of the length of the ventral edge of the premaxilla, from its anterior tip to the premaxilla–maxilla suture (figure 2*b*). The mesial premaxillary surface bears a distinct 'pocket' bounded internally by a thin extension of bone (figure 1*c*). The pocket opens posteriorly, adjacent to the anterior end of the maxilloturbinals, and extends anteriorly, narrowing and terminating at a

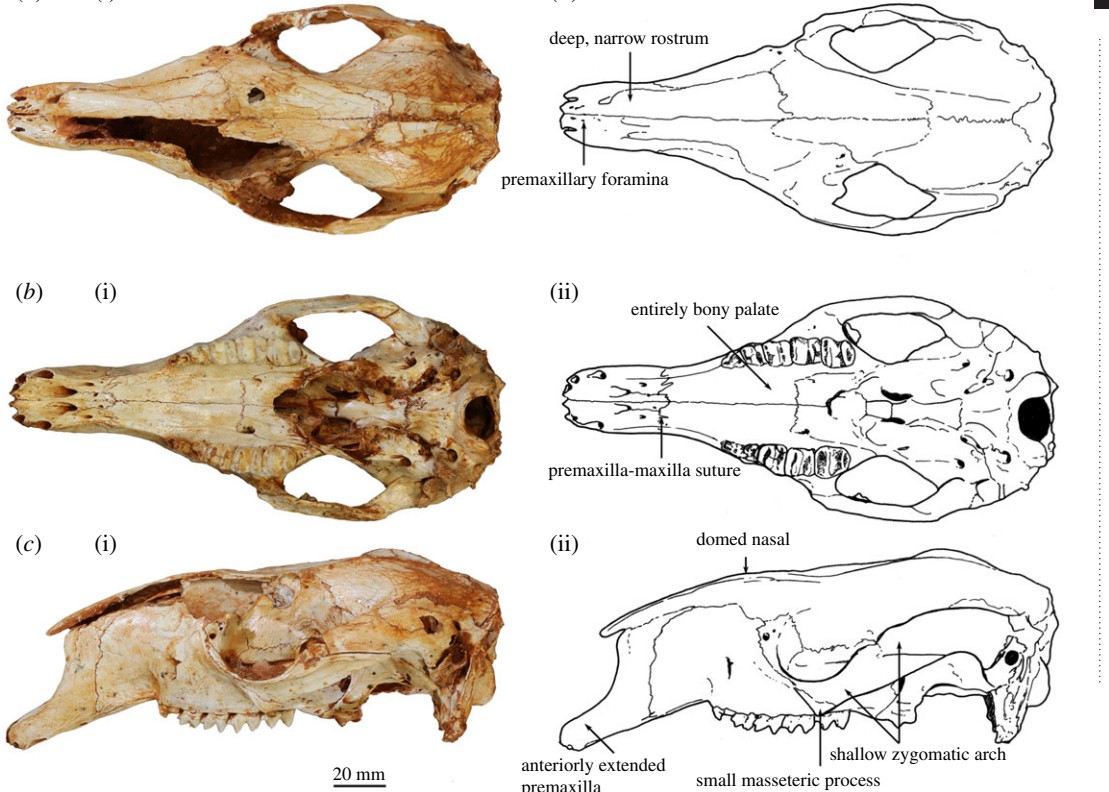

**Figure 2.** Adult cranium of *Congruus kitcheneri*. WAM 03.5.3 in (*a*) dorsal view, (*b*) palatal view and (*c*) left lateral view.

foraminous plate adjacent to the I3 alveolus. These foramina are linked via fine canals to those foramina dispersed across the anterior of the premaxilla. Viewed laterally, the premaxilla–maxilla suture ascends near-vertically, before inflecting posterodorsally and terminating midway along the nasal (figure 2*c*). Viewed ventrally, the intrapremaxillary sutures, which extend posteriorly from the incisive foramina, enclose a large extension onto the palate of the median (internal) portion of the premaxilla (figure 2*b*). A 1 mm diameter foramen is present on each side of this suture in WAM 03.5.3, but not in WAM 02.7.12. Both specimens have many finer foramina on the palatal aspect of the premaxilla. The incisive foramina extend anteriorly to a position adjacent to the I3 alveolus. The narial aperture is relatively straight-sided and narrow, with no distinct median ventral sulcus.

The long diastema extends for *ca* 60% of the rostrum length and is composed equally of premaxilla and maxilla (figure 2*c*). It is deflected anteroventrally at 20° to the alveolar margin of the cheek-tooth row. There is no C1 or vestigial C1 alveolus. The rostrum is deep, largely due to the posterior doming of the nasals to the same level as the neurocranium (figure 2*c*). The elongate nasals terminate above I3. As with the premaxilla, the most dorsal part of the maxilla is peppered with tiny foramina. The buccinator fossa is moderately deep, restricted to ventral third of the lateral rostral surface and extends from the anterior edge of P3 to the premaxilla–maxilla suture on the diastema border. The very short masseteric process is twisted laterally and composed entirely of maxilla (figure 2*c*). It terminates above the level of the alveolar margin and is positioned adjacent to the M4 protoloph. The large, narrow infraorbital foramen opens anteriorly, and is positioned directly above posterior root of P3 immediately anterior to the dorsal end of the anteorbital fossa. A subsidiary, 1 mm diameter foramen is located just anterodorsal of the infraorbital foramen. A small posterior (dorsal) lacrimal foramen opens dorsally and is separated from a similarly sized anterior (ventral) lacrimal foramen by small lacrimal tuberosity. A larger tuberosity, which is made up equally of lacrimal and frontal components, marks the dorsal extremity of the orbital rim. The nasolacrimal duct opens onto a broad, well-delineated concavity on the internal surface of the maxilla.

The frontals are narrow and bear a marked concavity centred on the cranial midline (figures 1*a,b* and 2*a,c*). The lateral margin of the frontal is sinusoidal in WAM 02.7.12, whereas it is much straighter in WAM 03.5.3, continuing the line of the temporal crest. Similarly, the postorbital constriction of WAM 02.7.12 is more marked due to the larger postorbital concavity on the frontal wall. The palatine bones

are solidly developed. The laterally oriented, anterior portion of maxillo–palatine suture lies adjacent to the abutment of M2–3. The palate is non-fenestrate, except that each palatine bears an anteriorly directed foramen on the anterolateral corner of the maxillo–palatine suture, adjacent to the M3. Posteriorly, the palate terminates adjacent to the posterior edge of M4. The temporal (parietal) crests are weakly developed. Posteriorly, they terminate at the lambdoid crest united and with no interparietal in WAM 02.7.12, but on either side of a large interparietal in WAM 03.5.3. The dome of the neurocranium, which is more marked in WAM 03.5.3 due to a slight deformation, attains its apex just posterior to the laterally oriented portion of the frontal–parietal suture. Anteriorly, the dorsal cranial surface descends into the mid-frontal concavity.

The zygomatic arch is slender and oriented posterodorsally at 30° to the alveolar margin of the cheek-tooth row (figures 1*a*,*b* and 2*a*,*c*). Viewed dorsally, the lateral edge is quite straight, but slightly more bowed adjacent to the posterior end of the temporal fossa in WAM 03.5.3. The posterior extremity of the jugal bears a small ectoglenoid process. The postorbital process of the jugal is not projected above the dorsal margin of the zygomatic process of the squamosal. The postglenoid process forms the posterior border of glenoid fossa. It is slightly more curved ventrally in WAM 02.7.12 and slightly thicker anteroposteriorly in WAM 03.5.3. The small, rugose, ectotympanic does not fully encircle the external auditory meatus, remaining open dorsally, and is only separated from the ovoid postzygomatic foramen by an extremely thin sliver of bone. The ectotympanic and postglenoid process are separated by a narrow gap. The large subsquamosal foramen is positioned adjacent to the posterodorsal extremity of the zygomatic process of the squamosal. This foramen is slightly smaller in WAM 03.5.3 than in WAM 02.7.12.

The basicranial plane lies on the same level as the palatal plane (figure 1*b*). The medial pterygoid origin is quite narrow and deep, and is flanked laterally by a groove extending anteriorly from the foramen ovale and by a well-developed anteroventral wing of alisphenoid. The posteroventral wing of the alisphenoid is enlarged and markedly extended posteriorly, uniting with the mastoid process of the periotic and exoccipital to form the paroccipital process. The median and posterior lacerate foramina are separated by a distinct anteromesially oriented flange of the alisphenoid. The basisphenoid is flexed anterodorsally at 20° relative to the basioccipital, which bears a low median keel. The basioccipital is slightly longer in WAM 03.5.3 than in WAM 02.7.12. As the lambdoid crest curves ventrolaterally it bifurcates into one crest that runs along the edge of the laterally flared mastoid process of the periotic and another ventrolaterally oriented crest formed along the lateral edge of the supraoccipital and exoccipital. The median occipital crest is low and rounded in WAM 02.7.12, but sharper in WAM 03.5.3. The ovoid foramen magnum is flanked by near-vertical occipital condyles, which are in turn bordered dorsally by deep sulci. The paroccipital process is relatively short, descending to the level of the cheek-tooth row. The mastoid process of the periotic terminates on the lateral edge of the paroccipital process level with the ventral extremity of the occipital condyle.

*Congruus kitcheneri* and *C. congruus* are more similar to each other in cranial size and morphology than to any other taxa, but they differ in several ways. Most obviously, *C. kitcheneri* is proportionally narrower across the rostrum and the frontals. The rostrum of *C. kitcheneri* is also more elongate, a difference largely due to the marked extension of the incisor-bearing portion of the premaxilla compared with that of *C. congruus*. The premaxillary pocket on the inside of the premaxilla is unique to *C. kitcheneri*, although in *C. congruus* there is a distinct sulcus in this region. The premaxilla of *C. kitcheneri* bears many more tiny foramina than that of *C. congruus*. The posterior doming of the nasals is more emphasized in *C. kitcheneri* than in *C. congruus*, its lacrimal tuberosities are smaller and less laterally projected, and its jugal is deeper along its length. The tympanic bulla of *C. kitcheneri* is slightly less inflated than in *C. congruus*, and its ectotympanic and external auditory meatus are proportionally smaller. The ectotympanic surface of *C. kitcheneri* is also less rugose. The occiput is proportionally taller in *C. kitcheneri* than in *C. congruus*.

**Upper dentition**. The arcuate I1 is high crowned, its anterior face either entirely smooth or bearing shallow, narrow grooves adjacent to its mesial and lateral borders (figure 1*e*–*g*). In unworn to slightly worn specimens, the occlusal surface is triangular, the crown is twice as wide here as it is anteroposteriorly deep. Oriented anteroventrally, I2 and I3 are much smaller than I1 and very similar to each other in size and morphology. Both incisors have distinctly small roots, particularly I3, due to the anterior extent of the premaxillary pocket. The occlusal surface of I2 is slightly wider posteriorly (and thus more triangular) than in I3. Toward the crown base, I2 is wider relative to its length, whereas I3 is longer relative to its width. The I3 crown is also deeper anteriorly than posteriorly. The I3 buccal crest, like the main (lingual) crest, runs the length of the occlusal surface. They are demarcated posteriorly by a short groove extending only a small way up the posterior edge of the

crown. In unworn specimens, the buccal crest is distinctly larger (i.e. more ventrally extended) than the main crest, a disparity that is particularly manifest in anterior or posterior view.

The I1 of *C. congruus* is unknown, but the I2–3 of both species are very similar. The buccal crest of I3 is slightly longer than the main crest in *C. kitcheneri*, and the two crests are more parallel, whereas the main crest curves toward the buccal crest posteriorly in *C. congruus*.

The cheek-tooth row is slightly curved (laterally convex) in occlusal view (figures 1*d* and 2*b*), although much of the curvature is due to the mesial offset of M4. In lateral profile, it is also slightly curved (ventrally convex). Both species are very similar in these attributes.

The P3 is an extended oval shape, which tapers to a point anteriorly. The crown is twice as long as it is wide posteriorly, and its anterior width is slightly less than its posterior width. The main crest is straight for its entire length or very slightly curved buccally at the posterior end in some specimens. It bears four cuspules with ascending ridgelets, which are better developed on the buccal side of the crown. In larger P3 specimens, one or two small accessory cuspules may be present midway along the buccal side at the crown base, but these are absent from smaller specimens. A distinct lingual cingulum is raised into a low crest along the length of the P3 and typically bears a pair of cuspules midway along. A very small posterior basin is enclosed by fine, low crests directed posteriorly and buccally from the posterolingual cusp that marks the posterior end of the lingual crest. The P3 is longer than M1, and intermediate in length between M2 and M3.

*Congruus kitcheneri* compares well with *C. congruus* in P3 relative size and morphology. However, *C. kitcheneri* differs from *C. congruus* in four distinct attributes. The P3 buccal surface in *C. kitcheneri* is convex buccally, not straight, in occlusal view. The lingual cingulum extends to the anterior end of the crown; it does not terminate adjacent to the anteriormost cuspule of the main crest. The cuspules and ridgelets of the main crest are very distinct, whereas they are poorly manifested in *C. congruus*. The posterior basin of *C. kitcheneri* is enclosed by a posterior crest, which is absent from *C. congruus*.

The upper molars are moderately high crowned. On M1 the protoloph is consistently narrower than the metaloph. On M2 they are more similar in width, but in some specimens, the protoloph is narrower and in others it is wider. The protoloph is consistently wider than the metaloph on M3–4. The loph crests are slightly concave posteriorly. Upper molar length relative to width increases from M1 to M4. Loph faces are smooth with no enamel crenulations. The precingulum extends the width of the molar, tapering toward its lingual extremity. It is connected to the paracone by a very weak preparacrista. The postprotocrista is fine, low and rapidly worn down. It peters out halfway down the middle of the metaloph anterior face. The postparacrista is finer than the postprotocrista and relatively straight (or very slightly curved in posteriorly). On the posterior molars of some unworn specimens, there is a slight hint of a premetacrista and postmetacrista, but otherwise these crests are absent. The fine, low postmetaconulecrista curves smoothly from the metaconule apex and terminates low on the posterobuccal corner of the crown.

*Congruus kitcheneri* differs from *C. congruus* in six upper molar attributes. The preparacrista is less distinct. The lophs are more tapered toward their crests, whereas both the lingual loph sides are more vertical and parallel to the buccal side in *C. congruus* (most obvious when viewed from the anterior or posterior). A small but distinct preprotocrista (forelink) is evident on the precingulum on M1–2 of *C. congruus*, but not in *C. kitcheneri*. The postparacrista in *C. kitcheneri* is typically straight, but in *C. congruus*, it is more distinctly curved in toward the tooth midline. The postprotocrista of *C. kitcheneri* is fine and low, but it is thicker and higher in *C. congruus*. There is a slight urocrista on M3 in *C. congruus*, but not in *C. kitcheneri*.

**Dentary**. The dentary is markedly gracile with an elongate diastema oriented at *ca* 10° to the cheek-tooth alveolar margin (figure 3). It is slightly deeper below p3 than m4. An ovoid anterior mental foramen is positioned immediately adjacent to and midway along the diastema, which curves down from the anterior root of p3 before straightening for the rest of its length. Mirroring the premaxilla, the lateral surface of the anterior portion of the dentary is peppered with tiny foramina. Reflecting the proportions of the dentary, the symphyseal plate is long and shallow, and only slightly rugose posteriorly in the region of cruciate ligament attachment. The symphysis terminates posteriorly *ca* 10 mm anterior to p3. The genial fossa is very small and shallow. Ramus depth below the abutment of m2 and m3 is 1.7–1.8 times its width. The digastric eminence is very slight and the digastric sulcus very shallow, and do not extend posteriorly beyond the anterior edge of the middle masseter muscle insertion, which results in the dentary being very thin through the transition from the horizontal to ascending ramus. The narrow, shallow buccinator sulcus extends from between the middle of the p3 roots to beneath the abutment of m2 and m3.

The anterior root of the ascending ramus lies well posterior to the m4. The anterior edge of the ascending ramus is oriented at 110° to the cheek-tooth alveolar margin. The inferior margin of the

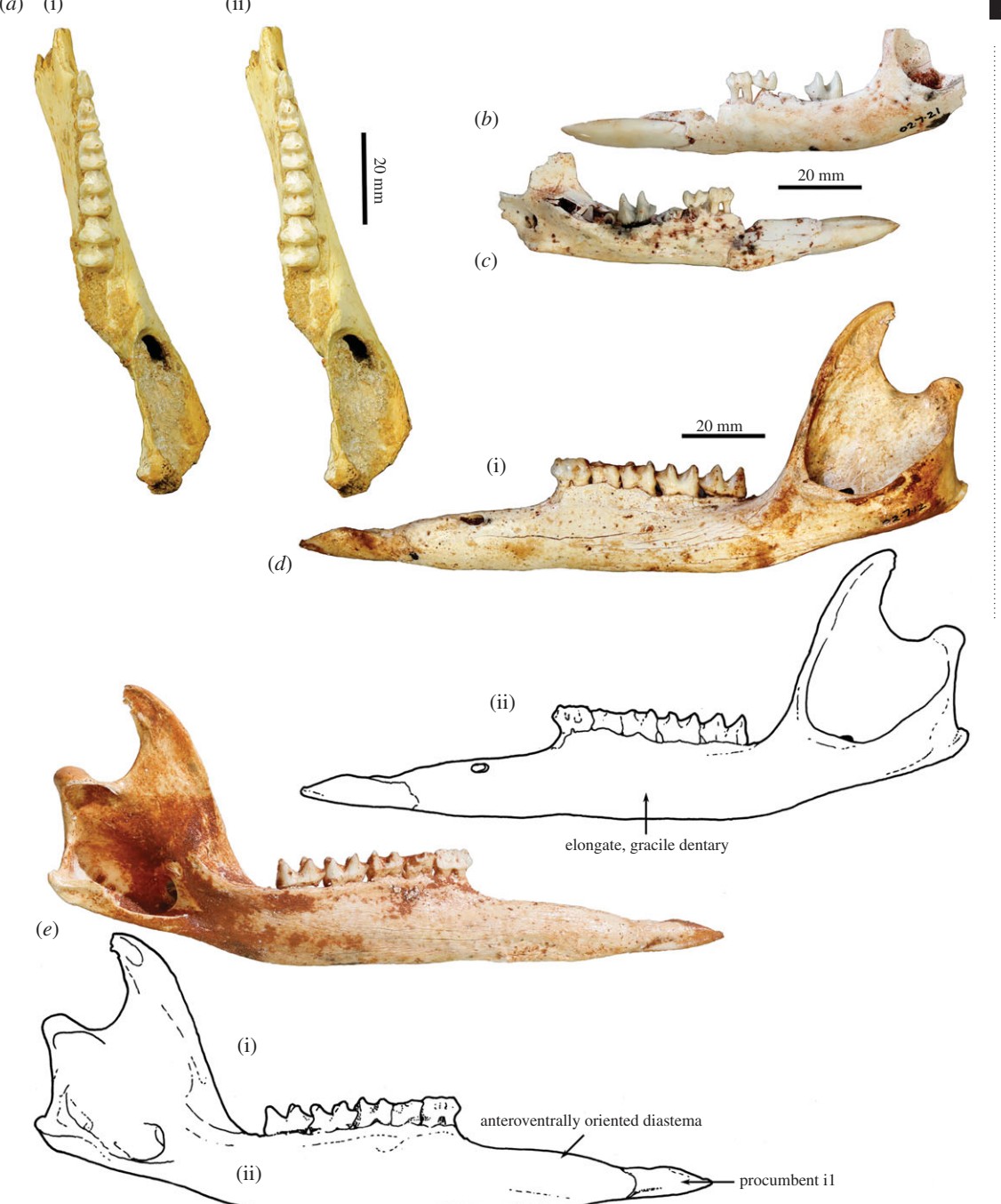

**Figure 3.** Dentary and lower dentition of *Congruus kitcheneri*. (*a*) Holotype right juvenile dentary (WAM 66.9.47) in occlusal view (stereo). Juvenile left dentary (WAM 02.7.21) in (*b*) lateral view and (*c*) mesial view. Left adult dentary (WAM 02.7.12) in (*d*) lateral view and (*e*) mesial view.

ascending ramus is inflected at 20° relative to that of the horizontal ramus. When viewed laterally, the posterior tip of the angular process (medial pterygoid fossa) extends posterior to the mandibular condyle and is on the same level as the crests of the unworn m3–4 crowns. The ventral border of the masseteric fossa is at a level just below the cheek-teeth alveolar margin, essentially level with the posterior end of buccinator sulcus. The anterior insertion area for the middle masseter is thin and restricted to the anteroventral rim of the masseteric fossa. The ovoid masseteric foramen opens into masseteric canal. Scarring suggests that the deep masseter terminated beneath the anterior root of the ascending ramus. Further deep masseter muscle scarring is evident within a sulcus immediately superior to the mesial edge of the masseteric foramen. The ovoid inferior mandibular foramen opens posteromesially. The mylohyoid groove is broad and shallow. The small mandibular condyle is ovoid

in dorsal view and slightly convex dorsally, with a distinct sulcus for the lateral pterygoid muscle on its mesio-inferior aspect.

**Lower dentition**. The i1 is procumbent and, when unworn, has a distinctly lanceolate shape (figure 3b–e). Its longitudinal axis essentially continues that of the body of the dentary. Although both are well developed, the ventral enamel flange is deeper than the dorsal flange, and their outer margins converge symmetrically to a point anteriorly. On the buccal surface, the ventral lobe of the enamel is more posteriorly extended than the dorsal lobe. In occlusal view, the anterior tip of i1 curves toward the cranial midline to meet the contralateral i1. Maximum extent of enamel on the buccal surface is *ca* four times its maximum depth. The occlusal facet is restricted to the anterior end of i1 and is primarily horizontal in minimally worn specimens. With increasing wear, the facet lengthens and becomes increasingly more declined relative to the longitudinal axis of the crown.

The cheek-tooth row is straight in occlusal view and in lateral profile, but it and the alveolar margin are inclined at *ca* 5° to ventral margin of the dentary, which results in a greater dentary depth beneath the anterior end compared with the posterior end of the cheek-tooth row.

The p3 is blade-like. Its main crest is oriented down the tooth midline and is straight for most of its length, before curving lingually at its posterior extremity. The main crest is composed of four cuspules, the middle two smaller and lower than the anteriormost and posteriormost cuspules. The distinctness of the middle two cuspules is rapidly obliterated with minor wear. The buccal and lingual sides of the crown converge relatively smoothly and symmetrically anteriorly. On average, p3 is *ca* 20% wider posteriorly than anteriorly, but this attribute is variable between specimens. Even more marked variation is observed in the relative length of p3, which is mostly, although sometimes only slightly, longer than m1, usually around the same length as m2, and consistently shorter than m3. Both the lingual and buccal bases of the crown fluctuate in the degree to which cingulids are expressed, varying from no discernible feature to a slight swelling to a distinct but smooth cingulid. When present, the lingual cingulid tends to be more distinct than its buccal equivalent.

The lower molars are moderately high crowned. When unworn, the thin lophid crests are distinctly convex posteriorly, a curvature that is strongly emphasized lingually on the hypolophid. Following only light wear, the curvature on both lophids is obliterated, and the crests manifest as straight, parallel and perpendicular to the tooth midline. The lophids are smooth and relatively thin anteroposteriorly, which results in them having quite steep anterior faces. The unworn cristid obliqua and paracristid are sharp and moderately distinct, terminating about halfway up the posterior faces of the protolophid and hypolophid of the anterior molar, respectively. However, these crests become rounded and low following only moderate wear. The cristid obliqua terminates at or just buccal to the tooth midline. At a comparable point at the anterior end of the tooth, the paracristid inflects lingually, before curving posteriorly and terminating at the base of the metaconid. On the anterobuccal corner of the crown, the low precingulid has a very similar but slightly less curved outline to the paracristid, which provides the trigonid/precingulid shelf with an outline that is just less than symmetrical. No premetacristid is present on any molars, whereas a very slight preentocristid, which is manifested as a ventrobuccally oriented eminence directed toward the centre of the interlophid valley, is evident on m1–2 when unworn. The posterior face of the hypolophid is smooth, flat and devoid of any noteworthy features.

**Atlas (C1)**. The dorsal arch is smooth, and moderately long craniocaudally, with a very small dorsal tubercle (figure 4a,c). The cranial articular facets are deep and dorsally tightly curved in WAM 02.7.12 (figure 4b), but are somewhat abraded in WAM 02.7.26 and were perhaps less tightly curved. Viewed dorsally (figure 4a), they project strongly from the cranial border of the dorsal arch. The ventral arch is unfused, but the broad neurapophyses, extending from the lateral bodies, converge towards the midline (figure 4b,c). The caudal articular facets are relatively large in surface area and are very shallowly concave. The bilateral wings are transversely broad, and in dorsal view slightly constricted at their base, before flaring distally to give a large surface area. The dorsolaterally positioned epineural canal (foramen alare) passes medially from above the root of the transverse process to enter the vertebral foramen caudal to the dorsal margin of the cranial articular fossa. A distinct groove marks the passage of the vertebral artery as it descends from the foramen cranial to the root of the transverse process. This is particularly strongly marked in WAM 02.7.26.

The cranial facets in *C. kitcheneri* project well forward of the dorsal arch compared with the condition observed in *M. fuliginosus*, *Os. rufus*, *T. billardierii* and *Pe. xanthopus*, in which the less projected articular facets produce a flatter cranial margin when viewed dorsally. In this feature, *C. kitcheneri* is most similar to *Dendrolagus*, but it is still more projected in the former. The absence of dorsal bilateral scars adjacent to dorsal tubercle distinguishes *C. kitcheneri* from *D. bennettianus*, *B. illuminata*, *B. nullarbora* and *Os. rufus*, in which there are marked semicircular depressions on either side of the small, anterior mid-dorsal crest.

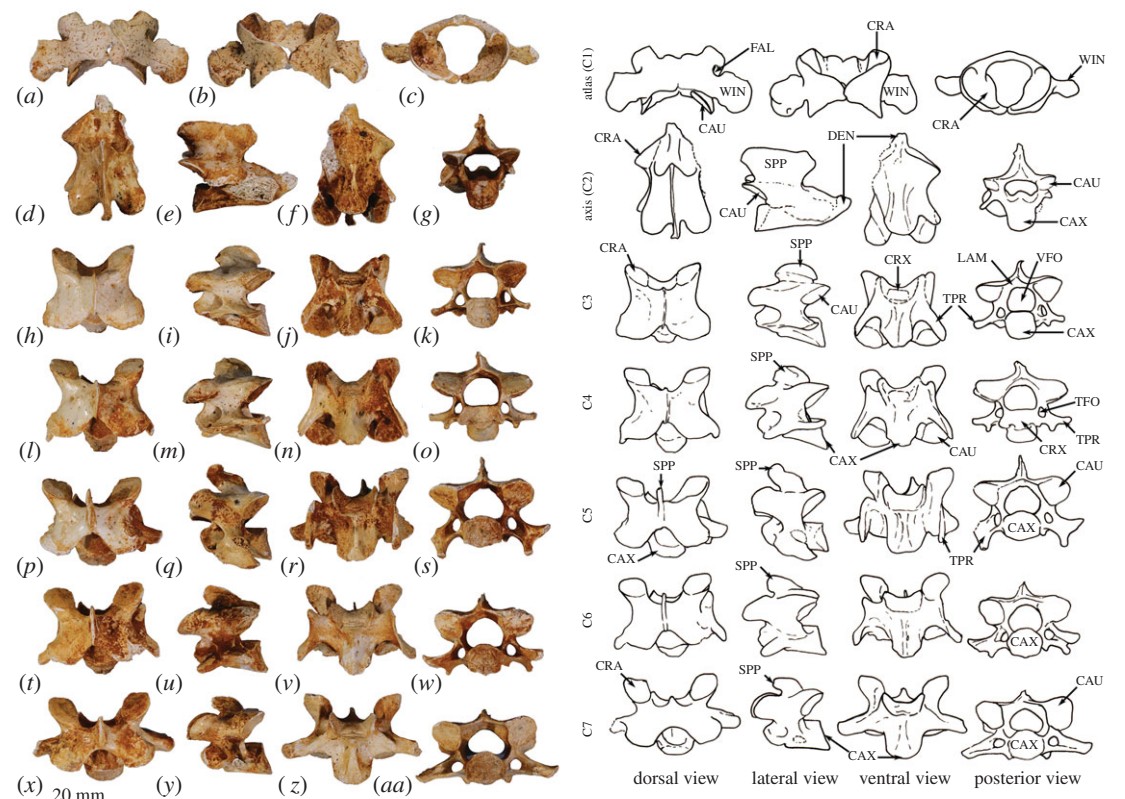

**Figure 4.** Cervical vertebrae of *Congruus*, WAM 02.7.12. (*a*) Atlas dorsal view; (*b*) atlas ventral view; (*c*) atlas cranial view; (*d*) axis dorsal view; (*e*) axis right lateral view; (*f*) axis ventral view; (*g*) axis caudal view; (*h*) third cervical vertebrae (C3) dorsal view; (*i*) C3 left lateral view; (*j*) C3 ventral view; (*k*) C3 caudal view; (*l*) fourth cervical vertebrae (C4) dorsal view; (*m*) C4 left lateral view; (*n*) C4 ventral view; (*o*) C4 cranial view; (*p*) fifth cervical vertebrae (C5) dorsal view; (*q*) C5 left lateral view; (*r*) C5 ventral view; (*s*) C5 caudal view; (*t*) sixth cervical vertebrae (C6) dorsal view; (*u*) C6 left lateral view; (*v*) C6 ventral view; (*w*) C6 caudal view; (*x*) seventh cervical vertebrae (C7) dorsal view; (*y*) C7 left lateral view; (*z*) C7 ventral view; (*aa*) C7 caudal view. Scale bar, 20 mm. Abbreviations: CAU, caudal articular surface; CAX, caudal extremity of centrum; CRA, cranial articular surface; CRX, cranial extremity; DEN, dens; FAL, foramen alare; LAM, lamina; SPP, spinous process; TFO, transverse foramen; TPR, transverse process; VFO, vertebral foramen; WIN, wing.

The cranial articular facets are more deeply curved dorsally than in *T. billardierii*, *D. bennettianus*, *M. fuliginosus* and *Os. rufus*, and are most similar to those of *Pe. xanthopus*. The broad, closed neurapophyses of *C. kitcheneri* appear most like those of *Pe. xanthopus*. A closed ventral arch of C1 is atypical of macropodines. It is occasionally observed in *M. fuliginosus*, though generally the bilateral neuropophyses remain separated mid-ventrally by a short gap. By contrast, the neurophyses of the atlas in sthenurines are very short and the ventral arch is open [22]. Other similarities in the form of the atlas between *C. kitcheneri* and *Pe. xanthopus* include a long (craniocaudally) dorsal arch, and tightly curved, cranially projecting articular facets for the occipital condyles.

**Axis (C2)**. The axis is robust and almost as deep dorsoventrally as it is long craniocaudally (figure 4*d–g*). The neural spine is trapezoidal from the lateral view (figure 4*d*) and does not extend beyond the margin of the caudal articular facets. The dens is short, robust and craniodorsally orientated. The caudal articular facets lie dorsal to the centrum (figure 4*g*). The cranial (atlantoaxial) articular facets are smoothly convex around an oblique craniolateral axis (figure 4*d*); the caudal articular facets are semicircular in shape. The centrum tapers caudally and extends caudoventrally towards a relatively small and square caudal extremity. The ventral surface of the centrum appears constricted in the caudal half, but has a broad ovoid tubercle in the cranial half (figure 4*f*). The dorsal roof of the vertebral canal (ventral surface of vertebral arch) is marked by a distinct mid-sagittal fossa cranially and by shallow, conjoined bilateral sulci caudally.

The axis of *C. kitcheneri* is distinct from that of all extant forms in having a relatively very narrow and square caudal extremity of the centrum with strongly developed ventral ridge. In all other taxa the caudal facet of the centrum is transversely wider than dorsoventrally deep. The strongly marked roof of the vertebral canal, with cranial fossa and caudal sulci is also very distinctive in comparison with extant

taxa. The spinous process is shorter in *C. kitcheneri* than in *D. bennettianus*, where it extends caudally well beyond the level of the articular facets.

**Cervical vertebra (C3–C7)**. C3–C5 are square in the transverse plane (figure 4*h–s*), while C6–C7 become transversely wider than dorsoventrally deep (figure 4*t–aa*). The vertebral canal is circular at C3, becomes wider ventrally through to C6 and is scalloped at C7 forming an inverse heart shape in section. From the dorsal view, C3–C6 are roughly square in shape and C7 is rectangular (transversely wider than craniocaudally long). The caudal articular processes (postzygapophyses) of C3 are transversely wider than the cranial articular processes (prezygapophyses), but for C4–C7 the cranial and caudal articular processes are of equivalent width. The spinous processes are abraded, but appear blunt in C3–C5 and more elongate in C6–C7. From the ventral view, the cervical vertebral bodies are narrow, relative to the shape of the neural arches from dorsal view; all centra have a pinched appearance of the ventral aspect, particularly C5–C7. The caudal extremity of each centrum extends caudoventrally and is small and circular in the transverse section. The transverse processes (ventral portion of complex diapophysis: pleurapophyses [38]) are short, robust and caudolaterally inflected, becoming longer and more strongly laterally directed by C7. The cranioventral branch of the transverse processes (anterior tubercle from the parapophyses [29]; tuberculum ventral) of C3–C6 appears small and blunt, and highly reduced in C7.

From the dorsal view, the square shape of C3–C7 in *C. kitcheneri* is more similar to those of *M. fuliginosus* and *Os. rufus* than *D. bennettianus*, *Pe. xanthopus* and *T. billardierii*, which are transversely wider by comparison. *Congruus kitcheneri* is distinct from *Os. rufus* and other extant macropodines in having relatively very narrow centra and small intervertebral joints. In other macropodines, the caudal extremity of the vertebral body is rectangular, being transversely much wider than dorsoventrally deep.

The neck length as a percentage of total presacral spine length (calculated as the sum of the cervical mid-ventral centra lengths divided by sum of all presacral mid-ventral centra lengths) is greater in *C. kitcheneri* (29–30%), than in *Os. rufus* (24.4%) and *D. bennettianus* (20%). Individually, the centra of the cervical vertebrae (C2–7) of *C. kitcheneri* is proportionately longer and narrower than *Os. rufus* and particularly *D. bennettianus* (figure 5).

**Thoracic vertebrae (T1–T14)**. There are 14 thoracic vertebrae with relatively short and narrow centra (figure 6*a–f*). The ventral surface of T1 has deep, clearly demarcated facets for the head of the first rib. The centra of T2–T11 are relatively small, with only a slight increase in size along with the series. T12–T14 become more lumbar in the form (transitional vertebrae) with increasingly large centra. Well-marked demifacets on each centrum reflect the articulation with the heads of the ribs: the fovea costales caudales are larger than the craniales and cause the lateral expansion of the dorsal portion of the caudal extremity, such that the centra have a heart-shaped outline in caudal view. The spinous process of T1 is broad in lateral view, and slightly cranially inclined relative to long axis of the centrum. In articulation, however, a lordosis at the transition between cervical and thoracic regions results in this spinous process being roughly vertical in orientation. The spinous process of T2 is longer than T1, but less robust and more perpendicular to craniocaudal axis of bone. The spinous processes T3–T10 are strongly inclined caudally and become shorter and more robust along with the series. T11 represents the anticlinal vertebra, T12–T14 are slightly cranially inclined. The thoracic transverse processes are robust and broad and have a facet for the articular part of the tubercle of the rib (diapophysis; fovea costalis processes transversi). In T1, the ventral aspect of the fovea lies just above the dorsal plane of the centrum and the fovea is deeply concave. In successive vertebrae, the transverse process becomes blunter and slightly more dorsally placed from the centrum (via the elongation of the pedicle of the neural arch) and the fovea becomes shallower. From T10 to T13, the tip of the transverse process becomes expanded above the fovea, particularly so in T13, and then in T14 this extension becomes completely separated to form a true anapophysis (after [29]), extending caudally from the dorsal base of the transverse process, above the caudal notch. The cranial articular processes (prezygapophysis) of T1 are broad and flat. From T2 to T11, the cranial articular processes are very short and have small ovoid, dorsally facing articular facets. In T12–14, the cranial articular facets of the prezygapophyses become increasingly enlarged, raised and medially inclined. Commensurate with this change, the mammillary processes (metapophyses; dorsal expansions above cranial articular facet) become increasingly lengthened and broadened. The caudal articular processes (zygapophyses) are similarly small to T10, and from T11 to T14 become increasingly enlarged, dorsally situated and laterally inflected.

*Congruus kitcheneri* is unusual in having 14 thoracic vertebrae, a condition shared with *Protemnodon*; all other macropodines have 13. While there is an addition thoracic vertebra, the total proportional length of the thoracic region in *C. kitcheneri* equates with those of *M. fuliginosus* and *Os. rufus*, which have 13

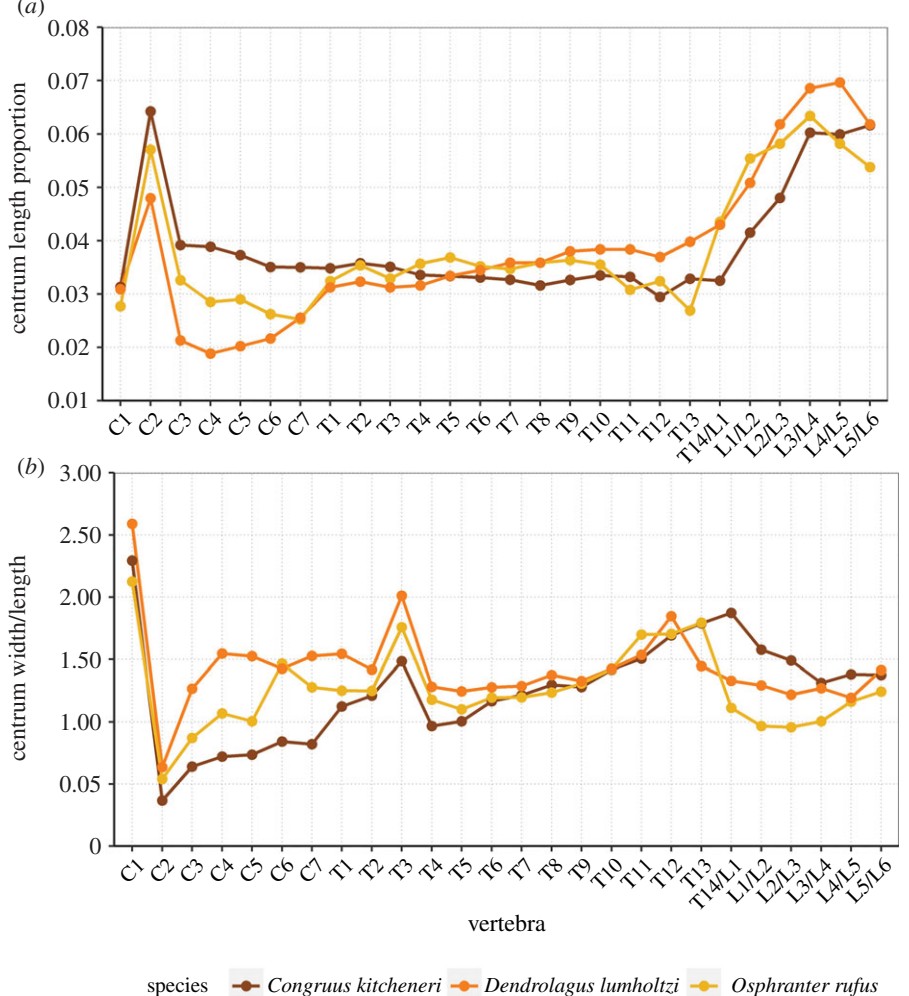

**Figure 5.** Presacral vertebral centrum dimensions. (*a*) Centrum length calculated as a proportion of total presacral centrum length and (*b*) centrum width divided by centrum length for individual vertebrae for *Congruus kitcheneri*, the tree-kangaroo *Dendrolagus lumholtzi* and the plains-dwelling kangaroo *Osphranter rufus*. From T14/L1, vertebrae number reflects different patterning in *C. kitcheneri* (14 thoracic and five lumbar vertebrae) and other taxa (13 thoracic and six lumbar vertebrae).

thoracic vertebrae. Morphologically, the thoracic spinous processes of *C. kitcheneri* are shorter, the thoracic centra smaller in transverse section, and intervertebral joints are relatively small and narrow in comparison with *M. fuliginosus* and *Os. rufus*. The relative length of individual thoracic centra does not appear to vary substantially between *C. kitcheneri*, *D. bennettianus* and *Os. rufus*, though the relative centrum width is less in T1–5 in *C. kitcheneri* (figure 5).

**Lumbar vertebrae (L1–L5).** The centrum of L1 is short and broad, and has a constricted ventral portion and sharp mid-ventral crest, which results in a heart-shaped transverse section (figure 6*g*,*h*). The centra of L2–L5 increase in length and become transversely wider than they are dorsoventrally deep (figure 6*i*–*o*). The centrum of L5 is ovoid in cranial view. The spinous processes are damaged, but that of L2 appears to have been long craniocaudally. The articular facets of both the pre- and postzygapophyses are relatively small, the mammillary processes (metapophyses) are moderately developed on L1–L2 and decrease in size from L3 to L5. The transverse process ('diapophysis' after [29]) of L1 is short and blunt, extending from the cranio-dorsolateral part of the centrum. The transverse processes of L2–L5 migrate dorsally to arise from the pedicle, and become longer and flatter in profile, as they extend laterally and then arc cranioventrally. Small anapophyses are present in the caudal notch in L1–L3 and are absent in L4–L5.

The lumbar centra are shorter, wider and less mesially tapered in *C. kitcheneri* than in *D. bennettianus*, *M. fuliginosus* and *Os. rufus* (figure 5). The spinous process of L2 is much wider in lateral view than in the latter species, in which L1 is noticeably broadest. The lumbar transverse processes are shorter in *C. kitcheneri* than in *M. fuliginosus* and *Os. rufus*. No distinct anapophyses are present above the caudal notch on the lumbar

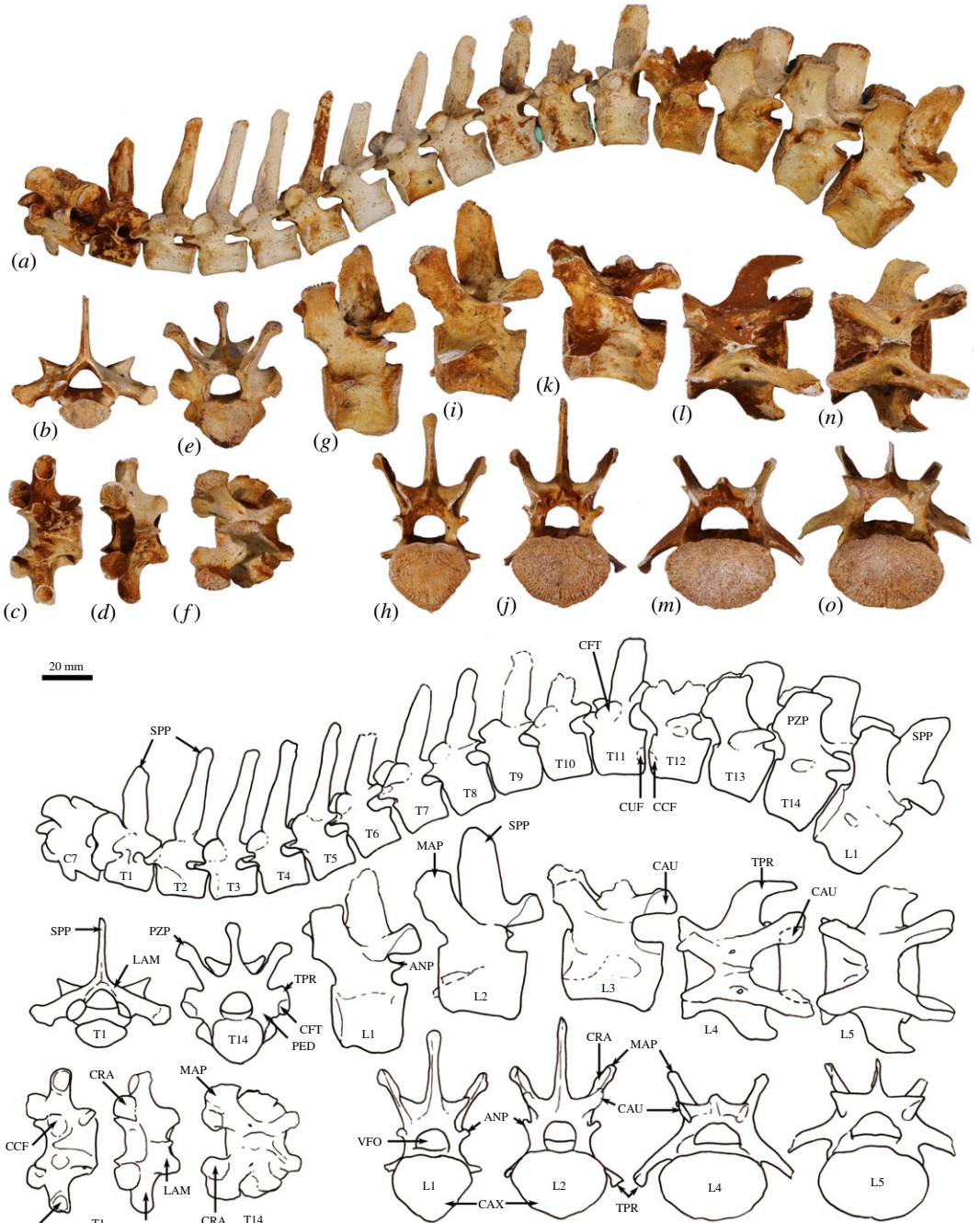

**Figure 6.** Thoracolumbar vertebrae of *Congruus kitcheneri*, WAM 02.7.12. (a) Semi-articulated vertebral series from C7 to L1 lateral view; (b) first thoracic vertebra (T1) cranial view; (c) T1 ventral view; (d) T1 dorsal view; (e) 14th thoracic vertebra (T14) cranial view; (f) T14 dorsal view; (g) first lumbar vertebra (L1) lateral view; (h) L1 caudal view; (i) second lumbar vertebra (L2) lateral view; (j) L2 caudal view; (k) third lumbar vertebra (L3) lateral view; (l) fourth lumbar vertebra (L4) lateral view; (m) L4 caudal view; (n) fifth lumbar vertebra (L5) lateral view; (o) L5 caudal view. Scale bar, 20 mm. Abbreviations: ANP, anapophysis; CAU, caudal articular surface; CAX, caudal extremity of centrum; CCF, cranial costal fovea; CFT, costal fovea of transverse process; CRA, cranial articular surface; CUF, caudal costal fovea; LAM, lamina; MAP, mammillary process; PED, pedicle of neural arch; PZP, prezygopophysis; SPP, spinous process; TPR, transverse process; VFO, vertebral foramen.

vertebrae of *C. kitcheneri*, in contrast with the relatively large anapophyses present on L1–L5 of *Os. rufus* and *D. bennettianus*. Small anapophyses characterize L1–L6 of *S. brachyurus*.

**Sacrum (S1–S2).** The sacrum is composed of two fused vertebrae. In cranial view, the S1 centrum is broadly ovoid and the vertebral foramen subtriangular (figure 7a). The prezygapophyses of S1 are relatively short and widely spaced, and the articular facets are relatively small. The ala formed by the

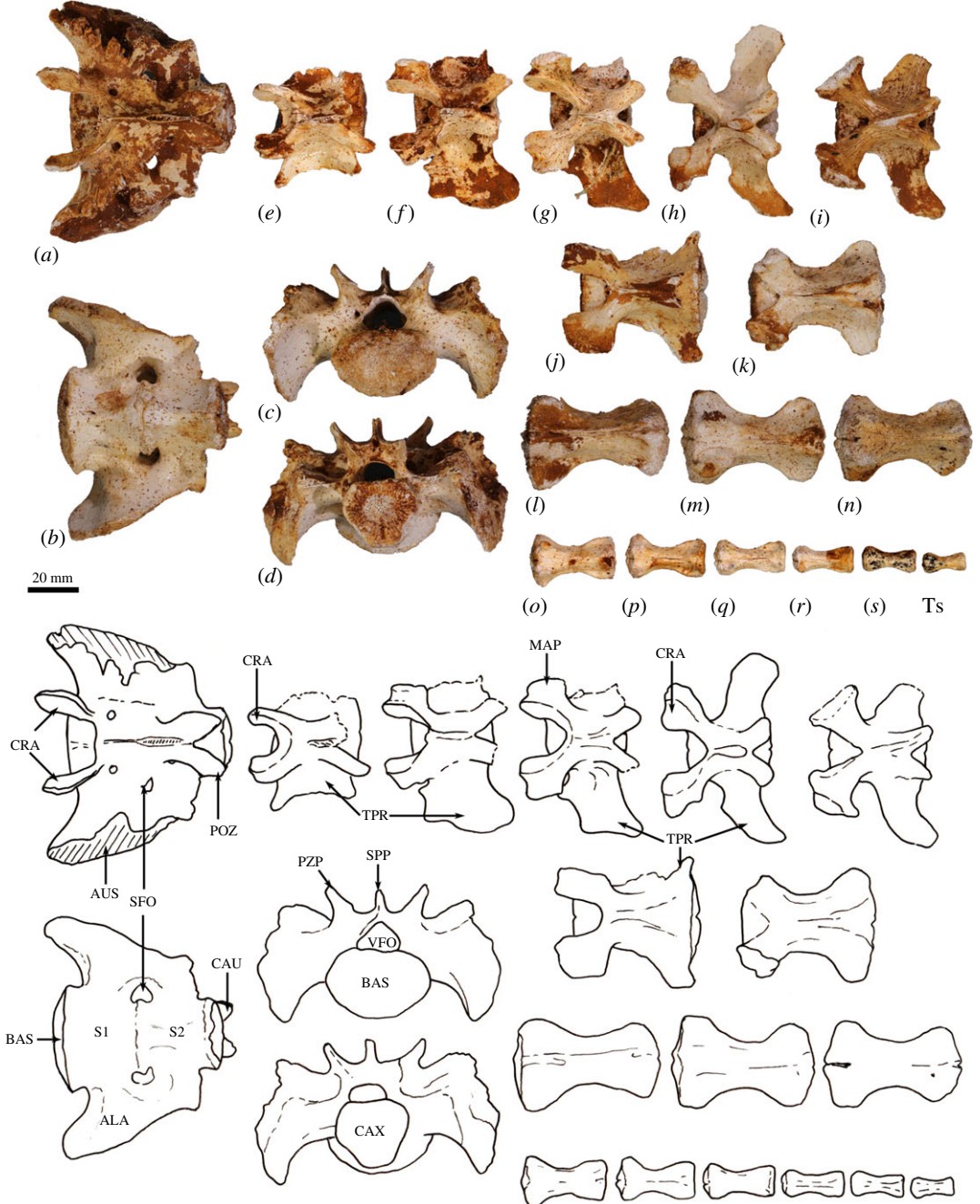

**Figure 7.** Sacrum and caudal vertebrae of *Congruus kitcheneri*, WAM 02.7.12. (*a*) Sacrum dorsal view; (*b*) sacrum ventral view; (*c*) sacrum cranial view; (*d*) sacrum caudal view; dorsal views of (*e*) first caudal vertebra (Ca1); (*f*) Ca2 dorsal view; (*g*) Ca3; (*h*) Ca4; (*I*) Ca5; (*J*) C6; (*k*) Ca7 (transitional vertebrae); (*l–n*) mid caudal vertebrae; (*o*) Ca17; (*p*) Ca18; (*q*) Ca19; (*r*) Ca20; (*s*) Ca21; (*t*) Ca22. Scale bar, 20 mm. Abbreviations: ALA, ala (wing) of sacrum; AUS, auricular surface of sacroiliac joint; BAS, base of sacrum; CAU, caudal articular surface; CAX, caudal extremity of centrum; CRA, cranial articular surface; MAP, mammillary process; POZ, postzygopophysis; PZP, prezygopophysis; S1, first sacral vertebra; S2, second sacral vertebra; SFO, sacral foramen; SPP, spinous process; TPR, transverse process; VFO, vertebral foramen.

fused transverse processes of S1 and S2 is dorsoventrally deep relative to the width of the sacrum (figure 6*c,d*). The auricular surface is a deep V-shape, and the sacral tuberosity (rough dorsal surface dorsal to facies auricularis) is large and deeply concave below a deeply rugose dorsal border. The spinous processes of S1 and S2 are fused. The caudal articular facets of the postzygapophyses are small.

In comparison with *C. kitcheneri*, the sacral base is more elliptical shallower than in *M. fuliginosus* and *Os. rufus*, and the vertebral foramen is wider and more ovoid. In *D. bennettianus, T. billardierii* and especially *Pe. xanthopus*, the sacrum is very broad and flat. The cranial articular facets of the

prezygapophyses of S1 in *M. fuliginosus* and *Os. rufus* have relatively much larger surface areas than those of *C. kitcheneri*. In *C. kitcheneri*, the continuous, fused sacral spinous processes contrast with the low and separate spinous processes in *M. fuliginosus* and *Os. rufus*. In *D. bennettianus* and *T. billardierii*, the sacral spinous processes are separate and uneven in height (high in S1 and very low in S2), while the sacral spinous processes in *Pe. xanthopus* have separate apices but are fused at their base.

**Caudal vertebrae (Ca1–Ca22)**. The first five (Ca1–Ca5; figure 7*e–i*) of the six proximal caudal vertebrae have short, deep centra, in which the mid-ventral length is shorter than the dorsal length, while Ca6 is more elongate (figure 6*j*). The second to fourth proximal centra are also wider (at their narrowest point) than they are long. Each has large transverse processes. The transverse processes of Ca1–Ca3 are longer (craniocaudal length) than their respective centrum length, and as transversely broad as they are long in Ca2–Ca3. In Ca1, the transverse process is narrow relative to craniocaudal length. In Ca4–Ca5, the transverse processes are transversely wider than they are long, and extend posteriorly towards their distal extremity, while in Ca6, the transverse process is relatively very reduced in size. The proximal caudal vertebrae have large prezygopophyses with small, ovoid articular facets and flared mammillary processes. The postzygapophyses extend beyond the caudal extremity of the centrum, and the neural spines are very low but distinct. The transitional caudal vertebra (Ca7; figure 7*k*) has smaller cranial and larger caudal transverse processes. The prezygapophyses of Ca7 are robust with no articular facets. The postzygapophyses are rudimentary, and the vertebral/neural canal is persistent but very small. The distal caudal vertebrae (Ca8–Ca22; figure 6*l–t*) have both cranial and caudal transverse processes, small prezygapophyses and small, paired cranial haemal spines, all of which diminish in size along with the series.

The high ratio of centrum width to length (robusticity) of the proximal caudal vertebrae of *C. kitcheneri* is more like those of *Petrogale* and particularly *Dendrolagus*, in comparison with ground-dwelling bipedal forms. Though broad, the transverse processes of the proximal series Ca1–Ca5 in *C. kitcheneri* are not as massive as those of *M. fuliginosus* and *Os. rufus*, in which they are transversely wider than craniocaudally long, and more posteriorly deflected. The haemal spines are much more widely spaced in *C. kitcheneri* than in *M. fuliginosus* and *Os. rufus*.

**Scapula**. The overall outline of the scapula is sub-trapezoidal (figure 8*a,b*). The supraspinous fossa is much smaller than (35% of the area of) the infraspinous fossa. The infraspinous fossa is quite long craniocaudally, with a curved vertebral border that results in fan shape. The scapular notch is moderately open, and the neck of scapula is very broad in dorsal view. The scapular spine is long and extends all the way to the cranial angle. The acromion is damaged but was apparently moderately long and broad, though not especially robust, and overlaps the cranial portion of the glenoid fossa in dorsal view. The elongate, ovoid glenoid fossa is particularly robust at the superior margin (figure 8*c*). In WAM 02.7.26 the glenoid fossa is more flared and rounded, and not quite so deeply concave from the ventral view in comparison with WAM 02.7.12. The supraglenoid tubercle (superior margin of glenoid) is flat and broad. The coracoid process is short, but distinct from the supraglenoid tubercle from which it is separated by a narrow groove and a small, deep oval fossa (nutrient foramen). In WAM 02.7.26, the coracoid process flares superiorly at its base to form a prominent bulb; the nutrient foramen between the supraglenoid tubercle and coracoid process is more ventrally placed than in WAM 02.7.12. The glenoid third of the caudal border is rugose and thickened, marking the origin of the long head of the mm. triceps brachii. Associated with this is a distinct, irregular fossa inferior to the margin of the glenoid, which is particularly strongly developed in WAM 02.7.26. The subscapular fossa is gently undulating and depressed towards the caudal border.

Overall, *C. kitcheneri* and *D. bennettianus* are similar in scapular morphology, sharing a relatively open scapular notch, an elongate, ovoid glenoid fossa with a robust supraglenoid margin, a relatively well-developed coracoid process, and a short, broad acromion process. By contrast, *M. fuliginosus* and *Os. rufus* have a more circular glenoid fossa, a blunter coracoid process and a much longer and more robust acromion process. *T. billardierii* has a similarly short, blunt acromion, but a more acute scapular notch than in *C. kitcheneri*, resulting in a more rectangular scapular shape overall in *T. billardierii*. In *S. brachyurus*, the acromion is similarly caudally displaced, but the scapular notch is much shallower and more open. The scapula of *C. kitcheneri* is similar to that of *B. illuminata* in having an elongate glenoid fossa and short, bulbous coracoid process, with a distinct intervening fossa, but distinct in that the scapular neck is much wider and the lengthened glenoid fossa is more robust cranially. The scapula of *Pr. brehus* has a similar overall shape to that of *C. kitcheneri*, dominated by a very broad triangular infraspinous fossa, and a very wide neck and glenoid fossa, but it is shorter (almost as broad as it is long) and has a more robust acromion.

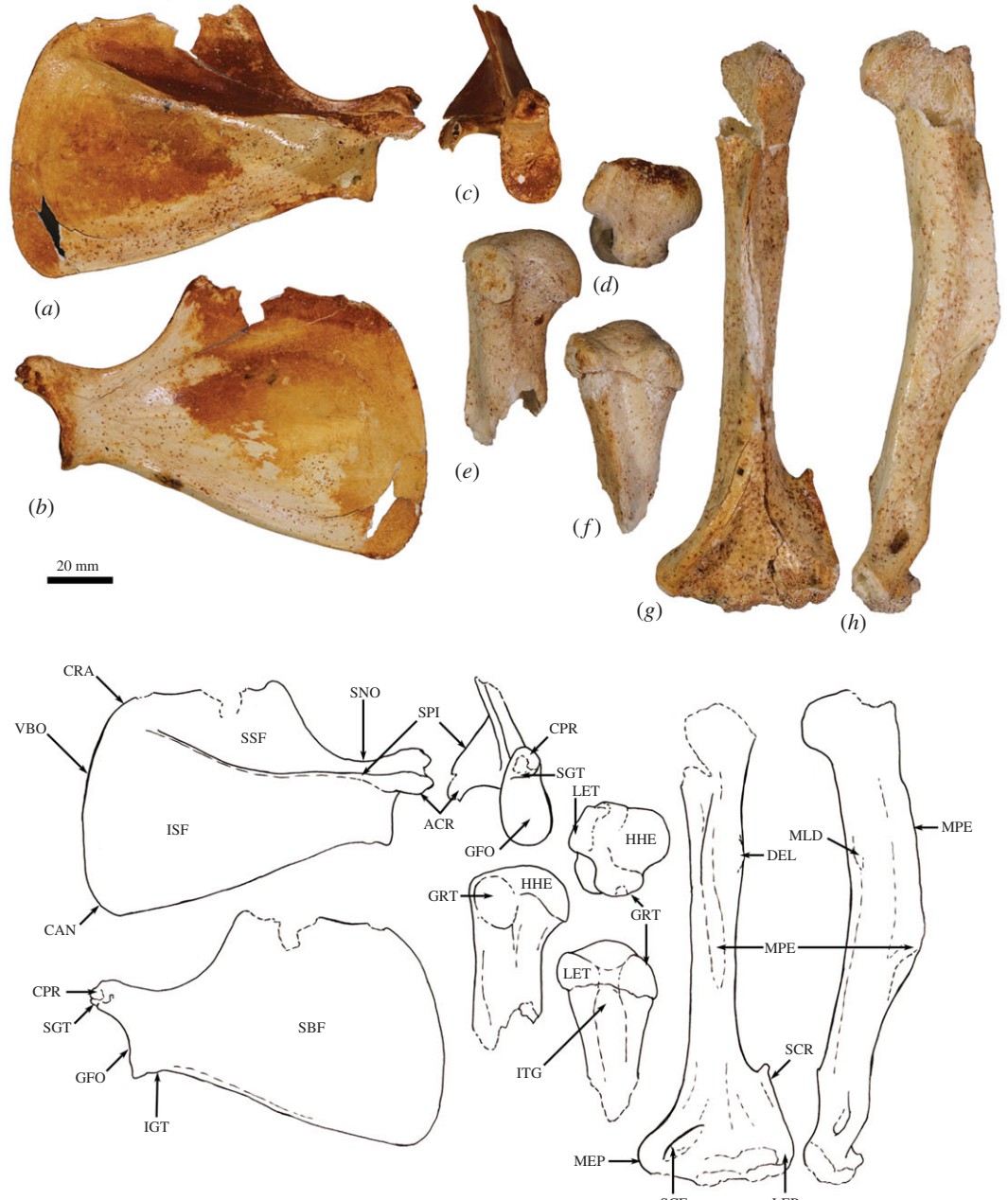

**Figure 8.** Forelimb elements of *Congruus kitcheneri*, WAM 02.7.12. (*a*) Right scapula in dorsal view; (*b*) scapula in ventral view; (*c*) scapula in proximal view; (*d*) right humerus in proximal view; (*e*) right humerus in lateral view; (*f*) right humerus in cranial view; (*g*) left humerus in cranial view; (*h*) left humerus in medial view. Scale bar equals 20 mm. Abbreviations: ACR, acromion; CAA, caudal angle; CPR, coracoid process; CRA, cranial angle; DEL, deltoid crest; GFO, glenoid fossa; GRT, greater tubercle; HHE, humeral head; IGT, infraglenoid tubercle; ISP, infraspinous fossa; ITG, intertubercular groove; LET, lesser tubercle; LEP, lateral epicondyle; MEP, medial epicondyle; MPE, insertion of m. pectoralis superficialis; SBF, subscapular fossa; SCF, supracondylar foramen; SCR, supracondylar ridge; SGT, supraglenoid tubercle; SNO, scapular notch; SPF, supraspinous fossa; SPI, scapular spine; TTU, teres tubercle; VBO, vertebral border.

**Clavicle**. The clavicle is robust and rounded in transverse section (figure 9*g,h*). The sternal end is expanded in superior–inferior axis. The body is almost straight and the acromial end curves gently in a posterolateral direction. The sternal end is globular and the facet for articulation with the manubrium of the sternum, though difficult to discern, appears to be relatively small and convex.

The clavicle of *C. kitcheneri* is much straighter and circular in transverse section compared with those of *M. fuliginosus*, *Os. rufus* and particularly *D. bennettianus*. The small sternal articulation is more like the condition observed in *D. bennettianus*, *Pe. xanthopus* and *T. billardierii* compared with the large, flattened articular surface with adjacent projecting tuberosity observed in large ground-dwelling species.

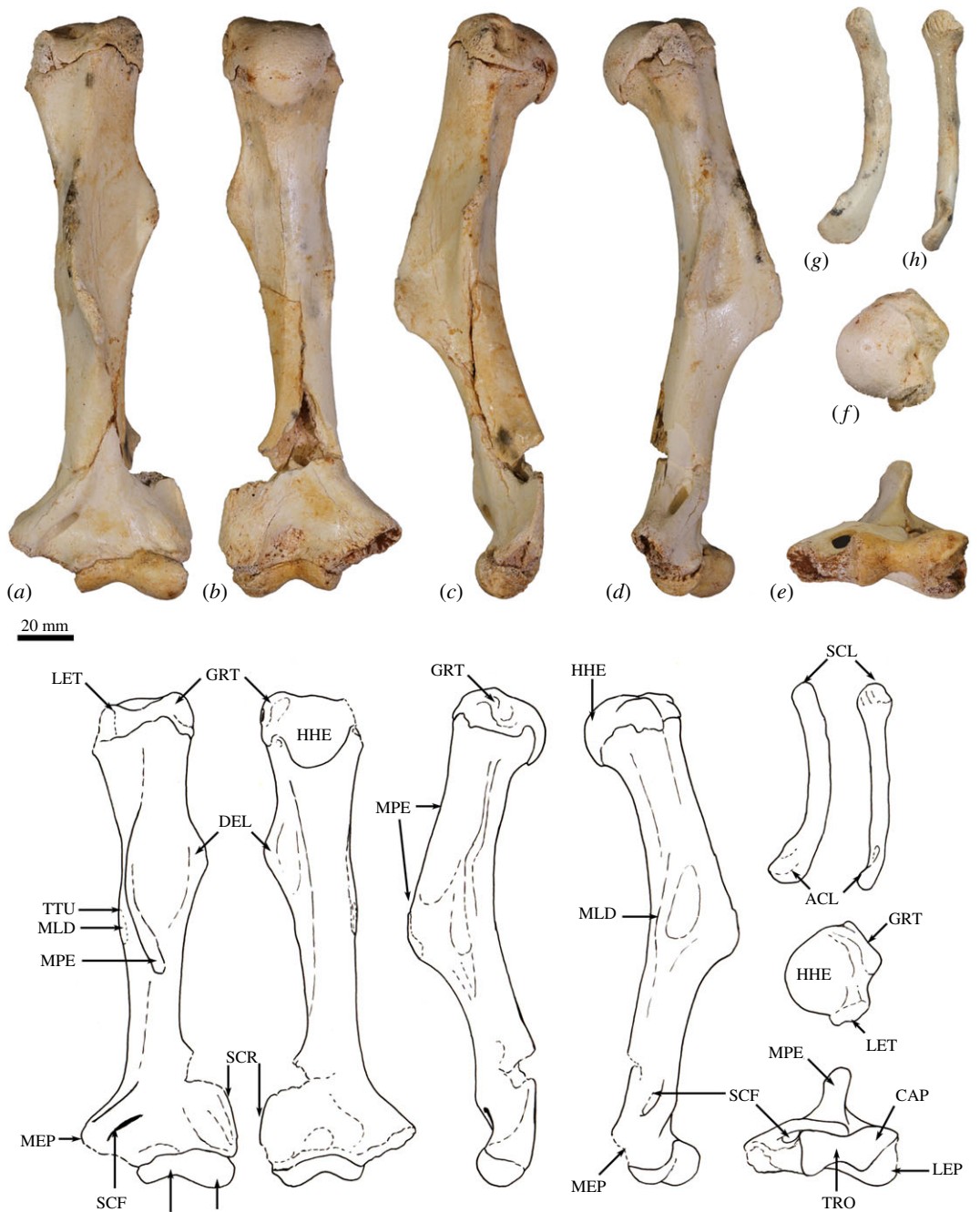

**Figure 9.** Forelimb elements of *Congruus kitcheneri*, WAM 02.7.26. (*a*) Left humerus in cranial view; (*b*) left humerus in caudal view; (*c*) left humerus in lateral view; (*d*) left humerus in medial view; (*e*) left humerus in distal view; (*f*) left humerus in proximal view; (*g*) right clavicle in dorsal view; (*h*) right clavicle in anterior view. Scale bar equals 20 mm. Abbreviations: ACL, acromial articular portion of clavicle; CAP, capitulum; DEL, deltoid crest; GFO, glenoid fossa; GRT, greater tubercle; HHE, humeral head; IGT, infraglenoid tubercle; ISP, infraspinous fossa; ITG, intertubercular groove; LET, lesser tubercle; LEP, lateral epicondyle; MEP, medial epicondyle; MLD, insertion of m. latissimus dorsi; MPE, insertion of m. pectoralis superficialis; SBF, subscapular fossa; SCF, supracondylar foramen; SCL, sternal articular portion of clavicle; SCR, supracondylar ridge; TRO, trochlear; TTU, teres tubercle.

**Humerus**. The humerus is long with strongly developed muscle attachments, particularly in WAM 02.7.26 (figure 9*a*–*f*). In cranial view, the shaft is quite straight, and the transverse width relatively similar throughout (figures 8*g* and 9*a*). In medial view, the shaft bows anteriorly (figures 8*h* and 9*d*). The articular surface of the head is hemispherical and smoothly rounded (figures 8*d*–*f* and 9*f*). The proximal tubercles are broad and do not extend above the articular surface of the head. The intertubercular (bicipital) groove is moderately deep and broad. The pectoral crest is long and high, forming a pronounced ridge that extends for roughly 60% of the diaphyseal length. In WAM 02.7.26,

the distal half of this crest is strongly projected anteriorly, forming a high, thick keel (figure 9*a,c*; MPE); the terminal portion curves slightly medially. In WAM 02.7.12, the deltoid crest forms a weak, oblique ridge one-third of the way along the lateral diaphysis. In WAM 02.7.26, the deltoid crest is much more strongly developed, forming an anteroposteriorly flattened projection situated approximately one-third of the length along the lateral diaphysis. The teres tubercle and insertion of m. latissimus dorsi are marked by a large irregular depression that is located roughly at the midpoint of the medial aspect of the shaft. In WAM 02.7.26, the depression is strongly marked and augmented by a rugose crest along the lateral margin (figure 9*d*; MLD). An additional muscle scar, presumably for part of m. triceps, is present on the proximal posterior aspect of the shaft. The distal extremity is broad with a large, robust medial epicondyle (figures 8*g* and 9*a*). The lateral epicondylar ridge is relatively small. The olecranon fossa is relatively shallow. The posterior face of the diaphysis is gently concave, forming a broad, subtriangular fossa. The anconeal fossa is very shallow, while the radial fossa is slightly deeper. The articular surface of the trochlea is strongly constricted mesially, and the capitulum is large, broad and rounded.

The relatively massive development of the muscle attachment sites on the humeral diaphysis of *C. kitcheneri* is distinctive, in particular the height of the pectoral crest. The humerus of WAM 02.7.12 is relatively long, moderately robust, but with more distinct muscle attachments when compared with *M. fuliginosus*, *Os. rufus*, *T. billardierii*, *S. brachyurus*, *D. bennettianus* and *B. illuminata*. Shared features with *D. bennettianus* and previously described species of *Bohra* (and distinct from *Macropus*, *T. billardierii* and *S. brachyurus*) include the relatively low proximal tuberosities, robust distal diaphysis, broad, shallow intertubercular groove, long pectoral crest, relatively proximally placed deltoid insertion, very pronounced teres–latissimus dorsi sulcus, distinct postero-proximal origin of m. triceps brachii and large medial epicondyle. The humeri of *Protemnodon anak* and *Pr. brehus* are more robust overall, especially in the craniocaudal axis of the proximal half of the shaft, and bear a deeper, more strongly developed bicipital groove passing between the more projecting proximal tubercles. In these species of *Protemnodon*, the medial epicondyle is less projecting than in *C. kitcheneri*, the trochlea is broader and the capitulum is very large and rounded. The massive development of the muscular crests in WAM 02.7.26 relative to WAM 02.7.12 is consistent with sexual dimorphism that has been reported in the forelimb of extant kangaroos [39]. The deltoid tubercle of *C. kitcheneri* is more medially projecting and more proximally placed than in large male *M. fuliginosus* and *Os. rufus*.

**Radius**. The radius (figure 10*a–e*) is slightly longer than the humerus and is robust along its length relative to the ulna. The radial head is ovoid (figure 10*b*), and the radial neck is thicker in the anteroposterior plane than the mediolateral plane. The radial tuberosity is large in surface area, moderate in height, and ovoid, with its long axis parallel to the long axis of the diaphysis. In the anteroposterior plane, the shaft is narrowest immediately distal to the radial tuberosity and then expands gradually along its length toward the distal epiphysis. The interosseous border forms a well-defined ridge that extends along most of the length of the diaphysis. The palmar surface of the distal shaft is broad, and flat to slightly concave, while the dorsal surface is convex. The anteromedial aspect of the middle third of the diaphysis is marked by a distinct elongate sulcus, bounded by a distinct ridge laterally (most likely for the m. pronator teres). This is particularly obvious in WAM 02.7.26. The distal epiphysis is wider in the mediolateral axis than dorsoventrally. The distolateral surface (insertion of the m. pronator quadratus) is flattened with distinctly raised borders. The concave ulnar notch is elongate and bounded by distinct ridges on either side. The articular surface for the scapholunatum is shallowly concave and extends smoothly onto the mesial aspect of the robust styloid process.

The distal expansion and flattening of the radial shaft in the dorsoventral plane in *C. kitcheneri* is most similar to *S. brachyurus* and *T. billardierii*, being intermediate between the very flattened and broad morphology of *D. bennettianus*, and the narrow, rounded distal diaphysis of *M. fuliginosus* and *Os. rufus*. The radius of *C. kitcheneri* is similar to those of *D. bennettianus* and *B. nullarbora*. It is distinguishable from those of *M. fuliginosus*, *Os. rufus*, *Pe. xanthopus*, *S. brachyurus* and *T. billardierii* in having a relatively long radial neck, larger surface area of the radial tuberosity, and more clearly demarcated insertion of the m. pronator quadratus on the mesial distal surface. The distal epiphysis of the radius of *C. kitcheneri* is similar in morphology to that of *Os. rufus*.

**Ulna**. The ulnar shaft is gently curved and mediolaterally compressed, with a sharply ridged interosseous border (figure 10*f,h,i*). The proximal half is flexed ventrally; the distal half is slightly medially flexed. The olecranon is moderately long, as judged from the diaphysis (proximal epiphysis missing from WAM 02.7.12 and WAM 02.7.26; figure 10*g,j*). The trochlear notch is deep, with a relatively narrow posterior margin and a pronounced anconeal process. The anteromedial portion of

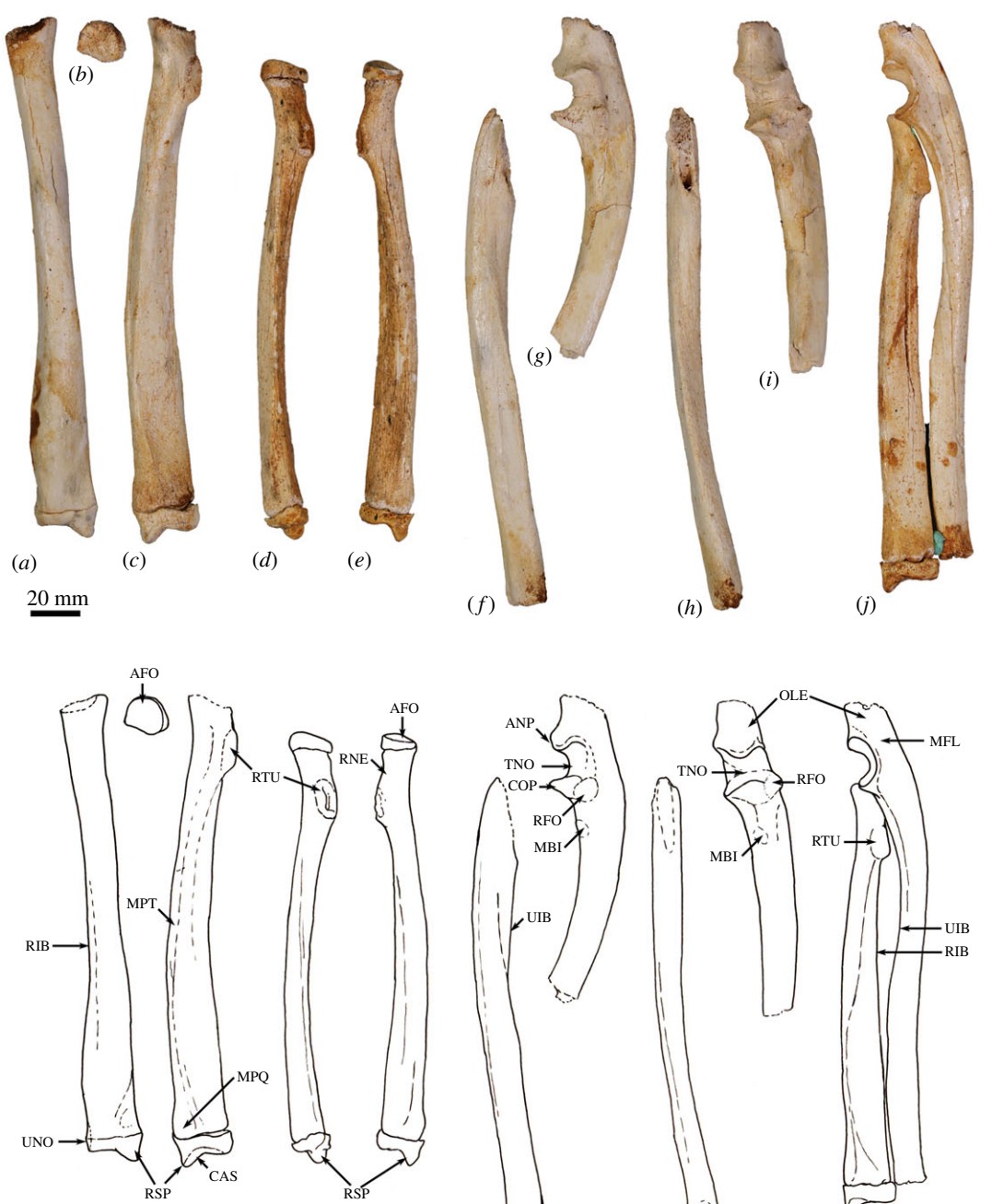

**Figure 10.** Radius and ulna of *Congruus kitcheneri*, WAM 02.7.12 and WAM 02.7.26. (*a*) Right radius 02.7.26 dorsal view; (*b*) right radius 02.7.26 proximal view; (*c*) right radius 02.7.26 palmar view; (*d*) left radius 02.7.12 dorsal view; (*e*) left radius 02.7.12 palmar view; (*f*) right ulna 02.7.26 distal shaft lateral view; (*g*) left ulna 02.7.26 proximal shaft lateral view; (*h*) right ulna 02.7.26 distal shaft cranial view; (*i*) right ulna 02.7.26 proximal shaft cranial view; (*j*) right radius and ulna 02.7.12 in articulation medial view. Scale bar equals 20 mm. Abbreviations: AFO, articular fovea of radial head; ANP, anconeal process; CAS, carpal articular surface; COP, coronoid process; MBI, insertion of m. biceps brachii; MFL, origin of flexor musculature; MPT, insertion of m. pronator teres; MPQ, insertion of m. pronator quadratus; OLE, olecranon; RFO, radial fossa; RIB, radial interosseous border; RNE, radial neck; RSP, radial styloid process; RTU, radial tuberosity (m. insertion of biceps brachii); TNO, trochlear notch; UIB, ulnar interosseous border; UNO, ulnar notch.

the trochlear notch is wider and more rounded than the posterior margin. The proximal radial fossa is semicircular in shape, and almost as long as it is wide. The proximocranial border is thickened with a slight rugosity marking the brachial/bicipital tubercle. The lateral aspect of the distal half of the diaphysis is flat, in contrast with the convexly rounded medial surface. The articular surface of the ulnar styloid process is small and almost spherical on a short, narrow neck.

The ulna of *C. kitcheneri* is most similar to *D. bennettianus* and species of *Bohra*. They share a diaphysis that is laterally compressed and moderately robust along its length, a shorter caudal border of the trochlear notch (more so in WAM 02.7.12 than in WAM 02.7.26), and a larger surface area of the proximal radial facet, which has a more lateral, rather than cranial, orientation. These features are more pronounced in the tree-kangaroos than in *C. kitcheneri*. In comparison, the ulnar diaphyses of *M. fuliginosus*, *Os. rufus*, *Pe. xanthopus*, *S. brachyurus* and *T. billardierii* are rounded in section and tapered distally, the posterior borders of the trochlear notch are wider and more pronounced, and the proximal radial facets are wider than long. The ulna of *Pr. brehus* is straighter overall and has a much more elongate olecranon than *C. kitcheneri*.

**Scapholunatum**. The radial facet is transversely broad, dorsoventrally compressed and smoothly convex (figures 11*a,b*, 12*a–d* and 13*a–d*). The facet for the hamatum is concave and moderately large. When articulated, the scapholunatum and hamatum form a high, smooth dome for articulation with the radius. The distal articular facet for the capitatum is deeply concave and forms a deep obtuse notch in the ventral view. Adjacent to this is a small, elongate, gently convex articular facet for the trapezoid. The medial process is robust, but relatively short and blunt; the facet for the trapezium is not clearly demarked. All features are more clearly demarked in the larger WAM 02.7.26, and the radial facet is more deeply convex than in WAM 02.7.12, but otherwise, the anatomy is very similar.

The relatively large and transversely broad radial articular surface of *C. kitcheneri* extends onto the dorsal surface of the bone to a much greater extent than in *M. fuliginosus* and *Os. rufus*. Also, the facet for hamatum is more concave, the facet for capitatum is much larger, and the lateral process for the trapezium and trapezoid is relatively reduced. These features of *C. kitcheneri* are similar to *D. bennettianus*, though less strongly developed, and in which the radial facet covers the entire proximodorsal surface of the bone.

**Triquetrum (cuneiform)**. The triquetrum is proximodistally compressed, and transversely wider than it is dorsoventrally deep (figures 11*a,b* and 12*m,n*). The fossa for ulnar styloid is circular and moderately deep. The pisiform facet is quadrangular and shallowly concave with a small, deep central foramen. An obtuse angle is formed between the articular facets for the ulna and pisiform, though the intervening crest does not rise above the lateral border. The lateral surface is relatively smooth, and the dorsal surface is very short in the anteroposterior axis.

The triquetrum of *C. kitcheneri* is relatively small and shallow in comparison with that of *M. fuliginosus* and *Os. rufus,* which have a much greater ventrolateral expansion. The styloid fossa is more square from proximal view (cf. triangular) and relatively compressed proximodistally. Overall, the bone is much smaller and smoother. In these features, *C. kitcheneri* is more similar to *D. bennettianus* and *B. illuminata*. The triquetrum of *C. kitcheneri* is distinguished from that of *D. bennettianus* in being longer dorsoventrally, with a much larger facet for the pisiform, and a more gently convex distal articular surface for the hamatum.

**Pisiform**. The pisiform is relatively short and dorsoventrally compressed (figures 11*a* and 12*o–r*). The proximal end is flattened and has a small, round articular facet. The ventral surface of the shaft is concave, and the distal end is bulbous and rounded.

The pisiform of *C. kitcheneri* is similar to that of *D. bennettianus* in being small in relative size to the adjacent carpal bones and having a more compressed shape proximally. In *C. kitcheneri* and *D. bennettianus*, the pisiform is roughly two-thirds the maximum length of the scapholunatum. In comparison, the pisiform of *M. fuliginosus* and *Os. rufus* is massive, and roughly the same length as the maximum dimension of the scapholunatum, with a large, flat, irregular proximal surface, a constricted shaft and very broad and robust distal end for the insertion of the m. flexor carpi ulnaris. The pisiform of *C. kitcheneri* is distinct from *D. bennettianus* in absolute size (roughly twice as large) and having a more concave ventral surface of the shaft.

**Trapezium**. The trapezium is transversely broad, such that it extends past the medial border of the scapholunatum. It is flattened in proximodistal axis and has a sharp distal process on the ulnar side from the dorsal view. The facet for the scapholunatum is flat and ovoid. The articular facet for the first metacarpal is transversely wider than it is deep. The facet for the trapezoid is concave and dorsoventrally deep, but is relatively short in the proximodistal axis. There is a small articular surface for the second metacarpal on the distal aspect toward the ventral/palmar edge.

The trapezium of *C. kitcheneri* is different from that of *M. fuliginosus* and *Os. rufus* in having a more sharply pointed distal process and flatter facet for the scapholunatum.

**Trapezoid**. The trapezoid (figure 12*s–t*) is the smallest of the carpal bones and fits tightly between the trapezium and capitatum. The proximodorsal surface is smoothly rounded. The distal articular surface (for metacarpal II) has a slight mesial crest visible from the dorsal view as a small mesial projection of the distal border.

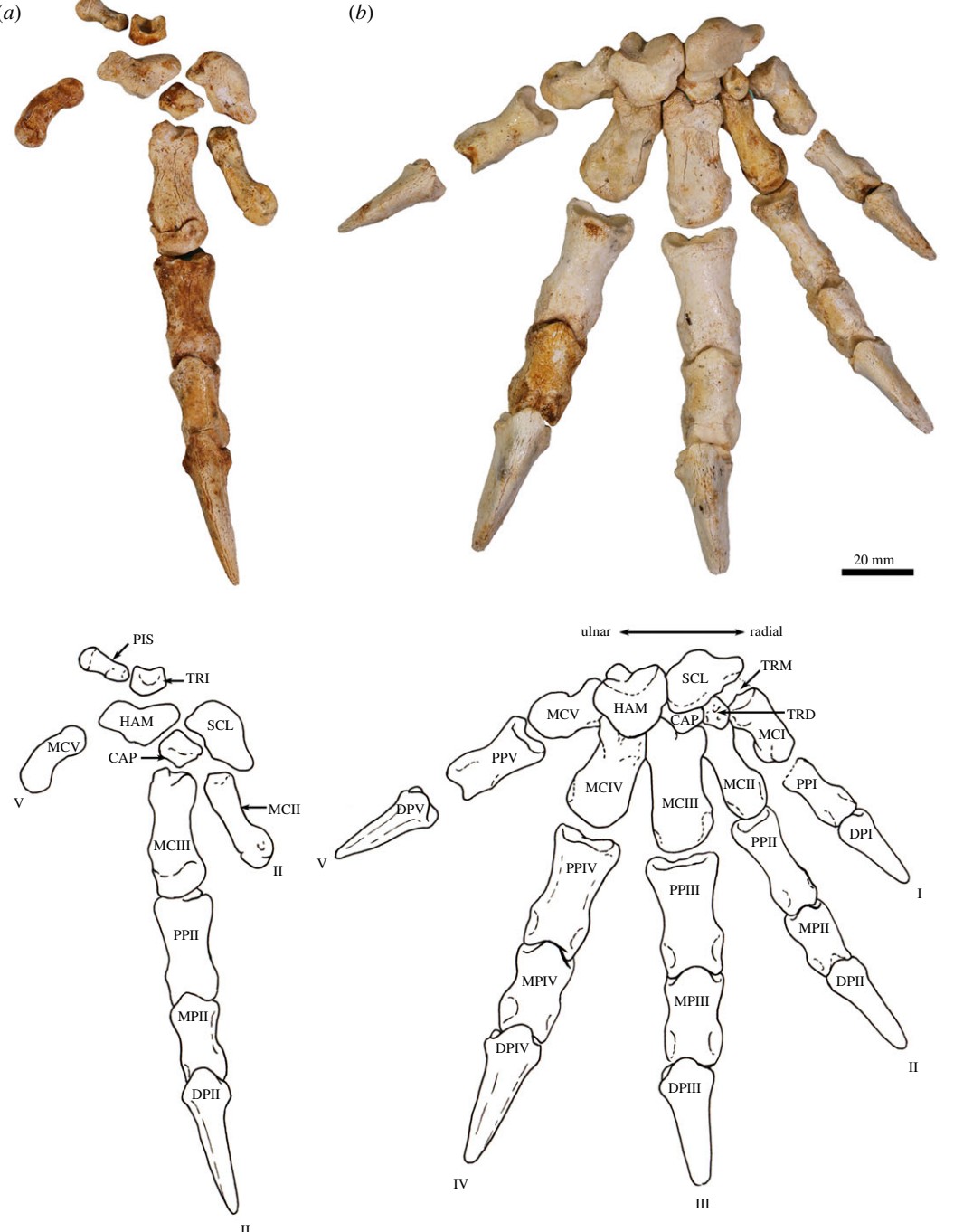

**Figure 11.** Right carpus and manus of *Congruus kitcheneri*. (*a*) WAM 02.7.12 and (*b*) WAM 02.7.26. Scale bar equals 20 mm. Abbreviations: CAP, capitate; DP(I-V), distal phalanges; HAM, hamate; MC(I-V), metacarpal bones; MP(II-V), middle phalanges; PIS, pisiform; PP(I-V), proximal phalanges; SCL, scapholunate; TRD, trapezoid; TRI, triquetral; TRM, trapezium.

There are no clear points of distinction of the trapezoid between species.

**Capitatum (magnum).** The capitatum is pyramidal in dorsal view (figures 11*a,b*, 12*i–l* and 13*i–l*). The proximal articular surfaces for the scapholunatum (dorsomedially) and hamatum (ventrolaterally) are smoothly convex. The relatively broad, steep, medial (scapholunatum) aspect and narrower, more acute lateral (hamatum) face, lock the capitatum between the two large carpal bones in an arrangement reminiscent of the trochlea of a talo-crural hinge joint, with malleoli on either side. The distal articular facet for metacarpal III is large and gently undulating, and is smoothly contiguous with a smaller, deeper facet for metacarpal II. The dorsal distal margin is flexed, with the apex pointing distally. This corresponds to the distinctly concave proximodorsal margin of metacarpal III to form a very tight-fitting and relatively immobile carpometacarpal joint.

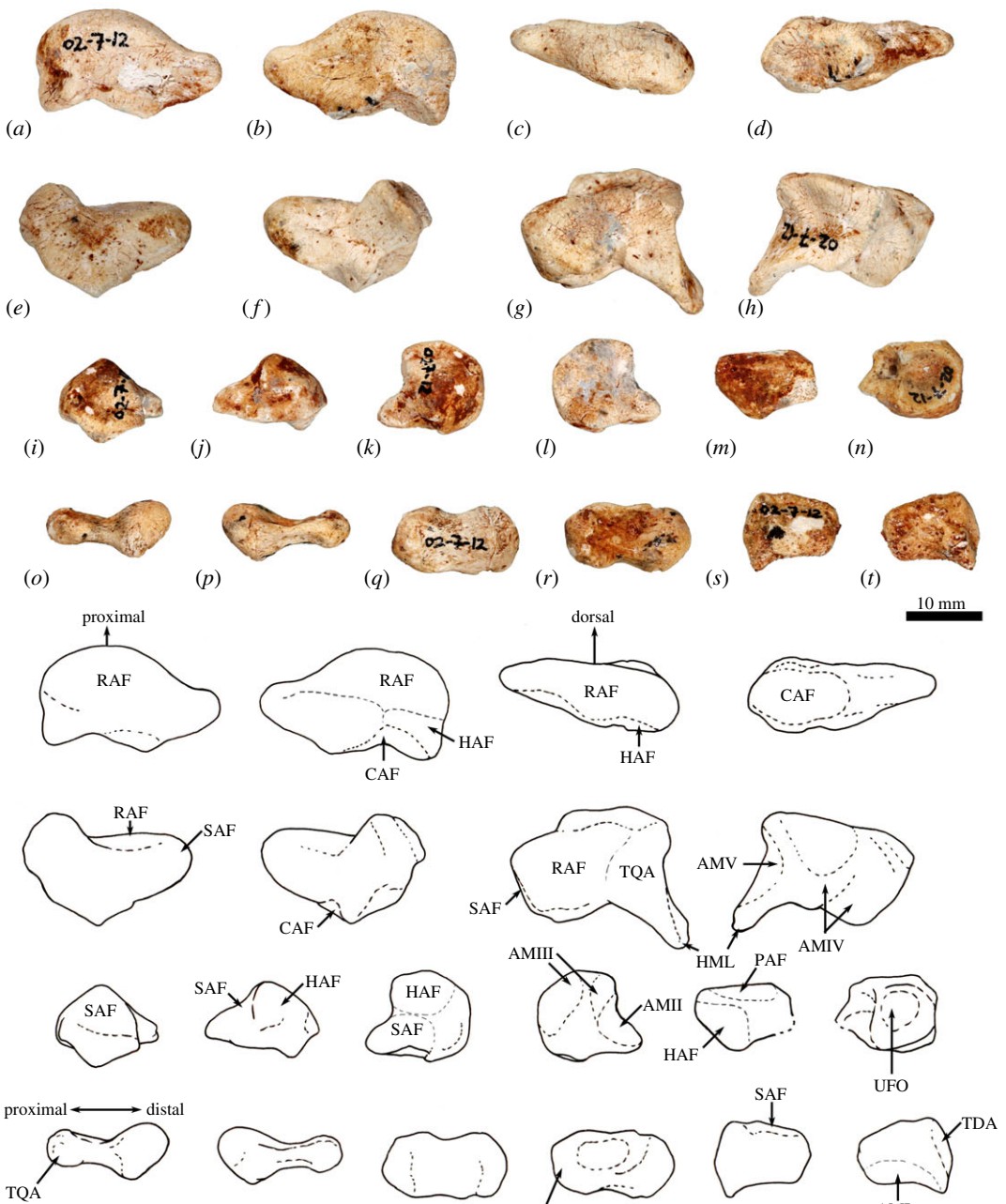

**Figure 12.** Carpal elements of *Congruus kitcheneri*, WAM 02.7.12. (*a*) Right scapholunate dorsal view; (*b*) right scapholunate ventral view; (*c*) right scapholunate proximal view; (*d*) right scapholunate distal view; (*e*) right hamate dorsal view; (*f*) right hamate ventral view; (*g*) right hamate proximal view; (*h*) right hamate distal view; (*i*) right capitate dorsal view; (*j*) right capitate dorsal view; (*k*) right capitate dorsal view; (*l*) right capitate dorsal view; (*m*) right triquetrum distal view; (*n*) right triquetrum proximal view; (*o*) right pisiform dorsal view; (*p*) right pisiform ventral view; (*q*) right pisiform proximal view; (*r*) right pisiform distal view; (*s*) right trapezoid proximal view; (*t*) right trapezoid distal view. Scale bar equals 10 mm. Abbreviations: AMI, articular facet for metacarpal I; AMII, articular facet for metacarpal II; AMIII, articular facet for metacarpal III; AMV, articular facet for fifth metacarpal; CAF, articular facet for capitate; HAF, articular facet for hamate; HML, hamulus (hamate process); RAF, radial articular facet; SAF, articular facet for scapholunate; TDA, articular facet for trapezoid; TQA, articular facet for triquetrum; UFO, ulnar fossa.

The trochlear-like hinge joint formed between the capitatum and the scapholunatum and hamatum in *C. kitcheneri* is distinctive, as is the flexed distodorsal margin over the articulation of metacarpal III. Both *M. fuliginosus* and *Os. rufus* are characterized by an articular surface for the metacarpal that is quite flat. In *D. bennettianus*, this surface is slightly flexed in the same manner as in *C. kitcheneri*, though to a lesser extent.

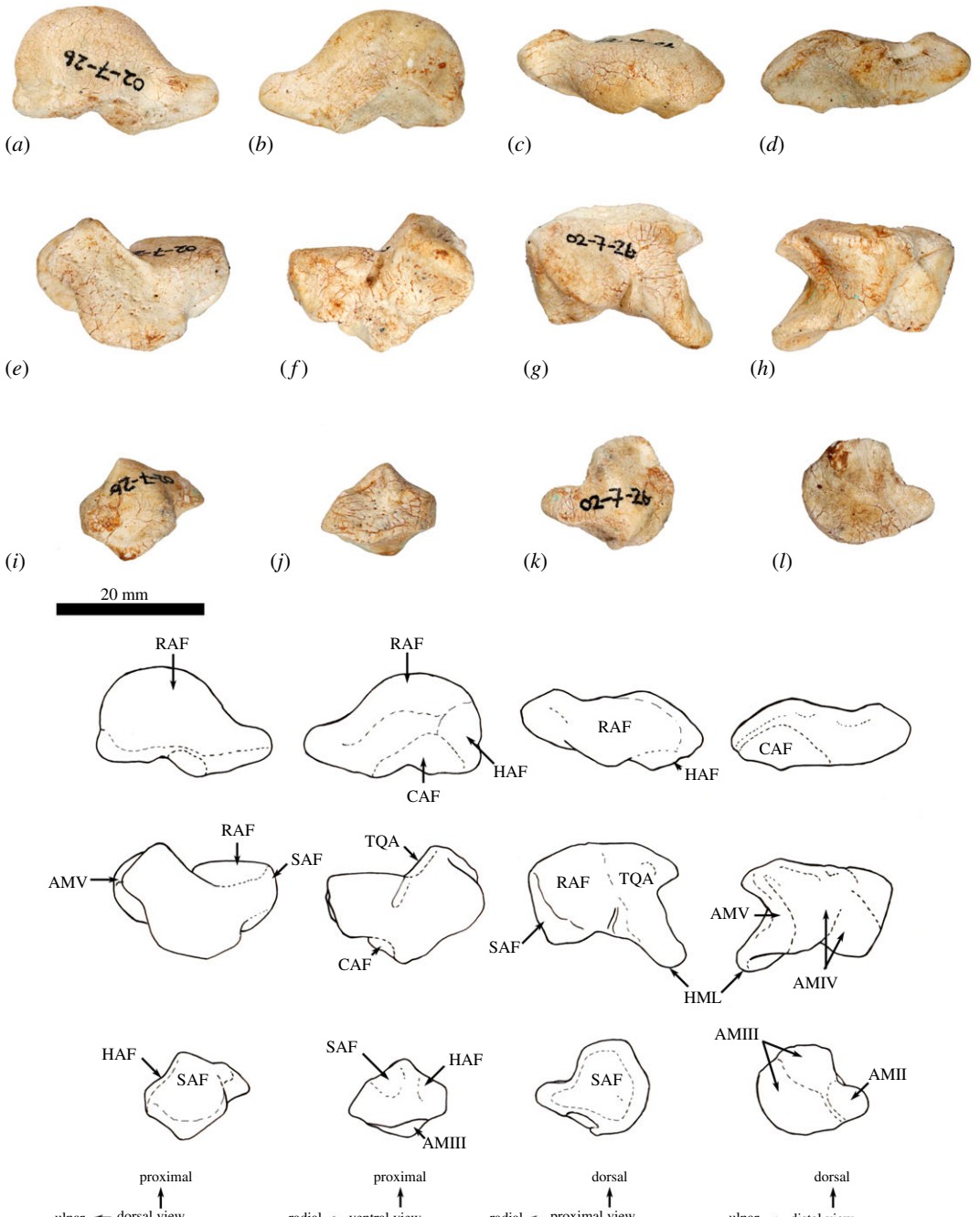

**Figure 13.** Carpal elements of *Congruus kitcheneri*, WAM 02.7.26. (*a–d*) Right scapholunate; (*e–h*) right hamate; (*i–l*) right capitate; from right to left: dorsal view, ventral view, proximal view, distal view. Scale bar equals 20 mm. Abbreviations: AMII, articular facet for metacarpal II, AMIII, articular facet for metacarpal III; AMV, articular facet for fifth metacarpal; CAF, articular facet for capitate; HAF, articular facet for hamate; HML, hamulus (hamate process); RAF, radial articular facet; SAF, articular facet for scapholunate; TQA, articular facet for triquetrum.

**Hamatum (unciform)**. The hamatum is boot-shaped in dorsal view and is transversely wider than proximodistally deep (figures 11*a,b*, 12*e–h* and 13*e–h*). The proximal facet for triquetrum is subtriangular and shallowly concave. The proximomesial process of the hamatum has a small convex articular facet for scapholunatum. The mesial facet for the capitatum is large and concave. The distal facet for the metacarpal IV is subtriangular, with an undulating dorsal margin. The facet for metacarpal V is deeply concave and aligned almost perpendicular to the facet for metacarpal IV, resulting in a strongly abducted posture for metacarpal V. The hamulus (hamatum process) is long and broad, and projects ventrolaterally, such that the canal for the combined flexor tendons is broad and round.

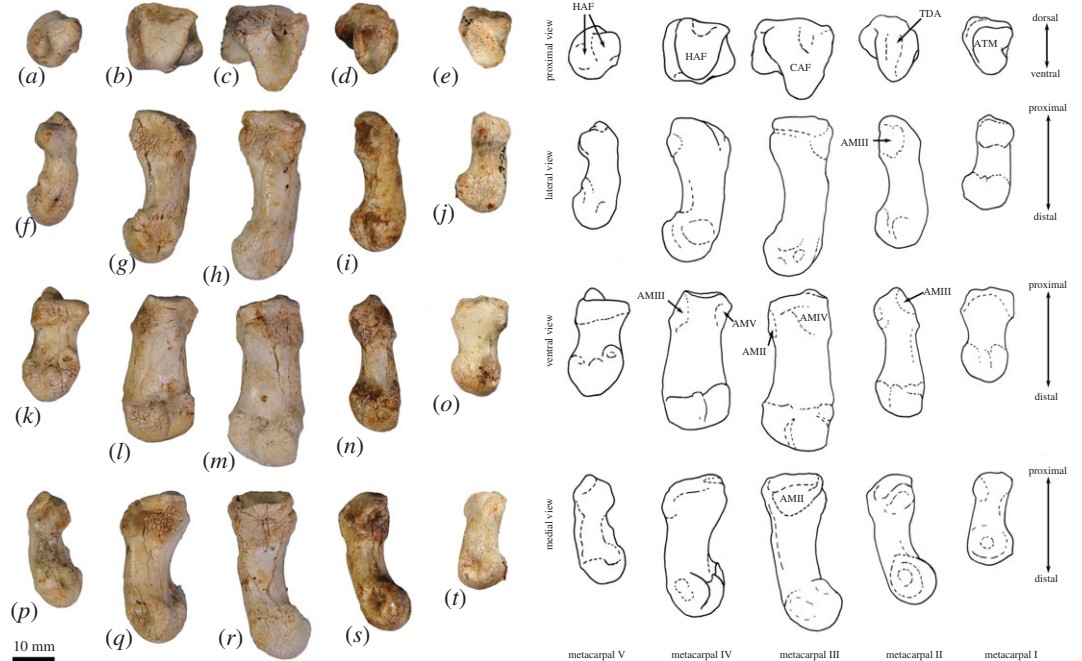

**Figure 14.** Metacarpal bones of *Congruus kitcheneri*, WAM 02.7.26. (*a–e*) Proximal view; (*f–j*) lateral view; (*k–o*) ventral view; (*p–t*) medial view; from right to left: metacarpals I–V. Scale bar equals 10 mm. Abbreviations: AMII, articular facet for metacarpal II, AMIII, articular facet for metacarpal III; AMV, articular facet for fifth metacarpal; ATM, articular facet for trapezium; CAF, articular facet for capitate; HAF, articular facet for hamate; TDA, articular facet for trapezoid.

The hamatum of *C. kitcheneri* is smaller than that of *M. fuliginosus* and *Os. rufus*, in which the articular facet for the triquetrum has almost a 90° flexion in dorsal view. In *M. fuliginosus* and *Os. rufus*, the scapholunatum overlaps the hamatum proximomesially, forming a pseudo-saddle joint. In *C. kitcheneri*, the proximomesial process of the hamatum abuts the mesial concavity of the scapholunatum, forming a junction that is much more in the longitudinal axis of the carpus. In *M. fuliginosus* and *Os. rufus*, the hamulus is much longer, more robust and distally extended. In all of these features, the hamatum of *C. kitcheneri* is much more similar to that of *D. bennettianus*, though in the latter the hamatum is even more dorsoventrally compressed and has a more deeply concave facet for metacarpal IV and a more reduced hamulus.

**Metacarpals**. All five metacarpals are short and robust (figure 14). Metacarpals III–IV are slightly longer and substantially more robust than metacarpal II. Metacarpal I is the shortest of the series and has an abducted posture in articulation. On metacarpal I, the articular facet for the trapezium is long in the dorsoventral axis, convex in lateral view and set deep in the proximal end, in contrast with the smaller, raised, subtriangular facet for the trapezoid. The shaft of metacarpal I is short and thick, with a slightly raised margin on the abaxial side. The distal articular facet for the proximal phalanx has a slight abaxial deflection, and the fossa for the collateral ligament on the abaxial side is deeper than that on the axial side. The proximal end of metacarpal II is mediolaterally compressed with a deep proximodorsal groove. There is no distinct articular surface for metacarpal I, but a large arcuate articular facet for metacarpal III is present on the axial side. There is a small proximodorsal scar for the m. extensor carpi radialis. The distal epiphysis is bulbous and asymmetrical, such that the articular surface is inclined away from the mesial axis of the hand, i.e. toward the abaxial side. The proximal end of metacarpal III is mediolaterally compressed and dorsoventrally deep, with an acute subtriangular carpal surface. Metacarpal III is asymmetrical in dorsal view, with an enlarged crest on the ulnar side and a large proximodorsal scar for the m. extensor carpi radialis. There is a slight degree of lateral torsion of the short, robust shaft, and the distal epiphysis is asymmetrical with the articular surface deviating slightly to the radial side. The proximal articular surface of metacarpal IV flares slightly to the ulnar side. Metacarpal IV has a wider carpal surface than metacarpal III, and a stout, robust shaft that deviates slightly to ulnar side. Metacarpal V is short and robust and forms a mobile saddle joint with the hamatum in a strongly abducted posture. The proximal surface has a short dorsal groove and a laterally directed process on the ulnar side. The shaft is dorsally convex and bulbous distally. The distal articular surface is inflected strongly to the ulnar side of the hand.

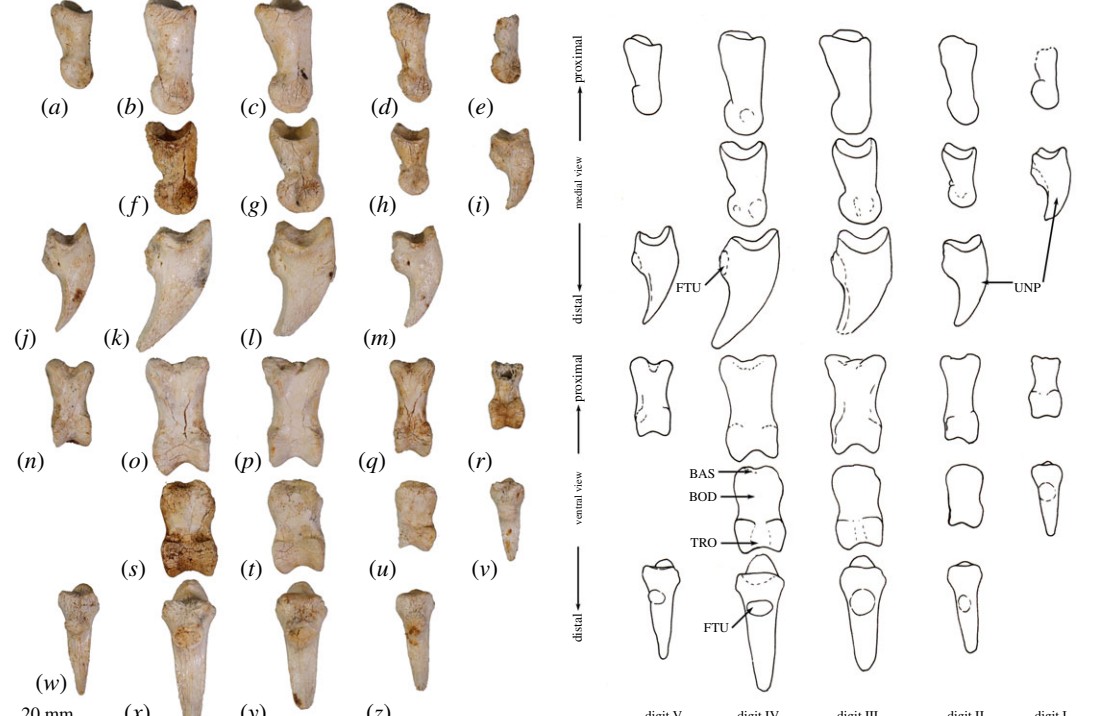

**Figure 15.** Manual phalanges (right) of *Congruus kitcheneri*, WAM 02.7.26. Medial view (*a–m*) and ventral view (*n–z*). (*a,n*) proximal phalanx V; (*b,o*) proximal phalanx IV; (*c,p*) proximal phalanx III; (*d,q*) proximal phalanx II; (*e,r*) proximal phalanx I; (*f,s*) middle phalanx IV; (*g,t*) middle phalanx III; (*h,u*) middle phalanx II; (*i,v*) distal phalanx I; (*j,w*) distal phalanx V; (*k,x*) distal phalanx IV; (*l,z*) distal phalanx III; (*m,z*) proximal phalanx II. Scale bar equals 20 mm. Abbreviations: BAS, base; BOD, body; FTU, flexor tubercle; TRO, trochlea; UNP, ungular process.

In comparison with other kangaroos of comparable size, the metacarpals of *C. kitcheneri* are very robust. In *C. kitcheneri*, metacarpal II is reduced in transverse width and robustness relative to metacarpals III–IV to a much greater extent than in the other taxa examined. The distal abaxial deflection of metacarpal II is similar to that of *D. bennettianus*. The proximal articular surface of metacarpal III forms a very acute triangle in shape, which is distinct from any other species examined. Similarly, the highly abducted posture and relative mobility of metacarpal I and especially metacarpal V is distinctive to *C. kitcheneri*.

**Manual Phalanges**. The manual phalanges (figure 15) are large and robust relative to the size of the carpals and metacarpals, in particular the phalanges of digits III–IV (figure 16*a*). In ventral view, the distal articular surfaces of the proximal phalanges form very deep trochlear grooves with high lateral crests. The middle phalanges are relatively long, and very similar in shape to the proximal phalanges, with dorsoventrally deep bodies and deep distal trochlear facets. There is an obvious sulcus on either side of the distal palmar aspect of each middle phalanx marking the insertion of the m. flexor digitorum superficialis. The distal phalanges are massive and strongly curved toward their distal extremity. The ungual processes, which would have supported the claws, are very deep in the dorsoventral plane and are subtriangular in cross-section, with a strong dorsal ridge. The flexor tubercles of each distal phalanx are relatively large, broad and rugose.

In comparison with kangaroos of comparable size, the phalanges of *C. kitcheneri*, particularly the middle and distal phalanges, are long relative to the length of the metacarpals. This is similar to the pattern observed in *D. bennettianus* and the opposite to those of ground-dwelling taxa, including *M. fuliginosus* and *Os. rufus*, which have much shorter manual digits. In the latter species, the middle phalanges are approximately half the length of their respective proximal phalanx, and are wider and more dorsoventrally flattened. Sthenurine kangaroos have long digits, but have much longer metacarpals, and the distal phalanges are typically long and flat. *Protemnodon brehus* has relatively short and broad manual phalanges. The distal phalanges of *C. kitcheneri* are relatively massive, long and strongly recurved in comparison with *M. fuliginosus* and *Os. rufus*. Although they are most similar in form to those of *D. bennettianus*, though in the latter species, the distal phalanges are larger and longer than the metacarpals.

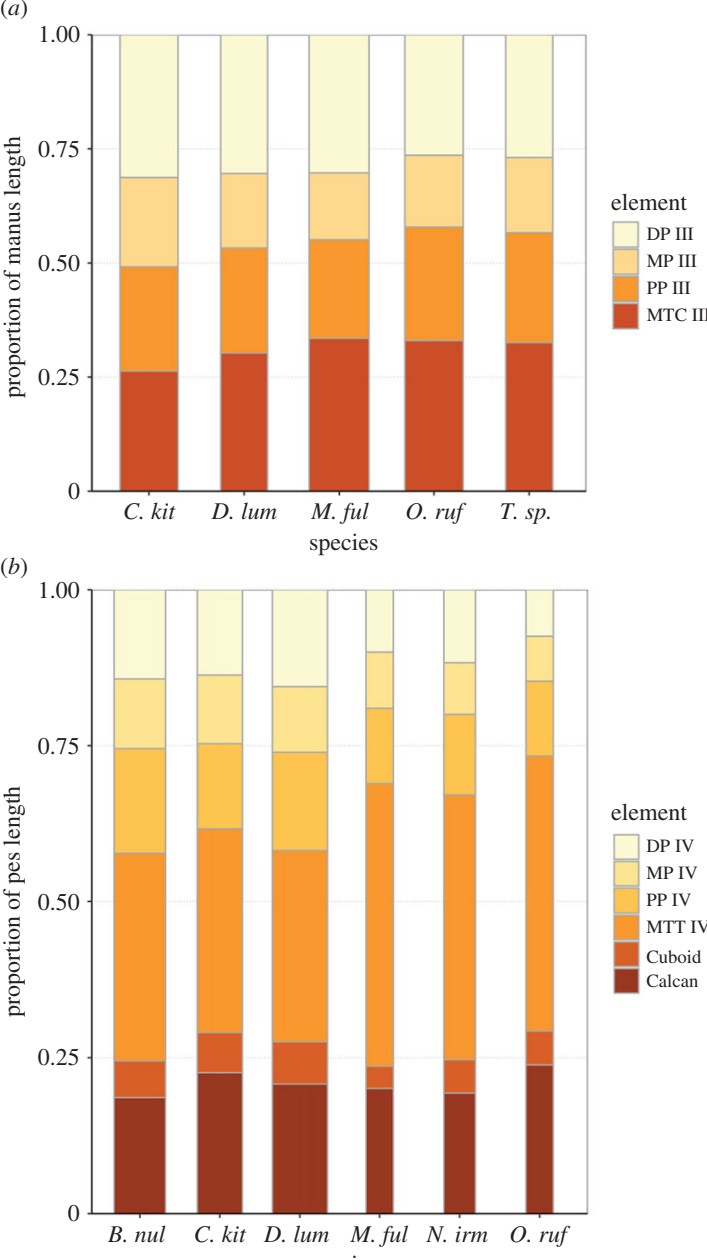

**Figure 16.** Comparative relative phalangeal length. (*a*) Proportional length of elements in the third manual digit ray (relative widths calculated average width of all elements divided by length of metacarpal III); (*b*) proportional length of elements of the fourth pedal digit ray as a proportion of the length of the pes (relative widths calculated average width of all elements divided by length of metatarsal IV). Taxa: *B. nul, Borha nullabora; C. kit, Congruus kitcheneri; D. lum, Dendrolagus lumholtzi; M. ful, Macropus fuliginosus; N. irm, Notamacropus irma; O. ruf, Osphranter rufus*. Abbreviations: Calcan, calcaneus; DP, distal phalanx; MP, middle phalanx; MTC, metacarpal; MTT, metatarsal; PP, proximal phalanx.

**Innominate**. The ilium is elongate and subtriangular in cross-section (figure 17*a,b*). The iliac fossa is deeply concave, while the gluteal fossa is shallow and transversely broad. The medial surface is subtriangular, flaring gradually towards the sacroiliac joint. The origin of m. rectus femoris forms a shallow subtriangular depression bounded by a rugose border and is separated from the iliocotylar tuberosity of the acetabular rim by a short non-articular area (*ca* 10 mm long; figure 17*b*). The acetabulum is large and deeply concave, with a high dorsal rim, and strongly developed iliocotylar tuberosity. The broad ischiocotylar and narrow pubocotylar tuberosities are very close to each other above a deep intercotylar notch. The ilio-ischiatic junction is marked by a slightly raised ischiatic spine. The ischiatic tuberosity is broad (figure 17*c*). The ischiatic table is relatively thin, and there is a

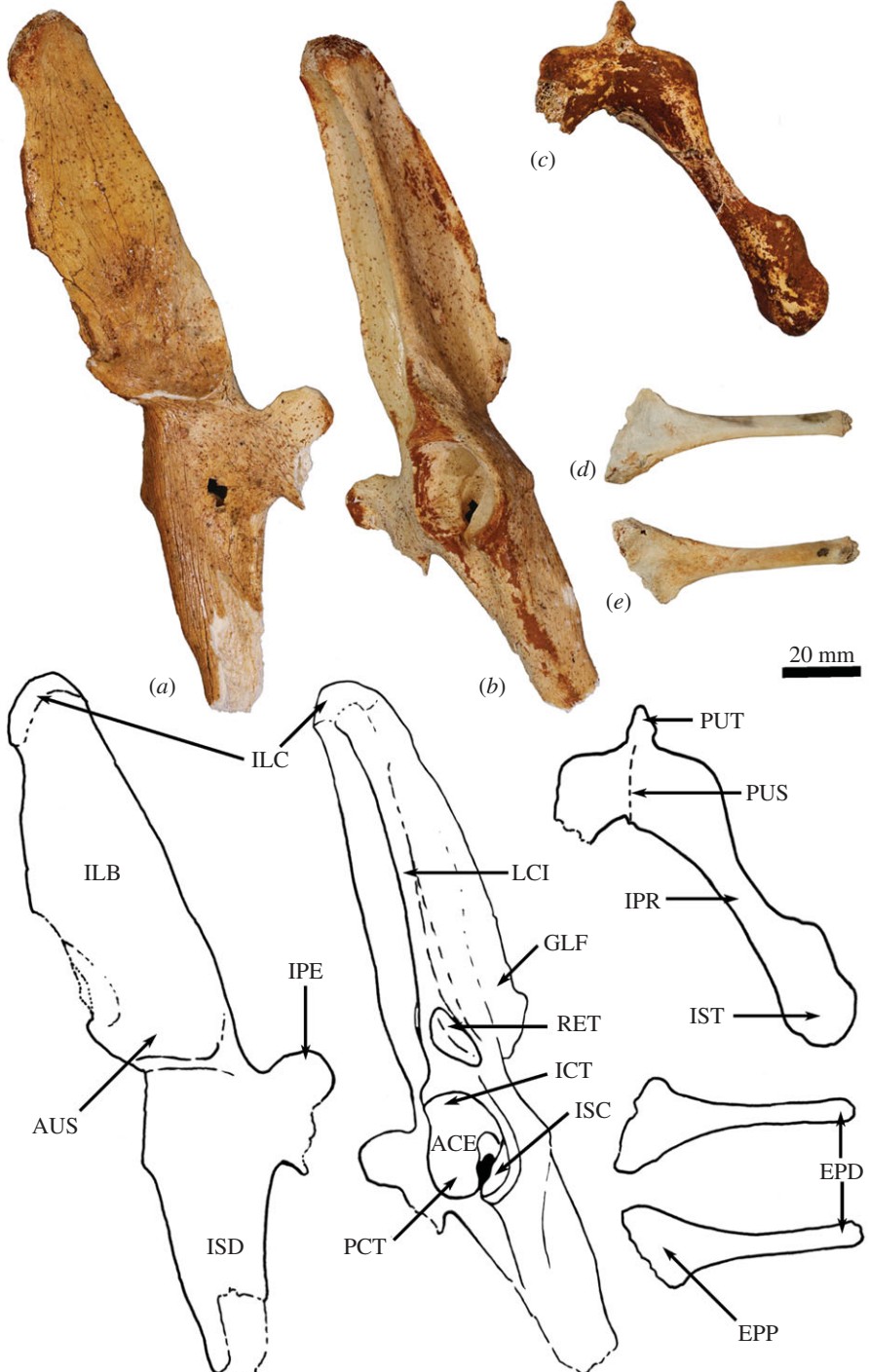

**Figure 17.** Pelvic elements of *Congruus kitcheneri*, WAM 02.7.12. Left innominate (*a*) medial view and (*b*) lateral view; (*c*) left pubis and ischium caudal view; left epipubic bone (*d*) medial view and (*e*) lateral view. Scale bar, 20 mm. Abbreviations: ACE, acetabulum; AUS, auricular surface; EPD, epipubic distal extremity; EPP, epipubic proximal extremity; GLF, gluteal fossa; ICT, iliocotylar tubercle; ILB, iliac body; ILC, iliac crest; IPE, iliopubic eminence; IPR, ishiopubic ramus; ISC, ischiocotylar tubercle; ISD, descending ramus of ischium; IST, ischiatic tuberosity; LCI, lateral crest of the ilium; PCT, pubocotylar tubercle; PUS, pubic symphysis; PUT, pubic tubercle; RET, tubercle of the m. rectus femoris.

low ridge demarking the boundary of the m. obturator externus from the ischiatic ramus. The craniolateral angle of the obturator foramen is obtuse. The caudal symphyseal epiphysis is very robust and transversely broad; the ventral border is almost square. In caudal view, the angle made between the two ischia is relatively acute (*ca* 75°). The medioventral ischiatic tubercle is prominent and robust, though not especially elongate. The iliopectineal eminence is very large and square in outline and

flared medioventrally (figure 17*a,b*). The lateral crest of ilium in WAM 02.7.26 is marked by an unusual deep, longitudinal groove along the posterior edge of the crest, within the gluteal fossa. A distinct, though much subtler, longitudinal groove is also present on the ilium in WAM 02.7.12.

The innominate of *C. kitcheneri* is most like those of *M. fuliginosus* and *Os. rufus*, though the ilium is proportionately shorter, iliac fossa is more deeply concave, descending body of the ischium is narrower and less concave mesially, and acetabulum is slightly deeper. The deeply concave iliac fossa of *C. kitcheneri* is most similar to those of *Pe. xanthopus* and *T. billardierii*, though in *Petrogale* spp. the ilium is strongly curved and flares laterally toward the iliac crest. The deep circular acetabulum of *C. kitcheneri* is distinct from those of the smaller ground-dwelling macropodines and *D. bennettianus*, wherein the acetabulum is relatively shallow and the intercotylar notch more open. The iliac crest is very much wider in *Pr. brehus* than in *C. kitcheneri*. The deep longitudinal groove along the caudolateral border of the ilium is unique to *C. kitcheneri*.

**Epipubic**. The epipubic bones are robust, with a straight and dorsoventrally flattened body (figure 17*d,e*). The proximal end is very broad and particularly robust around the pubic articulation. The origin of pectineus muscle is proximally located.

The epipubic bone of *C. kitcheneri* is most like that of *D. bennettianus*, in contrast with *Macropus fuliginosus*, *Os. rufus*, *Pe. xanthopus* and *T. billardierii*, in which the epipubic bones are more gracile, with a smaller proximal articular process and more distally positioned pectineal tubercle. The epipubic of *C. kitcheneri* is distinct from that of *D. bennettianus* in being more robust throughout, particularly proximally around the pubic articulation.

**Femur**. The diaphysis of the femur is very robust (figures 18*a–d* and 19*a–d*). The head is hemispherical and protrudes medially and slightly cranially (figure 18*e*). The cranial proximal surface, between the head and the lateral trochanteric crest is short and broad. The lateral crest from the greater trochanter is short, extending to level with the proximal third of the lesser trochanter. The lesser trochanter is elongate, and the distal half is distinctly concave on the cranial surface. From the cranial view, the narrowest part of the shaft is between the lesser trochanter and quadratus tubercle. From that point, the diaphysis expands gradually to the distal epiphysis. In lateral view, the shaft is very slightly cranially curved. The quadratus tubercle is rugose and strongly medially displaced. The distal diaphysis is broad and slightly compressed anteroposteriorly. The caudomedial fossa (m. gastrocnemius insertion) and caudolateral fossa (m. flexor digitorum superficialis insertion) are relatively deep. The trochlear crests are broad and smoothly rounded; the lateral crest is roughly twice as broad as, and slightly higher than, the medial crest. In distal view, the distal epiphysis is trapezoidal in outline. The medial condyle is slightly longer than the lateral condyle in the proximodistal axis. The lateral condyle is flared laterally for articulation with the fibula. In WAM 02.7.12, the distal condyles are relatively narrow and widely spaced, whereas in WAM 02.7.26 the condyles are much wider, resulting in a narrower intercondylar fossa.

The femur of *C. kitcheneri* is larger and more robust than in extant forms, which are all relatively more gracile. The femoral head is relatively small, the neck is short, and the proximal shaft is broad in comparison with *M. fuliginosus* and *Os. rufus,* as well as *Pr. anak*. In these aspects, the femur of *C. kitcheneri* is most similar to *T. billardierii*, in which the proximal end is broad, and the greater trochanter moderately long but very broad (particularly at its base adjacent to the trochanteric fossa). Other similarities between *C. kitcheneri* and *T. billardierii* include the long and concave lesser trochanter, and distally the low rounded trochlear crests, narrow distal condyles and moderately developed lateral epicondyle. The medial displacement of the m. quadratus femoris in *C. kitcheneri* is distinctive from all other taxa, including *Pr. anak* and *Pr. brehus*, in which the scar for the m. quadratus femoris lies in the middle of the caudal surface. In species of *Petrogale* the femur is long and gracile, and the m. quadratus femoris scar more proximally placed.

**Tibia**. The tibia is elongate with a high anterior tibial crest (figure 20*a*). The tibial plateau is transversely wider than it is anteroposteriorly deep. In the proximal view, the lateral tibial condyle is extended laterally, particularly in WAM 02.7.12. The lateral border of the epiphysis, running between the tibial crest and the fibular facet, is deeply concave in proximal view (figure 20*b,d*). The medial condyle is long and larger in surface area than the lateral condyle. The medial articular surface is shallowly concave, while the lateral surface is very slightly convex. The proximal fibular facet is orientated at *ca* 45° to the anteroposterior axis. The diaphysis is long and moderately robust. The medial edge is almost straight in cranial view (similar to that of *Os. rufus*), while the interosseous border is concave proximally, before flaring toward the proximal end of fibular facet. The fibular facet covers two-thirds of the length of the distolateral diaphysis. A distinct longitudinal groove runs adjacent to the fibular facet from the mid-shaft for two-fifths of the length of the tibial diaphysis. The

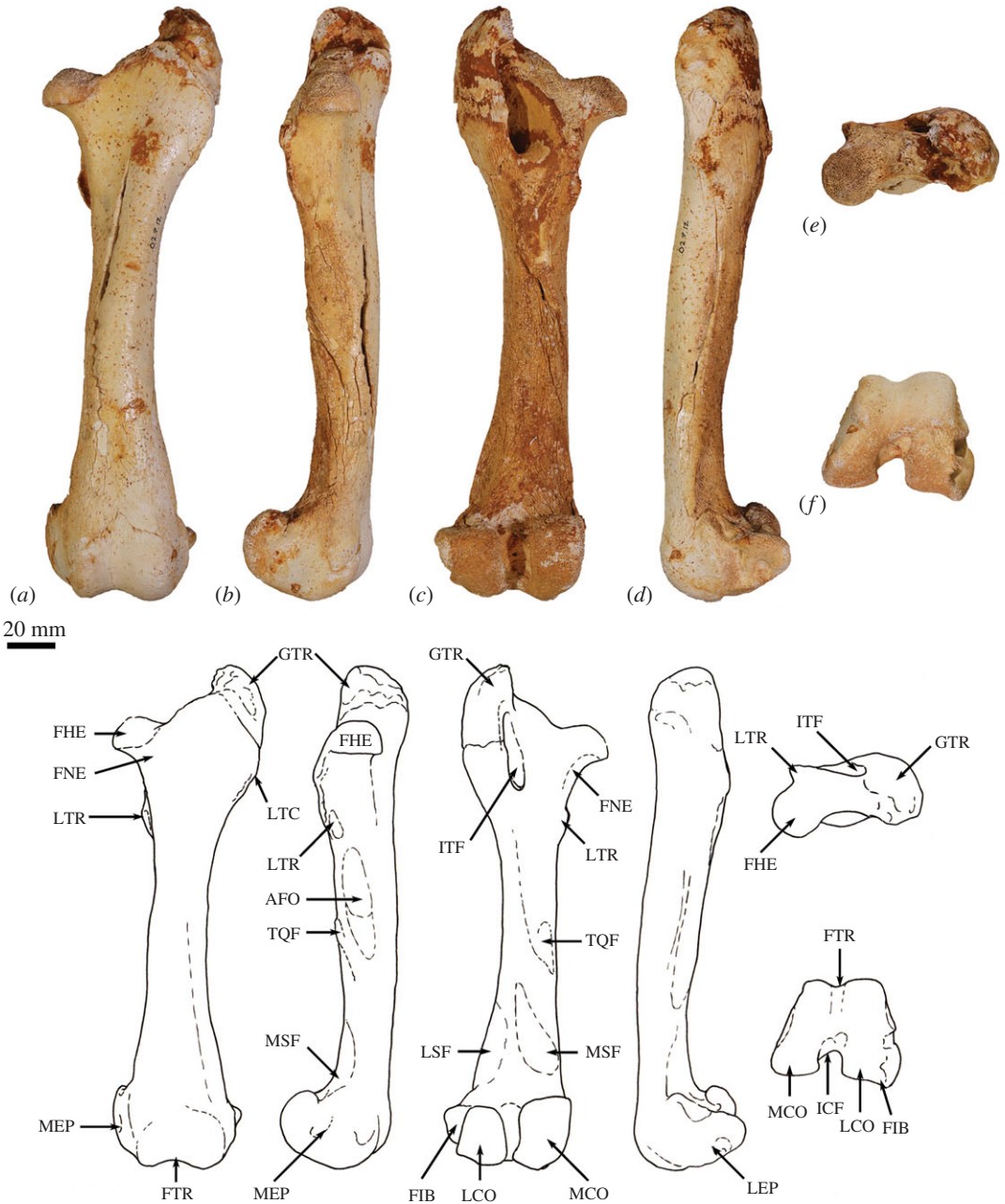

**Figure 18.** Femur of *Congruus kitcheneri*, WAM 02.7.12. (*a*) Left femur in cranial view; (*b*) left femur in lateral view; (*c*) left femur in caudal view; (*d*) left femur in medial view; (*e*) left femur in proximal view; (*f*) left femur in distal view. Scale bar equals 20 mm. Abbreviations: AFO, fossa for m. adductors; FHE, femoral head; FIB, fibular articular surface; FTR, femoral trochlea; GTR, greater trochanter; ICF, intercondylar fossa; ITF, intertrochanteric fossa; LCO, lateral condyle; LEP, lateral epicondyle; LSF, lateral supracondylar fossa (for m. flexor digitorum superficialis); LTC, lateral trochanteric crest; LTR, lesser trochanter; MCO, medial condyle; MEP, medial epicondyle; MSF, medial supracondylar fossa (for m. gastrocnemius); TQF, tubercle for m. quadratus femoris.

medial malleolus is blunt and robust (figure 20*f,g*). The distal articular surface is deep medially and flared laterally.

The crural index (tibia-to-femur-length ratio) of *C. kitcheneri* is 1.6, which is intermediate between those of large ground-dwelling macropodine taxa (in which it approaches 2.0) and smaller ground-dwelling forms, such as *T. billardierii* (1.3), *N. eugenii* (1.2) and rock-wallabies e.g. *Pe. lateralis* (1.3). In species of *Dendrolagus*, the tibia is shorter than the femur. The proximal tibial epiphysis of *C. kitcheneri* is wider than it is anteroposteriorly long, rather than the reverse, which distinguishes it from the ground-dwelling macropodines, including *Protemnodon*. In this characteristic, it is similar to

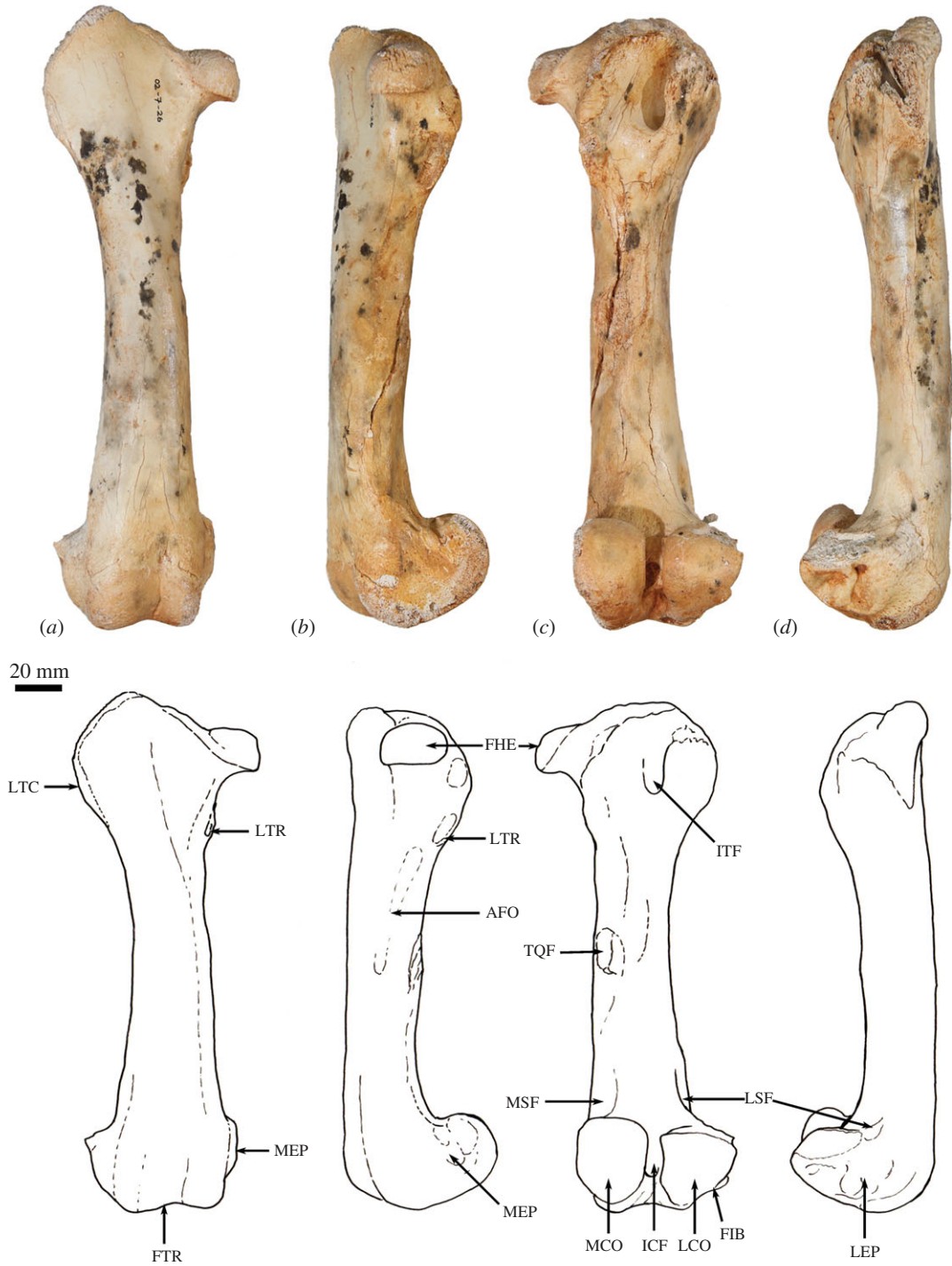

**Figure 19.** Femur of *Congruus kitcheneri*, WAM 02.7.26. (*a*) Left femur in cranial view; (*b*) left femur in lateral view; (*c*) left femur in caudal view; (*d*) left femur in medial view. Scale bar equals 20 mm. Abbreviations: AFO, fossa for m. adductors; FHE, femoral head; FIB, fibular articular surface; FTR, femoral trochlea; ICF, intercondylar fossa; ITF, intertrochanteric fossa; LCO, lateral condyle; LEP, lateral epicondyle; LSF, lateral supracondylar fossa (for m. flexor digitorum superficialis); LTC, lateral trochanteric crest; LTR, lesser trochanter; MCO, medial condyle; MEP, medial epicondyle; MSF, medial supracondylar fossa (for m. gastrocnemius); TQF, tubercle for m. quadratus femoris.

*D. bennettianus,* in which the anterior tibial crest is very much reduced, resulting in a very short, broad tibial plateau, while in *Petrogale* the length and width of the tibial plateau are close to equal. The larger area of the medial versus lateral condyle is also more like that of *D. bennettianus*. In surface area, the condyles are generally more subequal is size in *M. fuliginosus, Os. rufus* and *Pe. lateralis*. In *S.*

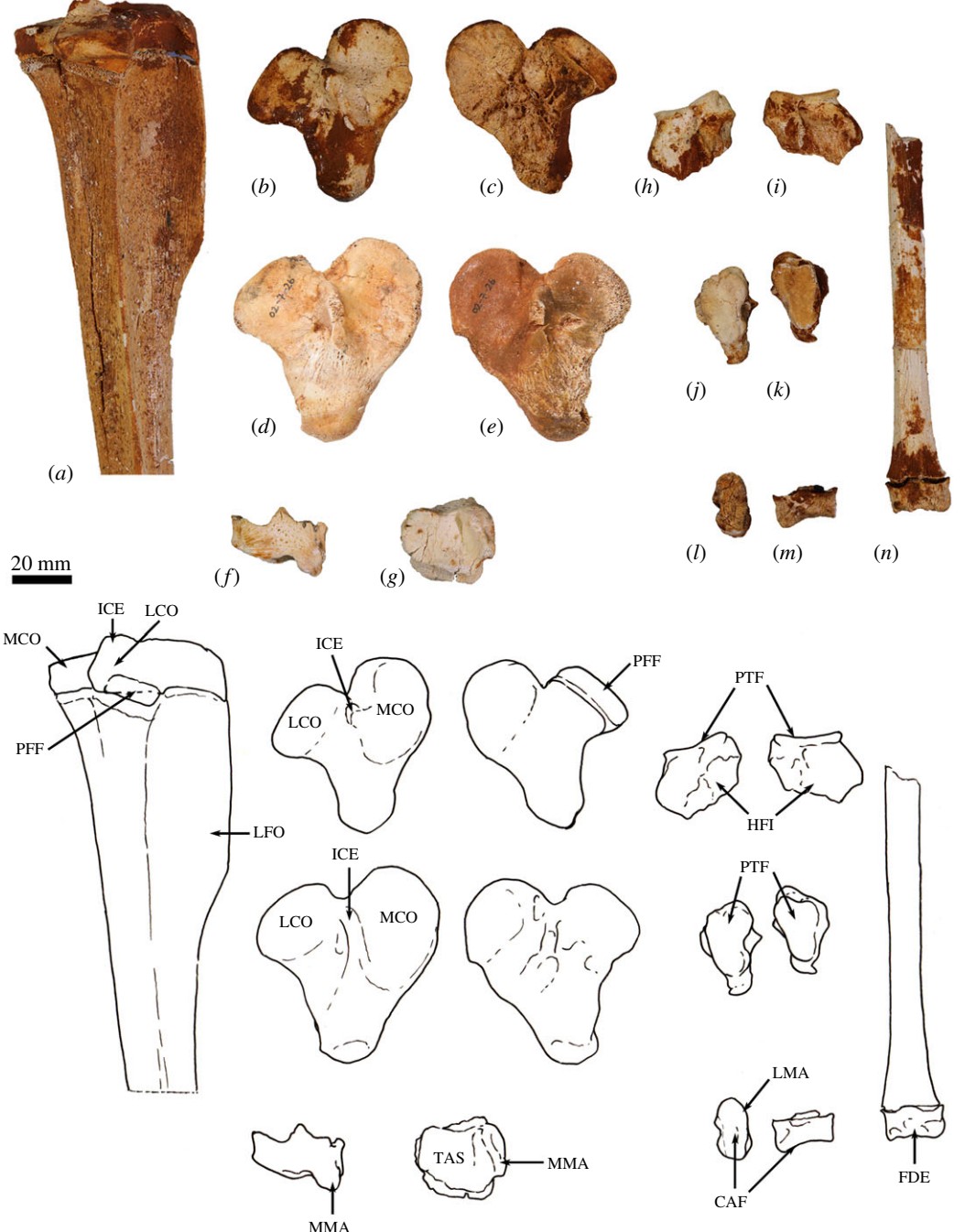

**Figure 20.** Tibia and fibula of *Congruus kitcheneri*, WAM 02.7.12 and WAM 02.7.26 (above), and annotated outline drawings (below). (*a*) Proximal right tibia 02.7.12 caudolateral view; (*b*) left proximal epiphysis tibia 02.7.12 proximal view; (*c*) left proximal epiphysis tibia 02.7.12 distal view; (*d*) left proximal epiphysis tibia 02.7.26 proximal view; (*e*) right proximal epiphysis tibia 02.7.26 proximal view; (*f*) distal right tibial epiphysis 02.7.12 cranial view; (*g*) distal right tibial epiphysis 02.7.12 distal view; (*h*) proximal left fibular epiphysis 02.7.12 medial view; (*i*) proximal right fibular epiphysis 02.7.12 medial view; (*j*) proximal left fibular epiphysis 02.7.12 proximal view; (*k*) proximal right fibular epiphysis 02.7.12 proximal view; (*l*) distal right fibular epiphysis 02.7.12 distal view; (*m*) distal right fibular epiphysis 02.7.12 medial view; (*n*) distal right fibular shaft 02.7.12 lateral view. Scale bar equals 20 mm. Abbreviations: CAF, calcaneal articular facet; FDE, fibular distal epiphysis; HFI, head of fibula; ICE, intercondylar eminence; LCO, lateral condyle; LFA, lateral fossa; LMA, lateral malleolus; MCO, medial condyle; MMA, medial malleolus; PFF, proximal fibular facet; PTF, proximal tibial facet; TAS, talar articular surface.

*brachyurus*, the lateral condyle is larger than the medial condyle, and the anterior tibial crest is much blunter. The proximal fibular facet of *C. kitcheneri* is much shorter than in *M. fuliginosus* and *Os. rufus*. The lateral groove on the middle of the diaphysis of *C. kitcheneri* is unique among macropodines.

**Fibula**. The proximal articular facet of the fibula is expanded anteroposteriorly into a moderately long and shallow shelf that articulates with the groove of the outer tibial condyle (figure 20*h–k*). The medial edge of the articulation curves downwards into a distinct lip. Below the lip runs a deep, narrow transverse groove. Posterolaterally, the articular surface expands over a broad, rounded tuberosity. The distal half of the diaphysis is flattened and strongly concavo-convex in section, where it articulates along the tibia (figure 20*n*). The distal epiphysis is transversely compressed and craniocaudally deep, and the articular facet is gently concave (figure 20*l–m*).

The proximal fibular epiphysis of *C. kitcheneri* is similar to those of large ground-dwelling forms (*M. fuliginousus* and *Os. rufus*) in being fairly low and flat and that the shelf of the articular surface is relatively shallow. By contrast, in *Pe. lateralis* and *S. brachyurus* the articular shelf is more deeply cut and the lateral aspect of the fibular head extends proximally above the articular facet. In *D. bennettianus*, the articular shelf is very deep and proximal expansion of the fibular head relatively massive in comparison with *C. kitcheneri*. *Protemnodon brehus* has a more deeply concave articular shelf and more pronounced proximolateral expansion of the fibular head. The articular surface of the distal fibular epiphysis in *C. kitcheneri* is shallowly concave, similar to *M. fuliginosus*, in contrast with *D. bennettianus* that has more deeply concave articular surface and a rounded cranial prominence.

**Calcaneus**. The tuber calcanei is narrow cranially and expands caudally towards the epiphysis, which is wider than it is deep (figure 21*a–e*). The rugose plantar surface is subtriangular in outline and slightly concave. The cranial margin of the rugose surface is rounded before a broad plantar sulcus crosses almost directly transversely from the flexor sulcus of the sustentaculum tali. The cranial plantar tuberosity on the boundary of the ventromedial facet of the calcaneal-cuboid articulation is small and weakly developed. The sustentaculum tali is broad transversely and dorsoventrally deep, with a deep flexor sulcus that lies in a plane halfway between the dorsal and plantar surfaces of the calcaneal tuberosity. From the medial aspect, the sustentaculum tali follow a ventrally convex arc that passes anteroventrally along the medial side of the base of the tuber calcanei.

The lateral talar facet is moderately high and cylindrical (not tapered medially), while the medial facet is ovoid with a smoothly convex caudal boundary onto sustentaculum and clearly marked cranial border (figure 21*a*). The lateral and medial facets are separated by a very narrow gap (*ca* 1 mm). The anteromedial facet for articulation with talar head is abraded, though appears to have been relatively small. The fibular facet is large, laterally expanded and subtriangular in outline. A deep concave area immediately cranial to the fibular contact represents the attachment of the fibular ligaments. The dorsomedial facet for the cuboid is transversely broad and rectangular from the cranial view (figure 21*f*), and convex and obliquely orientated from the dorsal view. The dorsolateral facet is slightly smaller and squarer in outline than the dorsomedial facet. The step between the two facets is obliquely orientated when viewed dorsally (figure 21*a*). The small ventromedian facet is contiguous with dorsolateral facet. It is transversely wider than deep, and in lateral view, is more caudally placed than the dorsal facets, being separated mesially from the lateral half of the dorsomedial facet by a steep notch.

The calcaneus of *C. kitcheneri* is much wider than in any extant macropodines except for the species of *Dendrolagus*. In overall shape and proportions, the calcaneus of *C. kitcheneri* most resembles that of *T. billardierii*, with a moderately long tuber calcanei that is broad caudally but constricted cranially. In *S. brachyurus* and *Pr. anak*, the tuber calcanei is subtriangular in plantar view, but not as broad caudally. In *M. fuliginosus*, *Os. rufus* and *Pe. xanthopus* the tuber calcanei is longer, narrower and more rectangular, whereas in *D. bennettianus*, the tuber is much shorter and wider. The calcaneus of *C. kitcheneri* is distinct from that of *T. billardierii* in being absolutely much larger, with flatter caudal talar facets, a more obliquely oriented dorsomedial cuboid facet, a deeper ventromedian cuboid facet, and a more ventrally placed rather than cranially extended tubercle for the cranial fibular ligament. The wider-than-deep cuboid facets distinguish *C. kitcheneri* from all ground-dwelling forms, including *Pr. anak*.

**Talus (Astragalus)**. The talus is transversely wider than it is long craniocaudally (figure 21*g–i*). In articulation, the talar head does not extend to the cranial border of the calcaneus and has no articulation with the cuboid. The trochlear fossa is longer than it is transversely broad and smoothly concave, with the deepest part of the groove being medially displaced (figure 21*j*). The trochlear crests run parallel to one another and follow a path that is only very slightly oriented laterally (*ca* 15°) to the longitudinal axis of the bone. The medial crest is higher and more steeply bordered than the gently rounded lateral crest. The medial malleolus is expanded and separated from the body of the astragalus by a shallow, ovoid malleolar fossa. The talar neck and head are transversely broad and the articular surface for the navicular is slightly obliquely oriented. Ventrally, the concave facets for articulation with the calcaneus are large and deeply concave (figure 21*h*). The lateral and medial

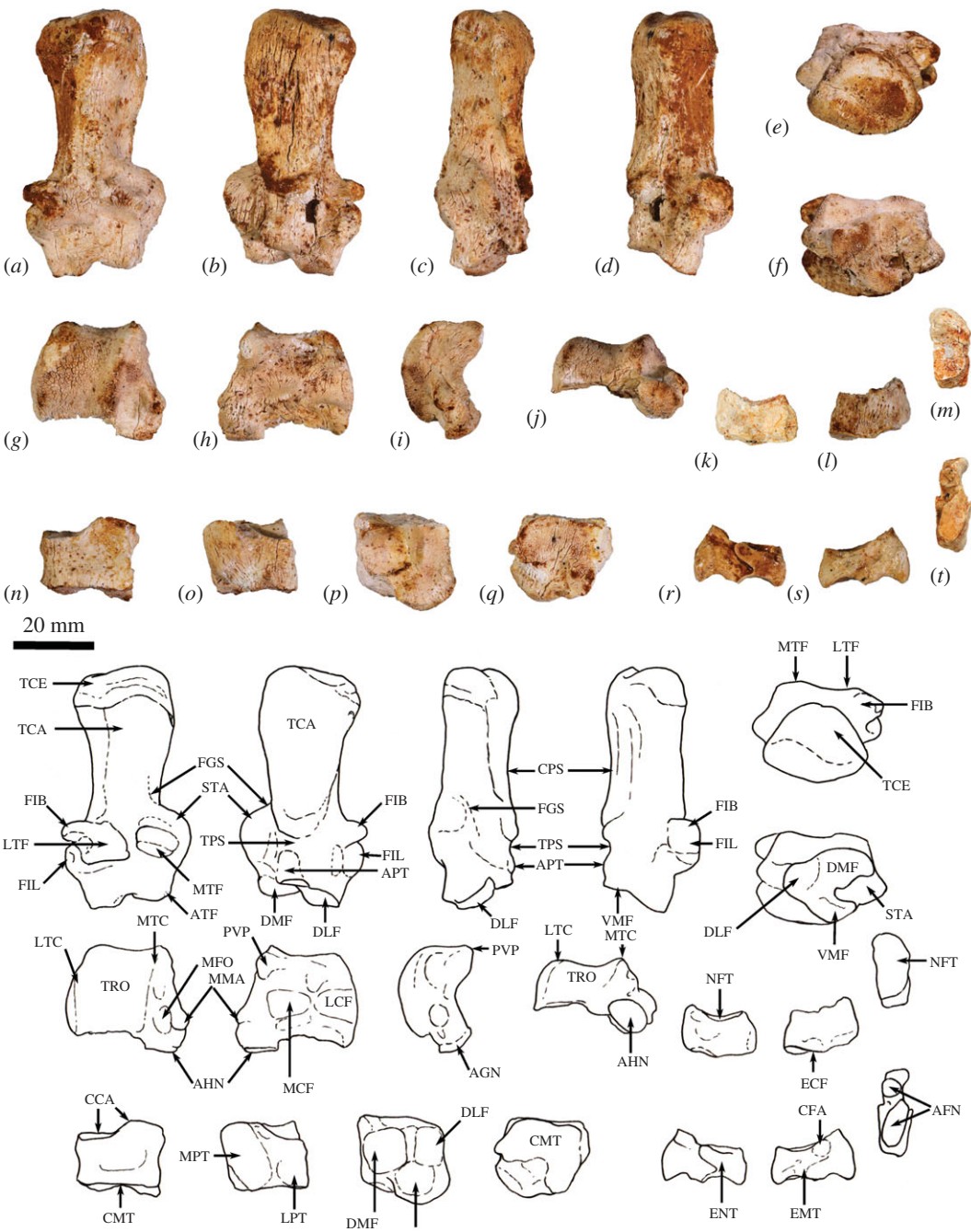

**Figure 21.** Tarsal elements of *Congruus kitcheneri*, WAM 02.7.12. (*a*) Calcaneum dorsal view; (*b*) calcaneum plantar view; (*c*) calcaneum medial view; (*d*) calcaneum lateral view; (*e*) calcanuem proximal view; (*f*) calcaneum distal view; (*g*) talus dorsal view; (*h*) talus plantar view; (*i*) talus medial view; (*j*) talus distal view; (*k*) navicular axial view; (*l*) navicular abaxial view; (*m*) navicular proximal view; (*n*) cuboid dorsal view; (*o*) cuboid ventral view; (*p*) cuboid proximal view; (*q*) cuboid distal view; (*r*) ectocuneiform axial view; (*s*) ectocuneiform abaxial view; (*t*) ectocuneiform proximal view. Scale bar equals 20 mm. Abbreviations: AFN, articular facets for navicular; AHN, head of talus (articular facet for navicular); APT, anterior plantar tuberosity; ATF, anteromedial facet for head of talus; CCA, calcaneal-cuboid articulation; CFA, articular facet for cuboid; CMT, cuboid facet for metatarsal IV; CPS, calcaneal plantar surface; DLF, dorsolateral facet of calcaneal-cuboid articulation; DMF, dorsomedial facet of calcaneal-cuboid articulation; ECF, facet for ectocuneiform; EMT, ectocuneiform facet for metatarsal IV; ENT, facet for entocuneiform; FGS, flexor groove of sustentaculum tali; FIB, fibular articular surface; FIL, tubercles for fibular ligaments; LCS, lateral calcaneal facet; LPT, lateral plantar tubercle; LTC, lateral trochlear crest; LTF, lateral facet for talus; MCF, medial calcaneal facet; MFO, malleolar fossa; MMA, medial malleolus; MTC, medial trochlear crest; MTF, medial facet for talus; NFT, navicular facet for talus; PVP, posteroventral process; STA, sustentaculum tali; TCA, tuber calcanei; TCE, epiphysis of tuber calcanei; TPS, transverse plantar sulcus; TRO, trochlea (talo-crural joint); VMF, ventromedial facet of calcaneal-cuboid articulation.

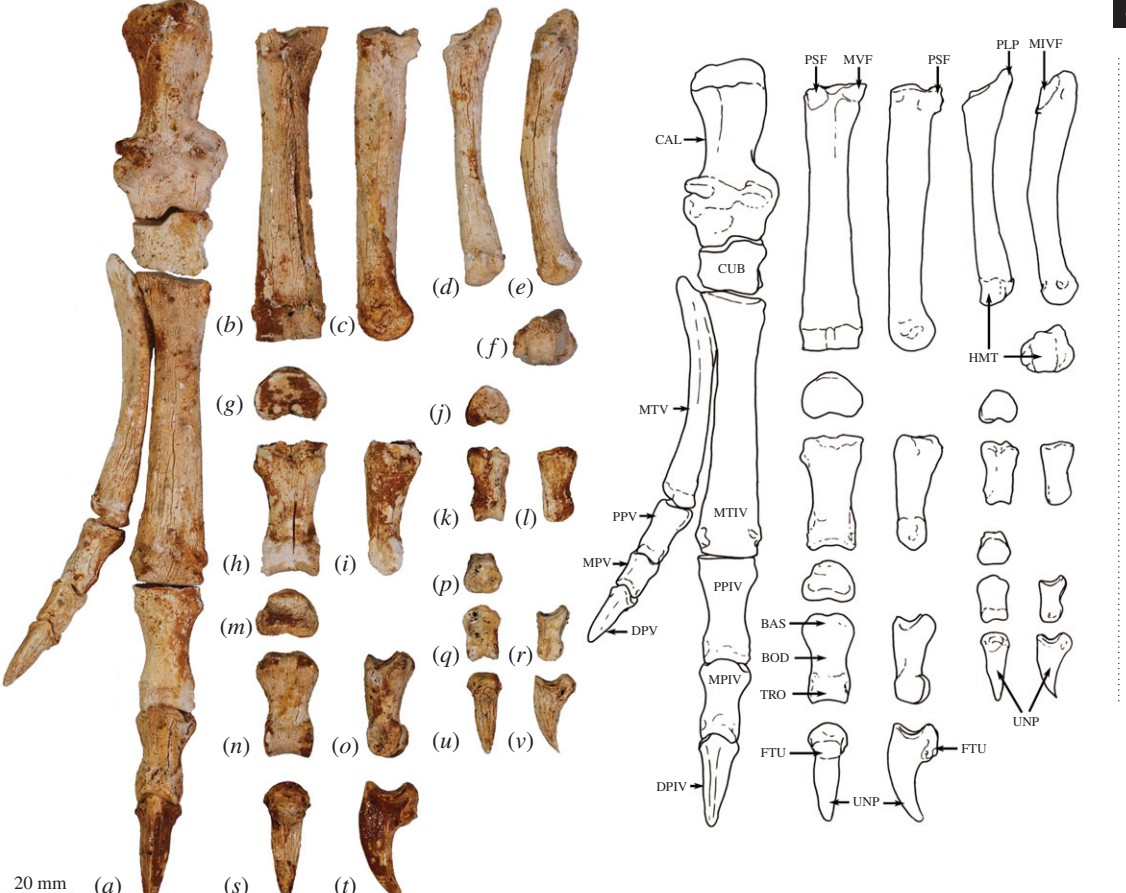

**Figure 22.** Pedal elements of *Congruus kitcheneri*, WAM 02.7.12. (*a*) Pes semi-articulated dorsal view; (*b*) metatarsal IV plantar view; (*c*) metatarsal IV medial view; (*d*) metatarsal V plantar view; (*e*) metatarsal V medial view; (*f*) metatarsal V distal view; (*g*) proximal phalanx IV proximal view; (*h*) proximal phalanx IV plantar view; I proximal phalanx IV medial view; (*j*) proximal phalanx V proximal view; (*k*) proximal phalanx V ventral view; (*l*) proximal phalanx V medial view; (*m*) middle phalanx IV proximal view; (*n*) middle phalanx IV ventral view; (*o*) middle phalanx IV medial view; (*p*) middle phalanx V proximal view; (*q*) middle phalanx V ventral view; (*r*) middle phalanx V medial view; (*s*) distal phalanx IV ventral view; (*t*) distal phalanx IV medial view; (*u*) distal phalanx V ventral view; (*v*) distal phalanx V medial view. Scale bar equals 20 mm. Abbreviations: BAS, base; BOD, body; CAL, calcaneum; CUB, cuboid; DP(IV-V), distal phalanges; FTU, flexor tubercle; HMT, head of metatarsal; MIVF, facet for metatarsal IV; MP(IV-V), middle phalanges; MTIV, metatarsal IV; MTV, metatarsal V; MVF, facet for metatarsal V; PLP, posterolateral process; PP(V-IV), proximal phalanges; PSF, facet for plantar sesamoid (proximal plantar process); TRO, trochlea; UNP, ungular process.

portions are separated by a narrow but distinct groove. The posteroventral process is moderately long and relatively robust (figure 21*i*), and in articulation, this continues the arc of the calcaneal sustentaculum tali. On the medial face, the fossa for the posterior tibiotalar ligament is relatively deep with a strongly marked dorsal margin.

The talus of *C. kitcheneri* is most similar to those of *T. billardierii* and *S. brachyurus* in overall morphology, being slightly wider than long, with a relatively square trochlea, subequal length of lateral and medial trochlear crests, stout proximoventral process, and broad, relatively open ventral facets for the talocalcaneal articulation. In *D. bennettianus* and the species of *Bohra*, the talus is much shorter craniocaudally, the trochlea wider, the talar head more obtusely oriented and the ventral talocalcaneal facets more smoothly confluent. In *M. fuliginosus*, *Os. rufus* and *Pe. xanthopus*, the talus is longer than it is wide and marked by long processes that extend caudoventrally from the dorsal margin of the high medial trochlear crest toward the outside of the very narrow sustentaculum tali. They are also characterized by a distinct craniomesial facet that articulates with the cuboid.

**Cuboid.** Viewed dorsally, the cuboid is roughly rectangular, though the caudal border is 'stepped' for the articulation with the calcaneus (figures 21*n* and 22*a*). The dorsal face is very shallowly concave. The three proximal facets for the calcaneus are clearly demarked: the dorsomedial facet is distinctly concave while the ventromedian facet is separated from the dorsomedial facet by a narrow non-articular groove

(figure 21*p*). The dorsolateral facet is caudoventrally oriented to meet the small, ovoid ventromedial facet. The medial surface of the cuboid is deeply concave below the border of the calcaneal facet. A deep groove running obliquely towards the proximomesial surface of the medial plantar tuberosity, which is massive, both in width and length, covering almost the entire plantar aspect of the bone (figure 21*o*). The oblique groove for the peroneal tendon is shallow and passes around the anterolateral aspect of the medial plantar tuberosity. The lateral plantar process, in contrast, is greatly reduced, and the longitudinal plantar sulcus passing around the cranial edge of the plantar tubercle is poorly delineated.

The cuboid of *C. kitcheneri* is most similar to that of *T. billardierii* in overall morphology, though absolutely much larger. The cuboid of *T. billardierii* is more dorsally concave, the craniomedial plantar process more distinct, and the facet for metatarsal IV covers a larger surface area. In *M. fuliginosus* and *Os. rufus*, the cuboid is dorsoventrally much deeper than in *C. kitcheneri* due to the very extended plantar tuberosity. The cuboid of *D. bennettianus* is distinct in being craniocaudally and dorsoventrally compressed, with very smoothed articular facets for the calcaneus.

**Navicular**. The navicular is dorsoventrally deep and transversely narrow (figure 21*k–l*). The proximal articular facet for the talar head is elongate and moderately concave. The distal articular area for the ectocuneiform is divided into a smaller ventral facet and an elongate dorsal facet; the latter blends smoothly with the small medial facet for the entocuneiform.

The articular facet for the talar head on the navicular in *C. kitcheneri* is less deeply curved than in *M. fuliginosus* and *Os. rufus*. The distal articular facet for the ectocuneiform is not divided in *M. fuliginosus* and *Os. rufus*, and the bone is shallower in the dorsoventral plane than it is in *C. kitcheneri.* By contrast, the navicular of *D. bennettianus* is wider than it is deep.

**Ectocuneiform**. The ectocuneiform is transversely narrow, but deep in the dorsoventral plane (figure 21*r–t*). The proximal articulation for the navicular is long and extends along the entire proximal length of the bone in two distinct facets corresponding with those of the navicular. There is a small distinct facet on the medial aspect of the entocuneiform. Laterally, there is a well-marked, elongate facet in the dorsal half that articulates with both the cuboid and metatarsal IV.

The ectocuneiform of *C. kitcheneri* appears similar to those of *M. fuliginosus* and *Os. rufus*, but its articular facet for the navicular is much longer, and the hooked plantar expansion is much shorter.

**Metatarsal IV**. Metatarsal IV is large and robust (figure 22*a–c*). The diaphysis is constricted proximally relative to the proximal articular surface, and then expands smoothly from halfway along towards the broad distal articular facet. The dorsal surface of the shaft is almost flat, being only slightly convex in the proximomedial portion. The proximal articular facet for the cuboid is inclined from the lateral to the medial side, and shallowly concave. The plantar tuberosity is small but deep with a small plantar sesamoid facet facing anteroventrally. The facets for metatarsal V are combined into a single, elongate ovoid facet that faces ventrolaterally on the expanded proximolateral tuberosity. The distal articular surface is broad and cylindrical in dorsal view. Distally, the medial side is slightly deeper than the lateral side. Ventrally, the median articular crest is small and narrow in comparison with the greatly expanded lateral and medial crests, which are aligned with the long axis of the bone and deeply flared ventrally. The proximal plantar crest is relatively low and broad.

Metatarsal IV of *C. kitcheneri* is shorter (37% relative to femur length) and more robust than in most extant ground-dwelling macropodine taxa. In *M. fuliginosus*, *N. eugenii*, *Os. rufus* and *Pe. xanthopus*, metatarsal IV is long (greater than 40% femur length; species of *Thylogale* and *Petrogale* 40–45%; *Macropus*, *Notamacropus* and *Osphranter* greater than 50%), straight and gracile, with very limited change in transverse width throughout the shaft in comparison with *C. kitcheneri*. In *S. brachyurus* and species of *Dendrolagus* the length of metatarsal IV relative to femur length is less than 37%. Metatarsal IV of *Pr. brehus* is more robust than that of *C. kitcheneri*. In *T. billardierii*, the distal shaft does broaden but the distal articular surface is proportionately not as wide. Relative to the combined length of the calcaneus and cuboid, the metatarsal IV of *C. kitcheneri* approaches the short relative length observed in *D. bennettianus,* though the shaft of the bone is less dorsoventrally compressed.

**Metatarsal V**. The shaft of metatarsal V is curved and accentuated by the strongly developed proximolateral process and the laterally expanded distal portion (figure 22*d–e*). A shallow lateral sulcus extends along the proximal half of the bone, producing a strong dorsal crest proximally. In lateral view, this crest is convex and, together with the concave ventral surface, gives the whole bone an arched profile. The proximal facet for the lateral tuberosity of the cuboid is transversely wide and triangular in outline, sweeping upwards with the extension of the proximolateral process. The facets for metatarsal IV are completely joined to form a broad, ovoid facet that reflects the morphology of the opposing facet and abuts the dorsal margin of the cuboid facet. The distal articular epiphysis is asymmetrical; the medial and median crests are large and roughly equal in size, while the lateral crest is greatly reduced.

Metatarsal V is short and relatively robust, and more laterally flexed than in *M. fuliginosus*, *N. eugenii* and *Os. rufus*, wherein it is long, slightly laterally flexed and, for the most part, very transversely narrow. In *Pr. brehus*, metatarsal V is more robust than in *C. kitcheneri*. The proximolateral process is larger in *C. kitcheneri* than in the ground-dwelling species, and bears closer resemblance to the enlarged morphology observed in species of *Bohra* and *Dendrolagus*.

**Pedal phalanges**. Proximal phalanx IV is robust and has a distinct fossa in the dorsal midline, immediately preceding the articular margin. Middle phalanx IV is 75% of the length of proximal phalanx IV. The proximal end is concave with a rounded dorsal margin. The distal articular surface extends onto the dorsal surface, and the fossae for the collateral ligaments are deep and circular. Distal phalanx IV is large, dorsoventrally deep, and strongly recurved with a tall, acute triangular cross-section. The flexor tubercle is large, deep and rounded. Proximal phalanx V is approximately half the size of proximal phalanx IV. The proximal end is asymmetrical, with the articular surface skewed to the axial side. The distal end is saddle-shaped. Middle phalanx V is short and robust, and the proximal articular margin has a distinct mid-dorsal peak. The distal articular facet is dorsoventrally deep and convex, with a strongly defined dorsal margin that extends proximally on the dorsal surface. Distal phalanx V is dorsoventrally very deep, triangular in section, and strongly recurved.

Proximal phalanx IV of *C. kitcheneri* is slightly more robust than in small ground-dwelling macropodines, similar to those of *M. fuliginosus* and *Os. rufus*, though not as broad as in *Pr. brehus*. The distal extremity is most similar in morphology to that of *T. billardierii* and is less ventrolaterally flared than in *M. fuliginosus* and *Os. rufus*. Middle phalanx IV of *C. kitcheneri* is longer than in the extant macropodine taxa examined (figure 16*b*), i.e. in *D. bennettianus* it is 64% of the length of the proximal phalanx IV, in *T. billardierii* 58%, and in *Os. rufus* 56%. In addition, middle phalanx IV of *C. kitcheneri* is characterized by a relatively expansive distal articular surface, and uniquely deep, circular fossae for the collateral ligaments. The mid-dorsal peak on the proximal articular margin of middle phalanx V is similar to that of *T. billardierii*, being more pronounced than in *M. fuliginosus* and *Os. rufus*, but less so than in *D. bennettianus*. The long, strongly curved distal phalanges, with their deep, round flexor tubercles, resemble those of *D. bennettianus*, but they are deeper. In *M. fuliginosus* and *Os. rufus*, the ungual process is shorter and blunter, and bears a broad, flat flexor tubercle. The ungual process is also not especially long or curved in *Petrogale*.

# 4. Discussion

## 4.1. Sexual dimorphism

The two specimens described here are hypothesized to represent a male (WAM 02.7.26) and a female (WAM 02.7.12). Although their craniodental dimensions are very similar (table 1), the muscular attachments in the forelimb of WAM 02.7.26, and the robustness of its femur, suggest a more heavily built individual. This pattern of dimorphism is consistent with that observed in many extant kangaroos and wallabies, in which males have exaggerated development of the forelimbs [39–41]. Among macropodines, the ratio of male to female body mass is approximately 1.6 : 1, while for the largest species this ratio increases to 1.9 : 1 in *M. giganteus* and up to 2.5 : 1 in *Os. rufus* [42]. As forelimb muscle mass is positively allometric in males, larger males have the most strongly developed muscles and thus the most extreme muscle attachment scars on the humerus [39]. However, dimorphism in relative humeral length [43] and forelimb length [41,44,45], which becomes increasingly exaggerated in male macropodids, does not appear to exhibit the same pattern of dimorphism in the specimens of *C. kitcheneri* described here, perhaps suggesting a selective constraint related to behavioural differences between large ground-dwelling versus semiarboreal macropodid taxa.

## 4.2. Functional morphology

We believe the skeletons described here support a hypothesis that *C. kitcheneri* was probably a semiarboreal browsing kangaroo.

**Skull**. Overall, the skull of *Congruus kitcheneri* is closest morphologically to that of the species of *Dorcopsis*, including the position of the eye orbit, shallowness of the zygomatic arch, relative position and straightness of the cheek-tooth row, and gracile nature of the dentary. This indicates very similarly proportioned and oriented m. temporalis, mm. masseter and m. zygomaticomandibularis, reflecting adaptation to a diet placing only modest masticatory demands on the craniodental system.

In concert with the thinly enamelled, narrow molar lophs, weakly developed anteroposterior molar crests and blade-like premolar, this suggests a browse-dominated diet perhaps somewhere between that of the modern species of *Dorcopsis* and *Thylogale*. By contrast, the large kangaroos of the genera *Protemnodon*, *Macropus* and *Osphranter* have a deeper zygomatic arch, larger masseteric process and shorter premolar relative to the molars, reflecting the greater masticatory demands of their diets, which include substantial proportions of grass.

*Congruus kitcheneri* is set apart from all other macropodine taxa by the unique morphology of the anteorbital region of the skull. The deep, narrow rostrum may reflect the application of primarily orthal forces during incisor biting. The incisor-bearing portion of the premaxilla is long and extends anteriorly well beyond the tip of the nasals. These attributes, along with the comparatively small occlusal area of the upper incisors, suggest highly selective browsing. The enlarged 'pocket' on the mesial surface of the premaxilla, which is connected to the front of the premaxilla by many tiny foramina, is unique to *C. kitcheneri*. The sulcus on the mesial surface of the premaxilla of *C. congruus* probably represents an antecedent condition, being similar to, and probably homologous with, the lateral sulcus of the species of *Osphranter*, which inhabit warm to hot climates. The lateral sulcus is highly vascular, lined with respiratory epithelia and putatively involved in thermoregulation [46]. The hypertrophy of this feature in *C. kitcheneri* may be a more extreme adaptation to heat shedding aided by the intense vascularization of the premaxilla, but we cannot rule out an additional sensory role associated with the detection and selection of browse items.

**Axial skeleton**. In articulation, the occipito-atlantic joint of *C. kitcheneri* has less flexibility for flexion and extension than in *M. fuliginosus* or *Os. rufus*. This characteristic in the latter species perhaps reflects greater flexibility for grazing at ground level, or might be related to flexion and extension movements of the axial skeleton during bounding locomotion. Similarities in the morphology of the atlas (C1) of *C. kitcheneri* and *Pe. xanthopus* suggest more a bracing function, or less flexibility in movement, that may correspond with the need to stabilize the head during vertical leaps. In contrast with the atlanto-occipital joint, the relatively small, square and narrow centra of the axis and cervical vertebrae 3–7 suggests greater mobility, as does the relatively long neck overall. A relatively long and mobile neck would support reaching and manoeuvring of the head towards browse in an arboreal setting. This may be important for such a large animal, which would be restricted to larger supporting branches than the smaller tree-kangaroos.

Interestingly, the increase in relative neck length is inversely proportional to lumbar spine length in *C. kitcheneri* (26.5%) and *Os. rufus* (32.5%). The thoracic length (irrespective of thoracic number) is similar in both groups (43–44%); while tree-kangaroos have a shorter neck (*D. bennettianus* ratios C 20%: T 46.5%: L 33.5%). This suggests reduced flexibility in the lumbar spine for the relatively large *C. kitcheneri*, where a more mobile neck and head might be more advantageous than a flexible trunk. This adaptation may be analogous to sloths, which have relatively long necks (and unusually have a very variable number of cervical vertebrae, e.g. *Bradypus* species 8–10; *Choloepus* species 5–8) that they use to browse while the body is suspended or supported among the branches [47]. Among ground-dwelling taxa (mammals, reptiles, dinosaurs), foraging is regarded as the driving force for neck length relative [48]. Browsing antelopes, such as gerenuk (*Litocranius walleri*), and dama gazelle (*Gazella dama*), have longer necks than grazing species in order to reach higher foliage [48,49]. Among primates, larger bodied browsing apes have proportionately longer necks than smaller (more agile) monkeys [50]. As such, it seems appropriate to consider the relatively long and flexible neck of *C. kitcheneri* as an adaptation for browsing.

*Congruus kitcheneri* is unusual in having 14 thoracic and five lumbar vertebrae. Macropodine and sthenurine kangaroos typically have 13 thoracic and six lumbar vertebrae [51]. A partially articulated *Protemnodon* skeleton from Lake Callabonna (South Australia) has 14 ribs reflecting 14 thoracic centra, and as in *C. kitcheneri* the thoracic centra are relatively small in length. The similarity between these two closely related taxa [1], is strongly suggestive of this being a synapomorphy for this lineage. While there is an additional thoracic vertebra, the relative total length of the thoracic region in *C. kitcheneri* is not longer than in those taxa with 13 thoracic vertebrae. No obvious direct relationship was found between the length of the vertebral column and the number of vertebrae in primates [52], and it has been suggested that patterning of vertebrae at transitional zones may be flexible [53]. Thus, the presence of an extra thoracic vertebra in the *Congruus/Protemnodon* lineage probably does not represent a very significant modification, though may reflect a trend for greater mobility through the trunk. Overall, the morphology of the thoracic and lumbar vertebrae (much smaller articular facets and spinous processes) does give an appearance of greater mobility, and less resistance to bending than those of *M. fuliginosus* and *Os. rufus*, which are highly specialized for cursorial bipedal locomotion [51].

The profile of the relative lengths of the caudal vertebrae is similar to that of *Os. rufus* [54], though the transverse processes and centrum shape (robustness) suggest a more flexible tail in *C. kitcheneri* than in the large bipedal hopping species. A more flexible tail suggests less adaptation to efficiency for bipedal hopping and pentapedal locomotion, but rather greater range of movement through multiaxial planes observed in rock-wallabies and tree-kangaroos where the tail is used as a counter-balance during locomotion in complex environments [55].

**Forelimb**. The forelimb elements reflect a heavily muscled, yet flexible limb structure, with large hands, long robust digits and massive recurved claws.

The morphology of the glenohumeral joint, including an elongate, ovoid glenoid cavity with robust supraglenoid tubercle and strongly developed coronoid process, together with low proximal crests of the humerus, suggests a much greater range of movement for protraction at the shoulder than in extant kangaroos and wallabies (*Macropus* spp., *Pe. xanthopus*, and *T. billardierii*). A greater range of motion in protraction in *C. kitcheneri* that would allow the arms to be raised above the head would enhance reach in three dimensions, which might be useful during browsing (as also interpreted for some sthenurines; [22]). Further, such enhanced mobility is also characteristic in climbing animals, including species of *Dendrolagus*, where reach and flexibility are important [11,56,57].

The humerus of *C. kitcheneri* is distinctive, with exaggerated crests and sulci for the attachment of pectoralis, deltoideus, latissimus dorsi and teres major muscles, the radius is moderately robust, especially distally, and the ulna is straight and moderately robust. While large muscular crests on the humerus and robust antebrachium could potentially be linked to digging, the medial epicondyle of the humerus and the olecranon of the ulna are not especially large as would be expected for a digging mammal [58–60]. Rather, the features of the long bones of the forelimb in *C. kitcheneri*, and particularly the humerus, correspond with adaptations for climbing. Species of *Dendrolagus* have the greatest forelimb muscle mass among marsupials [61] with a strong emphasis on muscles responsible for forelimb adduction, grasping and gripping in order to maintain contact with the substrate (m. pectoralis, m. latissimus dorsi, m. teres major, and m. subscapularis) [11]. Enlarged pectorals with improved mechanical advantage via a more distal insertion on the humerus, as evidenced by the relatively long pectoral ridge on the humerus, enable the animal to maintain contact with its substrate and provide propulsive force during vertical ascent. Indeed, the pectoral muscle group in *D. lumholtzi* is proportionately almost twice as large as the same muscle group in *N. eugenii* [11,62]. Marked insertion for the m. teres major/m. latissimus dorsi in *C. kitcheneri*, suggests powerful adduction and retraction of the humerus for climbing, as found in species of *Dendrolagus* and *Bohra*, in contrast with ground-dwelling macropodines.

The radius and ulna of *C. kitcheneri* are slightly shorter than the humerus, and relatively more robust than in large ground-dwelling kangaroos, and the radial tubercle for the insertion of the m. biceps brachii is more pronounced. The shorter radius and ulna would have the effect of improving the mechanical advantage of muscles acting both for limb adduction (as described above) and elbow flexion (via m. biceps brachii and m. brachialis) by reducing the length of the out-lever (load-arm) of the limb, while the more robust form of these bones would provide enhanced strength under weight-bearing situations. Both these adaptations would be advantageous for a large-bodied climbing animal and are consistent with the morphology of the forelimb in species of *Dendrolagus* [11] as well as large-bodied primates [63–65]. The enlarged radial tubercle in *C. kitcheneri*, in comparison with ground-dwelling forms, suggests a greater role of the m. biceps brachii, which would be required to resist elbow extension and facilitate elbow flexion through a range of rotated elbow postures under weight-bearing conditions during climbing.

The articular surfaces of the carpal bones have adaptations for weight-bearing together with a high degree of flexibility, particularly in flexion and extension of the carpus. The scapholunatum and hamatum together form a very high dome for articulation with radius, suggesting a wide range of movement, and with a large surface area to spread the compressive load from the robust bones of the forearm. The articulation between scapholunatum-capitatum is reminiscent of a hinge joint that would provide a greater degree of flexion and extension between the proximal and distal carpal rows than is typical of macropodines. The metacarpal bones are very robust with enlarged articular heads, which provides a large surface area to spread the forces acting through the metacarpophalangeal joints, as described previously in tree-kangaroos [11]. Adaptations for gripping by the hands and feet are the most characteristic adaptation of climbing animals [49,66]. The carpometacarpal joints at both the first, and especially the fifth digit rays of *C. kitcheneri* have the capacity for a large range of adduction, which would also facilitate highly modified grasping actions of the hand for enhanced contact with the substrate during climbing.

The long, massive digits with very large and strongly recurved ungual phalanges are, visually, the most remarkable feature of this species, and seem most likely to reflect an adaptation for climbing. While the claws of ground-dwelling macropodines are moderately long and relatively flat, the claws of arboreal marsupials are dorsoventrally deep, laterally compressed, and strongly recurved [11,67]. (This is in contrast with digging marsupials, which tend to have long, but transversely wider claws that are more spatulate than hooked in the form [68]. The flexor muscle insertions on the middle and distal phalanges are strongly marked, reflecting well-developed digital flexor muscles and thus strong flexion of the digits, and is strongly suggestive of adaptation for climbing.

**Hindlimb**. The hindlimb elements are macropodine in form and most like those of generalized, closed habitat species such as *T. billardierii*, *S. brachyurus* and *Dorcopsulus* spp., reflecting bounding locomotion through closed/dense habitats, in contrast with the much more gracile-limbed species of *Macropus*, *Notamacropus*, and *Osphranter*, which are adapted for locomotion through more open habitats [51]. The tarsus and pes, however, suggest adaptation towards climbing. The large, distally flared calcaneus and smoothing of the calcaneal-cuboid articulation would allow some rotation for eversion of the foot, as described in species of *Dendrolagus* [9,10,24], though to a lesser degree. Similarly, the shortened pes with short and robust metatarsals (relative to both calcanus and femur length) is reminiscent of tree-kangaroos, or at least intermediate between rock-wallabies (*Petrogale*) and tree-kangaroos (*Dendrolagus*). Another feature that appears convergent with tree-kangaroos is the medially positioned quadratus scar on femur, which suggests a strong emphasis on adduction in the hindlimb as would be advantageous during climbing [12].

Like the manus, the most distinctive features of the pes in *C. kitcheneri* are the long, deep and strongly recurved ungual processes of the distal pedal phalanges. Deeply curved claws on the feet of ground-dwelling animals are unusual, presumably as they would wear substantially in contact with the ground. While kangaroos may employ the claws on the pes as weapons for male–male interactions, the ungual elements tend to be straight rather than curved. However, in many arboreal animals, curved claws reflect adaptation for enhancing the contact between the animal and uneven substrates [49,69]. While the morphology of the pes, including the tarsal bones and the relative development of the metatarsals and phalanges, is not as modified as in species of *Dendrolagus* and *Bohra*, we believe that the large, curved claws only make sense as an indication of some capacity for tree-climbing in *C. kitcheneri*.

# 5. Conclusion

Basic skeletal descriptions and functional analyses are essential for illuminating the ecology of extinct marsupials and making informed judgements about what roles they played in late Cenozoic ecosystems. This paper presents the first published skeletal description of an extinct Australian macropodine that is not a tree-kangaroo. Unexpectedly, functional analysis of the skeleton reveals that *Congruus kitcheneri* was also adapted for climbing trees, although it was larger and not as specialized for arboreal living as the tree-kangaroos. Powerful adduction of the fore- and hindlimbs, together with grasping hands and strongly curved claws on both the manus and pes would have all facilitated enhanced capacity to maintain contact with uneven arboreal substrates. Adaptation of the axial skeleton for enhanced mobility, particularly a relatively long and mobile neck, suggests a unique strategy among macropodines for large-bodied arboreal browsing. These inferences align well with those drawn from the craniodental system, which point toward adaptations for selective browsing. Together, this suggests an ecological niche more akin to some large cercopithecine primates than to artiodactyls, with which kangaroos are often compared. Future detailed functional analyses of the axial skeleton, hands and feet will be of particular importance for understanding the very likely underestimated diversity and complexity of locomotory adaptations in kangaroos.

Data accessibility. All data are included in the manuscript, or have been uploaded as part of the electronic supplementary material.

Authors' contributions. N.M.W. and G.J.P. designed the study and co-wrote the paper. N.M.W. described and compared postcranial material and collected postcranial measurements; G.J.P. did the same for the skull and teeth. Both authors gave final approval for publication.

Competing interests. We have no competing interests.

Funding. Financial support for the Thylacoleo Caves research programme has been provided by the following organizations: Australia–Pacific Science Foundation (APSF17-09), Australian Research Council (FT130101728; DP190103636), Flinders University, Geological Survey of Western Australia, Murdoch University, National Geographic Society, Rio Tinto WA Future Fund, Sixty Minutes (Channel 9, Australia) and the Western Australian Museum.

Acknowledgements. We thank Mikael Siversson and Helen Ryan (Western Australian Museum) for the loan of fossil specimens. We are grateful to Peter Ackroyd, Paul Devine, Ken Boland, Ray Gibbons, George MacLucas, June MacLucas and Eve Taylor for bringing their Thylacoleo Caves discoveries to our attention, especially Paul and Ray for discovering the two specimens that are most central to this paper. We also thank Clay Bryce, Carey Burke, Grant Gully, Lindsay Hatcher, John Long, Ernie Lundelius, Dirk Megirian, Mark Norton and the many enthusiastic, hard-working Thylacoleo Caves volunteers from Flinders University and elsewhere for their field assistance. The Thylacoleo Caves lie beneath the traditional lands of the Mirning People, and we recognize their long and enduring connection to this region.

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
