## [Peer Review File · Royal Society Open Science]

Review History

RSOS-202216.R0 (Original submission)

Review form: Reviewer 1

Is the manuscript scientifically sound in its present form?

Yes

Are the interpretations and conclusions justified by the results?

Yes

Is the language acceptable?

Yes

Do you have any ethical concerns with this paper?

No

Have you any concerns about statistical analyses in this paper?

No

Recommendation?

Accept with minor revision (please list in comments)

Comments to the Author(s)

This study formally describes two skeletons and several crania belonging to a Pleistocene kangaroo, which here is reassigned to the genus *Congruus*. The authors present revised diagnoses based on cranial morphology, identify sexual dimorphism in the postcrania and argue for a semi-arboreal browsing habit for this species.

This is a tidy, well-structured, and exceptionally well-illustrated paper that comprehensively covers the anatomy of this extinct taxon, with appropriate comparisons made to contextualise the descriptions. I found the authors' assessment of sexual dimorphism in the limbs, given the marked morphological differences in these individuals of similar skull size, to be well supported, as are their other functional interpretations suggestive of a semi-arboreal habit.

The figures are clear, comprehensive and thorough. Along with the text descriptions and insightful functional morphology work, this paper will provide extremely valuable comparative datapoints to all future macropodoid anatomy work and is a strong contribution to the literature. The comparative measurements of other macropodoid species in the supplementary information are also to be commended as a useful open-access dataset that will be much appreciated by future workers.

The included data tables are clear and thorough, however regarding Tables 7, 8 & 9 (comparative postcranial measurements), visually comparing these columns of values is tricky in the current format. If the goal is to compare and contrast various proportions, a suggestion would be to create a simple additional figure presenting key sets of measures as bar or line plots so the relative differences and patterns pertinent to understanding *Congruus kitcheneri* can be more easily appreciated (perhaps something similar to Wells & Camens 2018). No additional analysis is necessary, but just presenting this information graphically rather than (or in addition to) columns of values would more convincingly reinforce the conclusions drawn. This might be particularly interesting for the forelimb (e.g. sexual dimorphism discussed on page 34) and the vertebral column (e.g. semi-arboreal *Congruus* vs terrestrial macropod vertebral regions discussed on page 35), especially if some of the taxa discussed in-text could be represented by data brought in from the supplementary material. I feel a couple of plots would visually support the quantitative arguments made in-text.

I have noted minor typographical and clarity-related comments in the attached PDF (Appendix A), and suggest a thorough copy-edit overall to catch any additional things I may have missed. Overall I do not have substantial recommendations for improvement, and pending the authors' consideration of these minor suggestions, I recommend this paper for publication in Royal Society Open Science.

Review form: Reviewer 2 (Alana Sharp)

Is the manuscript scientifically sound in its present form?

Yes

Are the interpretations and conclusions justified by the results?

Yes

Is the language acceptable?

Yes

Do you have any ethical concerns with this paper?

No

Have you any concerns about statistical analyses in this paper?

No

Recommendation?

Accept with minor revision (please list in comments)

Comments to the Author(s)

This is a very well written and interesting descriptive manuscript and reclassification of a fossil macropodine.

In terms of writing, I didn't find any typos or errors for correction, but I will admit I only skimmed the "Description and comparisons" section, focusing mainly on the elements that were more relevant to the interpretation of semiarborealism. The argument for a semiarboreal lifestyle is supported very well by the descriptions of the forelimbs and grasping digits. However, I would like to see more discussion and comparisons with rock-wallabies in the section of the hindlimbs and pes anatomy, because these have shorter and more flexible feet than other ground dwelling macropods and they weren't mentioned in the comparison (although they were mentioned for the caudal vertebrae). Could *C. kitcheneri* be moving around a rocky environment with uneven ground, using its hands and arms to grasp, stabilise and move between trees and rocky outcrops? If this is unlikely, perhaps due to other evidence of the habitat, then I think this can be discussed and justified a bit in the discussion.

Overall, however, I think this manuscript is very well written, and it highlights that macropods have more diverse locomotor adaptations.

Decision letter (RSOS-202216.R0)

Dear Dr Warburton

On behalf of the Editors, we are pleased to inform you that your Manuscript RSOS-202216 "The skeleton of *Congruus kitcheneri*, a semiarboreal kangaroo from the Pleistocene of southern Australia" has been accepted for publication in Royal Society Open Science subject to minor revision in accordance with the referees' reports. Please find the referees' comments along with any feedback from the Editors below my signature.

Please submit your revised manuscript and required files (see below) no later than 7 days from today's (ie 03-Feb-2021) date. Note: the ScholarOne system will 'lock' if submission of the revision

is attempted 7 or more days after the deadline. If you do not think you will be able to meet this deadline please contact the editorial office immediately.

on behalf of Professor Marcelo Sanchez (Associate Editor) and Kevin Padian (Subject Editor)
openscience@royalsociety.org

Editor comments:

Thanks for your submission. The AE and I agree with the reviewers and ask you to address their comments in your revision. Best wishes.

Reviewer comments to Author:

Reviewer: 1
Comments to the Author(s)

This study formally describes two skeletons and several crania belonging to a Pleistocene kangaroo, which here is reassigned to the genus *Congruus*. The authors present revised diagnoses based on cranial morphology, identify sexual dimorphism in the postcrania and argue for a semi-arboreal browsing habit for this species.

This is a tidy, well-structured, and exceptionally well-illustrated paper that comprehensively covers the anatomy of this extinct taxon, with appropriate comparisons made to contextualise the descriptions. I found the authors' assessment of sexual dimorphism in the limbs, given the marked morphological differences in these individuals of similar skull size, to be well supported, as are their other functional interpretations suggestive of a semi-arboreal habit.

The figures are clear, comprehensive and thorough. Along with the text descriptions and insightful functional morphology work, this paper will provide extremely valuable comparative datapoints to all future macropodoid anatomy work and is a strong contribution to the literature.

The comparative measurements of other macropodoid species in the supplementary information are also to be commended as a useful open-access dataset that will be much appreciated by future workers.

The included data tables are clear and thorough, however regarding Tables 7, 8 & 9 (comparative postcranial measurements), visually comparing these columns of values is tricky in the current format. If the goal is to compare and contrast various proportions, a suggestion would be to

create a simple additional figure presenting key sets of measures as bar or line plots so the relative differences and patterns pertinent to understanding *Congruus kitcheneri* can be more easily appreciated (perhaps something similar to Wells & Camens 2018). No additional analysis is necessary, but just presenting this information graphically rather than (or in addition to) columns of values would more convincingly reinforce the conclusions drawn. This might be particularly interesting for the forelimb (e.g. sexual dimorphism discussed on page 34) and the vertebral column (e.g. semi-arboreal *Congruus* vs terrestrial macropod vertebral regions discussed on page 35), especially if some of the taxa discussed in-text could be represented by data brought in from the supplementary material. I feel a couple of plots would visually support the quantitative arguments made in-text.

I have noted minor typographical and clarity-related comments in the attached PDF, and suggest a thorough copy-edit overall to catch any additional things I may have missed. Overall I do not have substantial recommendations for improvement, and pending the authors' consideration of these minor suggestions, I recommend this paper for publication in Royal Society Open Science.

Reviewer: 2

Comments to the Author(s)

This is a very well written and interesting descriptive manuscript and reclassification of a fossil macropodine.

In terms of writing, I didn't find any typos or errors for correction, but I will admit I only skimmed the "Description and comparisons" section, focusing mainly on the elements that were more relevant to the interpretation of semiarborealism. The argument for a semiarboreal lifestyle is supported very well by the descriptions of the forelimbs and grasping digits. However, I would like to see more discussion and comparisons with rock-wallabies in the section of the hindlimbs and pes anatomy, because these have shorter and more flexible feet than other ground dwelling macropods and they weren't mentioned in the comparison (although they were mentioned for the caudal vertebrae). Could *C. kitcheneri* be moving around a rocky environment with uneven ground, using it's hands and arms to grasp, stabilise and move between trees and rocky outcrops? If this is unlikely, perhaps due to other evidence of the habitat, then I think this can be discussed and justified a bit in the discussion.

Overall, however, I think this manuscript is very well written, and it highlights that macropods have more diverse locomotor adaptations.

===PREPARING YOUR MANUSCRIPT===

one version identifying all the changes that have been made (for instance, in coloured highlight, in bold text, or tracked changes);
a 'clean' version of the new manuscript that incorporates the changes made, but does not highlight them. This version will be used for typesetting.

===PREPARING YOUR REVISION IN SCHOLARONE===

- Any electronic supplementary material (ESM).
- If you are requesting a discretionary waiver for the article processing charge, the waiver form must be included at this step.
- If you are providing image files for potential cover images, please upload these at this step, and inform the editorial office you have done so. You must hold the copyright to any image provided.
- A copy of your point-by-point response to referees and Editors. This will expedite the preparation of your proof.

- Ensure that your data access statement meets the requirements at <https://royalsociety.org/journals/authors/author-guidelines/#data>. You should ensure that you cite the dataset in your reference list. If you have deposited data etc in the Dryad repository, please only include the 'For publication' link at this stage. You should remove the 'For review' link.
- If you are requesting an article processing charge waiver, you must select the relevant waiver option (if requesting a discretionary waiver, the form should have been uploaded at Step 3 'File upload' above).
- If you have uploaded ESM files, please ensure you follow the guidance at <https://royalsociety.org/journals/authors/author-guidelines/#supplementary-material> to include a suitable title and informative caption. An example of appropriate titling and captioning may be found at https://figshare.com/articles/Table_S2_from_Is_there_a_trade-off_between_peak_performance_and_performance_breadth_across_temperatures_for_aerobic_scope_in_teleost_fishes_/3843624.

Author's Response to Decision Letter for (RSOS-202216.R0)

See Appendix B.

Decision letter (RSOS-202216.R1)

Dear Dr Warburton,

It is a pleasure to accept your manuscript entitled "The skeleton of *Congruus kitcheneri*, a semiarborescent kangaroo from the Pleistocene of southern Australia" in its current form for publication in Royal Society Open Science. The comments of the reviewer(s) who reviewed your manuscript are included at the foot of this letter.

Kind regards,

Anita Kristiansen
Editorial Coordinator

on behalf of Professor Marcelo Sanchez (Associate Editor) and Kevin Padian (Subject Editor)
openscience@royalsociety.org

Appendix A**ROYAL SOCIETY
OPEN SCIENCE****The skeleton of *Congruus kitcheneri*, a semiarboreal kangaroo from the Pleistocene of southern Australia**

Journal:	Royal Society Open Science
Manuscript ID	RSOS-202216
Article Type:	Research
Date Submitted by the Author:	05-Dec-2020
Complete List of Authors:	Warburton, Natalie; Murdoch University, Medical, Molecular and Forensic Science; Murdoch University, Centre for Climate Impacted Terrestrial Ecosystems, Harry Butler Research Institute; Western Australian Museum, Earth and Planetary Science Prideaux, Gavin; Flinders University, College of Science & Engineering,
Subject:	palaeontology < BIOLOGY, ecology < BIOLOGY, taxonomy and systematics < BIOLOGY
Keywords:	Australia, mammal, taxonomy, biogeography, Pliocene, locomotion
Subject Category:	Organismal and Evolutionary Biology

Author-supplied statements

Relevant information will appear here if provided.

Ethics

Does your article include research that required ethical approval or permits?:

This article does not present research with ethical considerations

Statement (if applicable):

CUST_IF_YES_ETHICS :No data available.

Data

It is a condition of publication that data, code and materials supporting your paper are made publicly available. Does your paper present new data?:

Yes

Statement (if applicable):

All measurement data used within this study is presented in the manuscript or with the supplementary data. No code was used.

Conflict of interest

I/We declare we have no competing interests

Statement (if applicable):

CUST_STATE_CONFLICT :No data available.

Authors' contributions

This paper has multiple authors and our individual contributions were as below

Statement (if applicable):

NMW and GJP designed the study and co-wrote the paper. NMW described and compared postcranial material and collected postcranial measurements; GJP did the same for the skull and teeth. Both authors gave final approval for publication.

The skeleton of *Congruus kitcheneri*, a semiarboreal kangaroo from the Pleistocene of southern Australia

Natalie M. Warburton^{abc*}, Gavin J. Prideaux^d

^aMedical, Molecular and Forensic Sciences, College of Science, Health, Engineering and Education, Murdoch University, Murdoch, Western Australia 6150, Australia

^bCentre for Climate-Impacted Terrestrial Ecosystems, Harry Butler Research Institute, Murdoch University

^cDepartment of Earth and Planetary Science, Western Australian Museum, Kew Street, Welshpool WA.

^dCollege of Science & Engineering, Flinders University, South Australia 5042, Australia

*Corresponding author. Email: n.warburton@murdoch.edu.au

Abstract

The macropodine kangaroo, *Wallabia kitcheneri*, was first described in 1989 from a Pleistocene deposit within Mammoth Cave, southwestern Australia, on the basis of a few partial dentaries and maxilla fragments. Here we recognise *W. kitcheneri* within the Pleistocene assemblages of the Thylacoleo Caves, south-central Australia, where it is represented by several cranial specimens and two near-complete skeletons, a probable male and female. We reallocate this species to the hitherto monotypic genus *Congruus*. *Congruus kitcheneri* differs from all other macropodid species by having a highly unusual pocket within the wall of the nasal cavity. It is distinguished from *C. congruus* by having a longer, narrower rostrum, a taller occiput and a deeper jugal. *Congruus* is closest to *Protemnodon* in overall cranial morphology, but is smaller and less robust. In most postcranial attributes, *C. kitcheneri* resembles *Protemnodon brehus*, including general limb robustness and the atypical ratio of 14 thoracic to 5 lumbar vertebrae. It is distinguished by the high mobility of its glenohumeral joints, the development of muscle attachment sites for strong adduction and mobility of the forelimb, and large, robust manual and pedal digits with strongly recurved distal phalanges. These adaptations resemble those of tree-kangaroos more than ground-dwelling macropodines. We interpret to imply that *C. kitcheneri* was semiarboreal, with a propensity to climb and move slowly through trees. This is the first evidence for the secondary adoption of a climbing habit within crown macropodines.

KEYWORDS: Australia – mammal – taxonomy – biogeography – Pliocene – Pleistocene – locomotion

1. Introduction

Macropodids (kangaroos and relatives) descended from arboreal possum-like ancestors during the Paleogene before becoming the main ground-dwelling mammalian herbivores of the Australian continent over the past 20 million years (Myr) (Prideaux and Warburton 2010). There are >60 extant species, including bettongs, wallabies and kangaroos, and many more extinct taxa, including giant short-faced kangaroos. As in ungulates, smaller extant macropodids are mainly solitary fungivores or browsers (dicot consumers), while larger species tend to be gregarious grazers (grass consumers) (Arman and Prideaux 2015). The greatest diversity within the group is expressed by the subfamily Macropodinae, most genera of which likely originated during the late Miocene into very early Pliocene (Couzens and Prideaux 2018; Celik, Cascini *et al.* 2019).

Eleven of the 12 extant macropodine genera are characterised by species that are principally ground dwelling. They employ a bipedal hopping mode of locomotion when moving at speed, and a ‘pentapedal’ mode, involving use of the tail as a ‘fifth limb’ when moving slowly (Dawson, Warburton *et al.* 2015). These locomotory styles are facilitated by striking adaptations in physiology and skeletal anatomy, including elongation of the hindlimbs and feet, and reconfiguration of the ankle joint to minimise lateral and rotational movement during hopping (Prideaux and Warburton 2010). The one extant genus that deviates from this pattern is *Dendrolagus* (Müller 1840), which contains the tree-kangaroos of New Guinea and extreme northeastern Australia. They descended from ground-dwelling ancestors by ascending into the trees. Tree-kangaroos can hop bipedally as well as move their hindlimbs alternately (Flannery, Martin *et al.* 1996). Their arboreal adeptness is reflected in a range of skeletal modifications manifested also within the larger, extinct species of the genus *Bohra* Flannery & Szalay, 1982, which are known from the Pliocene and Pleistocene (Prideaux and Warburton 2008; Prideaux and Warburton 2009; Prideaux and Warburton 2010; Warburton, Harvey *et al.* 2011; Warburton, Yakovlev *et al.* 2012). 
[revised manuscript text omitted]
 (Warburton, Bateman *et al.* 2013). However, dimorphism in relative humeral length (Richards,
Grueter *et al.* 2015) and forelimb length (Poole, Carpenter *et al.* 1982a; Poole, Carpenter *et al.* 1982b;
Miller, Eldridge *et al.* 2010), which becomes increasingly exaggerated in male macropodids, does not appear
to exhibit the same pattern of dimorphism in the specimens of *C. kitcheneri* described here, perhaps
suggesting a selective constraint related to behavioural differences between large ground-dwelling versus
semi-arboreal macropodid taxa.

**4.2. Functional morphology**

We believe the skeletons described here support a hypothesis that *C. kitcheneri* was likely a semiarboreal
browsing kangaroo.

**Skull:**

Overall, the skull of *Congruus kitcheneri* is closest morphologically to that of the species of *Dorcopsis*,
including the position of the eye orbit, shallowness of the zygomatic arch, relative position and straightness
of the cheek-tooth row, and gracile nature of the dentary. This indicates very similarly proportioned and
oriented m. temporalis, mm. masseter and m. zygomaticomandibularis, reflecting adaptation to a diet placing
only modest masticatory demands on the craniodental system. In concert with the thinly-enamelled, narrow
molar lophs, weakly-developed anteroposterior molar crests, blade-like premolar, this suggests a browse-
dominated diet perhaps somewhere between that of the modern species of *Dorcopsis* and *Thylogale*. By
contrast, the large kangaroos of the genus *Protemnodon*, *Macropus* and *Osphranter* have a deeper zygomatic
arch, larger masseteric process and shorter premolar relative to the molars, reflecting the greater masticatory
demands of their diets, which includes substantial proportions of grass.

Congruus kitcheneri is set apart from all other macropodine taxa by the unique morphology of the anteorbital region of the skull. The deep, narrow rostrum may reflect the application of primarily orthal forces during incisor biting. The incisor-bearing portion of the premaxilla is long and extends anteriorly well beyond the tip of the nasals. These attributes, along with the comparatively small occlusal area of the upper incisors, suggest highly selective browsing. The enlarged ‘pocket’ on the mesial surface of the premaxilla, which is connected to the front of the premaxilla by many tiny foramina, is unique to *C. kitcheneri*. The sulcus on the mesial surface of the premaxilla of *C. congruus* probably represents an antecedent condition, being similar to and likely homologous with the lateral sulcus of the species of *Osphranter*, which inhabit warm to hot climates. The lateral sulcus is highly vascular, lined with respiratory epithelia and putatively involved in thermoregulation (Nelson, Warburton *et al.* 2017). The hypertrophy of this feature in *C. kitcheneri* may be a more extreme adaptation to heat shedding aided by the intense vascularisation of the premaxilla, but we cannot rule out an additional sensory role associated with detection and selection of browse items.

Axial skeleton:

In articulation, the occipito-atlantic joint of *C. kitcheneri* has less flexibility for flexion and extension than in *M. fuliginosus* or *O. rufus*. This characteristic in the latter species perhaps reflects greater flexibility for grazing at ground level, or might be related to flexion and extension movements of the axial skeleton during bounding locomotion. Similarities in the morphology of the atlas (C1) of *C. kitcheneri* and *P. xanthopus* suggest more a bracing function, or less flexibility in movement, that may correspond with the need to stabilize the head during vertical leaps. In contrast to the atlanto-occipital joint, the relatively small, square and narrow centra of the axis and cervical vertebrae 3-7, suggests greater mobility, as does the relatively long neck overall. A relatively long and mobile neck would support reaching and maneuvering of the head towards browse in an arboreal setting. This may be important for such a large animal, which would be restricted to relatively larger supporting branches than the smaller tree-kangaroos.

Interestingly, the increase in proportionate neck length is inversely proportional with lumbar spine relative length in *C. kitcheneri* (26.5%) and *O. rufus* (32.5%), while the thoracic length (irrespective of thoracic number) is similar in both groups (43-44%); while tree-kangaroos have a relatively shorter neck (*D. bennettianus* ratios C 20%: T 46.5%: L 33.5%). This suggests reduced flexibility in the lumbar spine for the relatively large *C. kitcheneri*, a more mobile neck and head might be more advantageous than a flexible trunk. This adaptation may be analogous to sloths, which have relatively long necks (and unusually have a very variable number of cervical vertebrae, e.g. *Bradypus* species C8-10; *Choloepus* species C5-8) that they use to browse while the body is suspended or supported amongst the branches (Buchholtz and Stepien 2009). Among ground-dwelling taxa (mammals, reptiles, dinosaurs), foraging is regarded as the driving force for neck length (Wilkinson and Ruxton 2012). Browsing antelopes, such as Gerenuk (*Litocranius walleri*), and Dama gazelle (*Gazella dama*), have relatively longer necks than grazing species in order to reach higher foliage (Coombs 1983; Wilkinson and Ruxton 2012), and among primates, larger bodied apes have proportionately longer necks than smaller (more agile) monkeys (Schultz 1938). As such, it seems appropriate to consider the relatively long and flexible neck of *C. kitcheneri* as an adaptation for browsing.

Congruus kitcheneri is unusual in having fourteen thoracic and five lumbar vertebrae. Macropodine and sthenurine marsupials typically have thirteen thoracic and six lumbar vertebrae (Janis, Buttrill *et al.* 2014). A partially articulated *in situ* preparation of a *Protemnodon* skeleton from Lake Callabonna has fourteen ribs reflecting fourteen thoracic centra, and as in *C. kitcheneri* the thoracic centra are relatively small in length. The similarity between these two closely taxa (Prideaux and Warburton 2010), is strongly suggestive of this being a derived feature in this lineage. While there is an addition thoracic vertebra, the relative total length

of the thoracic region in *C. kitcheneri* is not longer than those with thirteen thoracic vertebrae. No obvious
direct relationship was found between the length of the vertebral column and the number of vertebrae in
primates (Majoral, Berge *et al.* 1997), and it has been suggested that patterning of vertebrae at transitional
zones may be flexible (Hautier, Weisbecker *et al.* 2010). Thus the presence of an extra thoracic vertebrae in
the *Congruus/Protomnodon* lineage likely does not represent a very significant modification, though may
reflect a trend for greater mobility through the trunk. Overall, the morphology of the thoracic and lumbar
vertebrae (relatively much smaller articular facets and spinous processes) does give an appearance of greater
mobility, and less resistance to bending than those of *M. fuliginosus* and *O. rufus*, which are higher
specialised for cursorial bipedal locomotion (Janis, Buttrill *et al.* 2014).

[revised manuscript text omitted]

Data accessibility. All data are included in the manuscript, appendices, or have been uploaded as part of the supplementary material.

Competing interests. We have no competing interests.

Authors' contributions. NMW and GJP designed the study and co-wrote the paper. NMW described and compared postcranial material and collected postcranial measurements; GJP did the same for the skull and teeth. Both authors gave final approval for publication.

Funding. Financial support for the Thylacoleo Caves research program has been provided by the following organisations: Australia–Pacific Science Foundation (APSF17-09), Australian Research Council (FT130101728; DP190103636), Flinders University, Geological Survey of Western Australia, Murdoch

University, National Geographic Society, Rio Tinto WA Future Fund, Sixty Minutes (Channel 9, Australia) and the Western Australian Museum.

Acknowledgements. We thank Mikael Siverson and Helen Ryan (Western Australian Museum) for the loan of fossil specimens. We are grateful to Peter Ackroyd, Paul Devine, Ken Boland, Ray Gibbons, George MacLucas, June MacLucas and Eve Taylor for bringing their Thylacoleo Caves discoveries to our attention, especially Paul and Ray for discovering the two specimens that are most central to this paper. We also thank Clay Bryce, Carey Burke, Grant Gully, Lindsay Hatcher, John Long, Ernie Lundelius, Dirk Megirian, Mark Norton and the many enthusiastic, hard-working Thylacoleo Caves volunteers from Flinders University and elsewhere for their field assistance.

References

- Archer, M., Crawford, I.M., and Merrilees, D. (1980) Incision, breakage and charring, some probably man-made, in fossil bones from Mammoth Cave, Western Australia. *Alcheringa* **4**(115-131).
- Argot, C. (2001) Functional-adaptive anatomy of the forelimb in the Didelphidae, and the paleobiology of the Paleocene marsupials *Mayulestes ferox* and *Pucadelphys andinus*. *Journal of Morphology* **247**, 51-79.
- Arman, S.D., and Prideaux, G.J. (2015) Dietary classification of extant kangaroos and their relatives (Marsupialia: Macropodoidea). *Austral Ecology* **40**, 909-922.
- Ayliffe, L.K., Prideaux, G.J., Bird, M.I., Grün, R., Roberts, R.G., Gully, G.A., Jones, R., Fifield, L.K., and Cresswell, R.G. (2008) Age constraints on Pleistocene megafauna at Tight Entrance Cave in southwestern Australia. *Quaternary Science Reviews* **27**, 1784-1788.
- Black, K.H., Camens, A.B., Archer, M., and Hand, S.J. (2012) Herds overhead: *Nimbadon lavarackorum* (Diprotodontidae), heavyweight marsupial herbivores in the Miocene forests of Australia. *PloS one* **7**(11), e48213.
- Bloch, J.I., and Boyer, D.M. (2002) Grasping primate origins. *Science* **298**, 1606-1611.
- Buchholtz, E.A., and Stepien, C.C. (2009) Anatomical transformation in mammals: developmental origin of aberrant cervical anatomy in tree sloths. *Evolution & Development* **11**(1), 69-79.
- Cave, A.J.E. (1975) The morphology of the mammalian cervical pleurapophysis. *Journal of Zoology* **177**, 377-393.
- Celik, M., Cascini, M., Haouchar, D., Van Der Burg, C., Dodt, W., Evans, A.R., Prentis, P., Bunce, M., Fruciano, C., and Phillips, M.J. (2019) A molecular and morphometric assessment of the systematics of the *Macropus* complex clarifies the tempo and mode of kangaroo evolution. *Zoological Journal of the Linnean Society* **186**, 793-812.
- Coombs, M.C. (1983) Large mammalian clawed herbivores: a comparative study. *Transactions of the American Philosophical Society* **73**, 1-96.
- Couzens, A.M.C., and Prideaux, G.J. (2018) Rapid Pliocene adaptive radiation of modern kangaroos. *Science* **362**(6410), 72-75.

Dawson, L. (2006) An ecophysiological approach to the extinction of large marsupial herbivores in middle
and late Pleistocene Australia. *Alcheringa: An Australasian Journal of Palaeontology* **30**(S1), 89-114.

Dawson, R., Warburton, N.M., Richards, H., and Milne, N. (2015) Walking on five legs: investigating tail
use during slow gait in kangaroos and wallabies. *Australian Journal of Zoology* **63**, 192-200.

Dawson, R.S. (2015) Morphological correlates of pentapedal locomotion in kangaroos and wallabies
(Family: Macropodidae). The University of Western Australia,

Figueirido, B., and Janis, C.M. (2011) The predatory behaviour of the thylacine: Tasmanian tiger or
marsupial wolf? *Biology Letters* **7**, 937-940.

Flannery, T., Martin, R.W., and Szalay, A. (1996) 'Tree kangaroos : a curious natural history.' (Reed:
Chatswood, N.S.W.)

Flannery, T.F. (1989) A new species of Wallabia (Macropodinae: Marsupialia) from Pleistocene deposits in
Mammoth Cave, southwestern Western Australia. *Records of the Western Australian Museum* **14**, 299-307.

Flannery, T.F., and Szalay, F. (1982) *Bohra paulae*, a new giant fossil tree kangaroo (Marsupialia:
Macropodidae) from New South Wales, Australia. *Australian Mammalogy* **5**, 83-94.

Fleagle, J.G. (1999) 'Primate evolution and adaptations.' Second edn. (Academic Press: San Diego, USA)
596

Flower, W.H. (1867) On the development and succession of teeth in the Marsupialia. *Philosophical*
*Transactions of the Royal Society* **157**, 631-641.

Glauert, L. (1910) The Mammoth Cave. *Records of the Western Australian Museum and Art Gallery* **1**, 11-
36.

Glauert, L. (1914) The Mammoth Cave (continued). *Records of the Western Australian Museum and Art*
*Gallery* **1**, 244-251.

Grand, T.I. (1990) Body composition and the evolution of the Macropodidae (*Potorous*, *Dendrolagus* and
*Macropus*). *Anatomy and Embryology* **182**, 85-92.

Gray, J.E. (1821) On the natural arrangement of vertebrate animals. *London Medical Repository* **15**, 296-
310.

Harvey, K.J., and Warburton, N.M. (2010) Forelimb musculature of kangaroos with particular emphasis on
the tammar wallaby *Macropus eugenii* (Desmarest, 1817). *Australian Mammalogy* **32**, 1-9.

Hautier, L., Weisbecker, V., Sánchez-Villagra, M.R., Goswami, A., and Asher, R.J. (2010) Skeletal
development in sloths and the evolution of mammalian vertebral patterning. *Proceedings of the National*
*Academy of Sciences* **107**, 18903-18908.

Hildebrand, M., and Goslow, G.E.J. (2001) 'Analysis of Vertebrate Structure.' (John Wiley & Sons, Inc.:
USA)

Hopkins, S.S., and Davis, E.B. (2009) Quantitative morphological proxies for fossoriality in small
mammals. *Journal of Mammalian Evolution* **90**, 1449-1460.

Janis, C.M., Buttrill, K., and Figueirido, B. (2014) Locomotion in extinct giant kangaroos: where
sthenurines hop-less monsters? *PloS One* **9**, e109888.
- Janis, C.M., and Figueirido, B. (2014) Forelimb anatomy and the discrimination of the predatory behavior of
carnivorous mammals: The thylacine as a case study. *Journal of Morphology* **275**, 1321-1338.
- Jarman, P.J. (1989) Sexual dimorphism in Macropodoidea. In 'Kangaroos, Wallabies and Rat-kangaroos.'
(Eds. G Grigg, J Jarman and ID Hume) pp. 433-447. (Surrey Beatty and Sons Pty Ltd: New South Wales Pty
Ltd)
- Jarman, P.J. (1991) Social behaviour and organization in the Macropodoidea. In 'Advances in the study of
behaviour. Vol. 20.' (Eds. HJ Brockmann, PJB Slater, CT Snowdon, TJ Roper, M Naguib and KE Wynne-
Edwards) pp. 1-50. (Academic Press: New York)
- Luckett, W.P. (1993) An ontogenetic assessment of dental homologies in therian mammals. In 'Mammal
Phylogeny.' (Eds. FS Szalay, MJ Novacek and MC McKenna) pp. 182–204. (Springer-Verlag: New York)
- Majoral, M., Berge, C., Casinos, A., and Jouffroy, F.-K. (1997) The length of the vertebral column of
primates: an allometric study. *Folia Primatologica* **68**, 57-76.
- Marzke, M.W., and Marzke, R.F. (2000) Evolution and the human hand: approaches to acquiring, analysing
and interpreting the anatomical evidence. *Journal of Anatomy* **197**, 121-140.
- McNamara, J.A. (1994) A new fossil wallaby (Marsupialia: Macropodidae) from the south east of South
Australia. *Records of the South Aust Museum* **27**, 111-115.
- Michilsens, F., Vereecke, E.E., D'Aout, K., and Aerts, P. (2009) Functional anatomy of the gibbon forelimb:
adaptations to a brachiating lifestyle. *J. Anat.* **215**, 335-354.
- Miller, E.J., Eldridge, M.D.B., Cooper, D.W., and Herbert, C.A. (2010) Dominance, body size and internal
relatedness influence male reproductive success in eastern grey kangaroos (*Macropus giganteus*).
*Reproduction Fertility and Development* **22**(3), 539-549.
- Müller, S. (1840). In 'Verhandelingen over de natuurlijke Geschiedenis der Nederlandsche Overzee-
sche Bezittingen door de Leden der Natuurkundige Commissie in Indie en andere Schrijvers. Zoologie'. (Ed. CJ
Temminck). (In commissie bij. S. en J. Luchtmans en C.C. van der Hoek: Leiden)
- Murray, P.F. (1995) The postcranial skeleton of the Miocene kangaroo, *Hadronomas puckeridgei* Woodburne
(Marsupialia: Macropodidae). *Alcheringa* **19**, 119-170.
- Nelson, D.P., Warburton, N.M., and Prideaux, G.J. (2017) The anterior nasal region in the Red Kangaroo
(*Macropus rufus*) suggests adaptation for thermoregulation and water conservation. *Journal of Zoology* **303**,
301-310.
- Owen, R. (1866) 'On the Anatomy of Vertebrates: Birds and mammals.' (Longmans, Green and Co.)
- Oxnard, C.E. (1967) The functional morphology of the primate shoulder as revealed by comparative
anatomical, osteometric and discriminant function techniques *American Journal of Physical Anthropology*
**26**, 219-240.

- Poole, W., Carpenter, S., and Wood, J. (1982a) Growth of Grey Kangaroos and the Reliability of Age Determination from Body Measurements II.* The Western Grey Kangaroos, *Macropus fuliginosus fuliginosus*, *M. f. melanops* and *M. f. ocydromus*. *Wildlife Research* **9**, 203-212.
- Poole, W.E., Carpenter, S.M., and Wood, J.T. (1982b) Growth of grey kangaroos and the reliability of age determination from body measurements I. The eastern grey kangaroo, *Macropus giganteus*. *Australian Wildlife Research* **9**, 9-20.
- Prideaux, G.J. (2004) Systematics and evolution of the sthenurine kangaroos. *University of California Publications in Geological Sciences* **146**.
- Prideaux, G.J. (2006) Mid-Pleistocene vertebrate records: Australia. In 'Encyclopedia of Quaternary Science'. (Ed. SA Elias) pp. 1517–1537. (Elsevier: Oxford)
- Prideaux, G.J., Long, J.A., Ayliffe, L.K., Hellstrom, J.C., Pillams, B., Boles, W.E., Hutchinson, M.N., Roberts, R.G., Cupper, M.L., Arnold, L.J., Devine, P.D., and Warburton, N.M. (2007) An arid-adapted middle Pleistocene vertebrate fauna from south-central Australia. *Nature* **445**, 422-425.
- Prideaux, G.J., and Warburton, N.M. (2008) A new Pleistocene tree-kangaroo (Diprotodontia: Macropodidae) from the Nullarbor Plain of South-Central Australia. *Journal of Vertebrate Paleontology* **28**, 463-478.
- Prideaux, G.J., and Warburton, N.M. (2009) *Bohra nullarbora* sp. nov., a second tree-kangaroo (Marsupialia: Macropodidae) from the Pleistocene of the Nullarbor Plain, Western Australia. *Records of the Western Australian Museum* **25**, 165-179.
- Prideaux, G.J., and Warburton, N.M. (2010) An osteology-based appraisal of the phylogeny and evolution of kangaroos and wallabies (Macropodidae: Marsupialia). *Zoological Journal of the Linnean Society* **159**, 954-987.
- Richards, H.L., Grueter, C.C., and Milne, N. (2015) Strong arm tactics: sexual dimorphism in macropodid limb proportions. *Journal of Zoology* **297**(2), 123-131.
- Schultz, A.H. (1938) The relative length of the regions of the spinal column in Old World primates. *American Journal of Physical Anthropology* **24**, 1-22.
- Shute, E., Prideaux, G.J., and Worthy, T.H. (2016) Three terrestrial Pleistocene coucals (*Centropus*: Cuculidae) from southern Australia: biogeographical and ecological significance. *Zoological Journal of the Linnean Society* **177**, 964-1002.
- Shute, E., Prideaux, G.J., and Worthy, T.H. (2017) Taxonomic review of the late Cenozoic megapodes (Galliformes: Megapodiidae) of Australia. *Royal Society open science* **4**(6), 170233.
- Vizcaino, S.F., Farina, R.A., and Mazzetta, G.V. (1999) Ulnar dimensions and fossoriality in armadillos. *Acta Theriologica* **44**(3), 309-320.
- Warburton, N.M., Bateman, P.W., and Fleming, P.A. (2013) Sexual selection on forelimb muscles of western grey kangaroos (Skippy was clearly a female). *Biological Journal of the Linnean Society* **109**, 923-931.

Warburton, N.M., and Dawson, R. (2015) Musculoskeletal anatomy and adaptations. In 'Marsupials and
Monotremes: Nature's Enigmatic Mammals.' (Eds. A Klieve, S Johnston, P Murray and L Hogan) pp. 53-84.
(Nova Science Publishers)
Warburton, N.M., Gregoire, L., Jacques, S., and Flandrin, C. (2014) Adaptations for digging in the forelimb
muscle anatomy of the southern brown bandicoot (*Isoodon obesulus*) and bilby (*Macrotis lagotis*).
*Australian Journal of Zoology* **61**, 402-419.
Warburton, N.M., Harvey, K.J., Prideaux, G.J., and O'Shea, J.E. (2011) Functional morphology of the
forelimb of living and extinct tree-kangaroos (Marsupialia: Macropodidae). *Journal of Morphology* **272**,
1230-1244.
Warburton, N.M., and Prideaux, G.J. (2010) Functional pedal morphology of the extinct tree-kangaroo
*Bohra* (Diprotodontia: Macropodidae). In 'Macropods: The Biology of Kangaroos, Wallabies and Rat-
kangaroos.' (Eds. G Coulson and MDB Eldridge) pp. 137-151. (CSIRO Publishing)
Warburton, N.M., Yakovleff, M., and Malric, A. (2012) Anatomical adaptations of the hind limb
musculature of tree-kangaroos for arboreal locomotion (Marsupialia: Macropodinae). *Australian Journal of*
*Zoology* **60**, 246-258.
Wells, R.T., and Camens, A.B. (2018) New skeletal material sheds light on the palaeobiology of the
Pleistocene marsupial carnivore, *Thylacoleo carnifex*. *PloS one* **13**, e0208020.
Wells, R.T., and Tedford, R.H. (1995) *Sthenurus* (Macropodidae: Marsupialia) from the pleistocene of Lake
Callabonna, South Australia. *Bulletin of the American Museum of Natural History*(225), 112.
Wilkinson, D.M., and Ruxton, G.D. (2012) Understanding selection for long necks in different taxa.
*Biological Reviews* **87**, 616-630.
Woodburne, M.O. (1984) Families of marsupials: relationships, evolution and biogeography. In 'Mammals:
notes for a short course. Vol. Studies in Geology No. 8.' (Ed. TW Broadhead) pp. 48-71. (University of
Tennessee, Department of Geological Science,)

[revised manuscript text omitted]
 02.7.12 in cranial view; G, distal right tibial epiphysis 02.7.12 in distal view; H, proximal left fibular epiphysis 02.7.12 in medial view; I, proximal right fibular epiphysis 02.7.12 in medial view; J, proximal left fibular epiphysis 02.7.12 in proximal view; K, proximal right fibular epiphysis 02.7.12 in proximal view; L, distal right fibular epiphysis 02.7.12 distal view; M, distal right fibular epiphysis 02.7.12 medial view; N, distal right fibular shaft 02.7.12 lateral view. Scale bar equals 20 mm.

Abbreviations: CAF, calcaneal articular facet; FDE, fibular distal epiphysis; -HFI, head of fibula; ICE, intercondylar eminence; LCO, lateral condyle; LFA, lateral fossa; LMA, lateral malleolus; MCO, medial condyle; MMA, medial malleolus; PFF, proximal fibular facet; PTF, proximal tibial facet; TAS, talar articular surface.

Figure 19. Tarsal elements of *Congruus kitcheneri*. Holotype, WAM 02.7.12. A, calcaneum dorsal view; B, calcaneum plantar view; C, calcaneum medial view; D, calcaneum lateral view; E, calcaneum proximal view; F, calcaneum distal view; G, talus dorsal view; H, talus plantar view; I, talus medial view; J, talus distal view; K, navicular axial view; L, navicular abaxial view; M, navicular proximal view; N, cuboid dorsal view; O, cuboid ventral view; P, cuboid proximal view; Q, cuboid distal view; R, ectocuneiform axial view; S, ectocuneiform abaxial view; T, ectocuneiform proximal view. Scale bar equals 20 mm. Abbreviations: AFN, articular facets for navicular; AHN, head of talus (articular facet for navicular); APT, anterior plantar tuberosity; ATF, anteromedial facet for head of talus; CCA, calcaneal-cuboid articulation; CFA, articular facet for cuboid; CMT, cuboid facet for metatarsal VI; CPS, calcaneal plantar surface; DMF, dorsomedial facet of calcaneal-cuboid articulation; DLF, dorsolateral facet of calcaneal-cuboid articulation; ECF, facet for ectocuneiform; EMT, ectocuneiform facet for metatarsal IV; ENT, facet for entocuneiform; FGS, flexor groove of sustentaculum tali; FIB, fibular articular surface; FIL, tubercles for fibular ligaments; LCS, lateral calcaneal facet; LPT, lateral plantar tubercle; LTF, lateral facet for talus; LTC, lateral trochlear crest; MCF, medial calcaneal facet; MFO, malleolar fossa; MMA, medial malleolus; MTC, medial trochlear crest; MTF,

medial facet for talus; NFT, navicular facet for talus; PVP, posteroventral process; STA, sustentaculum tali;
TCA, tuber calcanei; TCE, epiphysis of tuber calcanei; TPS, transverse plantar sulcus; TRO, trochlea (talo-
crural joint); VMF, ventromedial facet of calcaneal-cuboid articulation.

170x213mm (300 x 300 DPI)

Figure 20. Pedal elements of *Congruus kitcheneri*. Holotype, WAM 02.7.12. A, pes semi-articulated dorsal view; B, metatarsal IV plantar view; C, metatarsal IV medial view; D, metatarsal V plantar view; E, metatarsal V medial view; F, metatarsal V distal view; G, proximal phalanx IV proximal view; H, proximal phalanx IV plantar view; I proximal phalanx IV medial view; J, proximal phalanx V proximal view; K, proximal phalanx V ventral view; L, proximal phalanx V medial view; M, middle phalanx IV proximal view; N, middle phalanx IV ventral view; O, middle phalanx IV medial view; P, middle phalanx V proximal view; Q middle phalanx V ventral view; R, middle phalanx V medial view; S, distal phalanx IV ventral view; T, distal phalanx IV medial view; U, distal phalanx V ventral view; V, distal phalanx V medial view. Scale bar equals 20 mm. Abbreviations: BAS, base; BOD, body; CAL, calcaneum; CUB, cuboid; DP(IV-V), distal phalanges; FTU, flexor tubercle; HMT, head of metatarsal; MIVF, facet for metatarsal IV; MP(IV-V), middle phalanges; MTIV, metatarsal IV; MTV, metatarsal V; MVF, facet for metatarsal V; PLP, posterolateral process; PP(V-IV), proximal phalanges; PSF, facet for plantar sesamoid (proximal plantar process); TRO, trochlea; UNP, ungular process.

Appendix B

Dear Professors Sanchez and Padian

We thank yourselves and the reviewers for taking the time to read through our lengthy manuscript and especially for their very kind appraisals of our work. This piece of work has been in progress for a number of years, and we are thrilled with the positive feedback.

We have addressed the issues raised as described below, and all changes to the manuscript have been highlighted in red font:

Reviewer: 1

Thank you very much for the positive feedback, including those relating to the illustrations. We hope this work will provide a useful resource for other workers in the field.

*REV 1: The included data tables are clear and thorough, however regarding Tables 7, 8 & 9 (comparative postcranial measurements), visually comparing these columns of values is tricky in the current format. If the goal is to compare and contrast various proportions, a suggestion would be to create a simple additional figure presenting key sets of measures as bar or line plots so the relative differences and patterns pertinent to understanding *Congruus kitcheneri* can be more easily appreciated (perhaps something similar to Wells & Camens 2018). No additional analysis is necessary, but just presenting this information graphically rather than (or in addition to) columns of values would more convincingly reinforce the conclusions drawn. This might be particularly interesting for the forelimb (e.g. sexual dimorphism discussed on page 34) and the vertebral column (e.g. semi-arboreal *Congruus* vs terrestrial macropod vertebral regions discussed on page 35), especially if some of the taxa discussed in-text could be represented by data brought in from the supplementary material. I feel a couple of plots would visually support the quantitative arguments made in-text.*

REVIEW: Two additional figures have been produced to highlight the patterns relating to axial skeleton proportions (fig. 5) and phalangeal proportions of the manus and pes (fig. 16A and B). We feel that these provide a good representation of the patterns noted in the text.

REVIEW: We have retained the tables 7-9 to include measurement data of the new *Congruus kitcheneri* material.

I have noted minor typographical and clarity-related comments in the attached PDF, and suggest a thorough copy-edit overall to catch any additional things I may have missed. Overall I do not have substantial recommendations for improvement, and pending the authors' consideration of these minor suggestions, I recommend this paper for publication in Royal Society Open Science.

REVIEW: Thank you very much. We have proof-read the manuscript again and made necessary changes to typographical errors.

Reviewer: 2

This is a very well written and interesting descriptive manuscript and reclassification of a fossil macropodine.

In terms of writing, I didn't find any typos or errors for correction, but I will admit I only skimmed the "Description and comparisons" section, focusing mainly on the elements that were more relevant to the interpretation of semiarborealism. The argument for a semiarboreal lifestyle is supported very well by the descriptions of the forelimbs and grasping digits. However, I would like to see more discussion and comparisons with rock-wallabies in the section of the hindlimbs and pes anatomy, because these have shorter and more flexible feet than other ground dwelling macropods and they weren't mentioned in the comparison (although they were mentioned for the caudal vertebrae). Could *C. kitcheneri* be moving around a rocky environment with uneven ground, using its hands and arms to grasp, stabilise and move between trees and rocky outcrops? If this is unlikely, perhaps due to other evidence of the habitat, then I think this can be discussed and justified a bit in the discussion. Overall, however, I think this manuscript is very well written, and it highlights that macropods have more diverse locomotor adaptations.

REVIEW: Thank you for the positive feedback. We have added in some more specific comments related to morphological comparisons with *Petrogale*, where relevant material could be found. Unfortunately quantitative data was very difficult to collect, and typically the pedal phalanges are retained within the skins in museum collections, and thus are not available to be measured. However, the following specific inclusions have been made:

INNOMINATE: The deeply concave iliac fossa of *C. kitcheneri* is most similar to those of *Pe. xanthopus* and *T. billardierii*, though in *Petrogale* spp. the ilium is strongly curved and flares laterally toward the iliac crest.

FEMUR: In *Petrogale* spp. the femur is long and gracile, and the m. quadratus femoris scar more proximally placed.

TIBIA: The crural index (tibia-to-femur-length ratio) of *C. kitcheneri* is 1.6, which is intermediate between those of large ground-dwelling macropodine taxa (in which it approaches 2.0) and smaller ground-dwelling forms, such as *T. billardierii* (1.3) and *N. eugenii* (1.2) and rock wallabies e.g., *Pe. lateralis* (1.3). In species of *Dendrolagus*, the tibia is shorter than the femur. The proximal tibial epiphysis of *C. kitcheneri* is wider than it is anteroposteriorly long, rather than the reverse, which distinguishes it from the ground-dwelling macropodines, including *Protemnodon*. In this characteristic, it is similar to *D. bennettianus*, in which the anterior tibial crest is very much reduced, resulting in a very short, broad tibial plateau, while in *Petrogale* spp. the length and width of the tibial plateau are close to equal. The relatively larger area of the medial versus lateral condyle is also more like that of *D. bennettianus*. In surface area, the condyles are generally more subequal in size in *M. fuliginosus*, *Os. rufus* and *Pe. lateralis*.

FIBULA: The proximal fibular epiphysis of *C. kitcheneri* is similar to those of large ground-dwelling forms (*M. fuliginosus* and *Os. rufus*) in being fairly low and flat and that the shelf of the articular surface is relatively shallow. In contrast, in *Pe. lateralis* and *S. brachyurus* the articular shelf is more deeply cut and the lateral aspect of the fibular head extends proximally above the articular facet.

METATARSALS: Metatarsal IV of *C. kitcheneri* is shorter (37% relative to femur length) and more robust than in most extant ground-dwelling macropodine taxa. In *M. fuliginosus*, *N. eugenii*, *Os. rufus* and *Pe. xanthopus*, metatarsal IV is long (>40% femur length; *Thylogale* spp. and *Petrogale* spp. 40-45%; *Macropus*, *Notamacropus* and *Osphranter* >50%), straight and gracile, with very limited change in transverse width throughout the shaft in comparison to *C. kitcheneri*. In *S. brachyurus* and *Dendrolagus* spp. the length of the fourth metatarsal relative to femur length is less than 37%.

PEDAL PHALANGES: The ungual process is also not especially long or curved in *Petrogale* spp.

PES FUNCTIONAL INTERPRETATION: Similarly, the shortened pes with short and robust metatarsals (relative to both calcaneal and femur length), is reminiscent of tree-kangaroos, or at least intermediate between rock-wallabies (*Petrogale* spp.) and tree-kangaroos (*Dendrolagus* spp.).